# MM-HELIX: Boosting Multimodal Long-Chain Reflective Reasoning with Holistic Platform and Adaptive Hybrid Policy Optimization

**Xiangyu Zhao**[*,1,2], **Junming Lin**[*,2,3], **Tianhao Liang**[*,4], **Yifan Zhou**[*,1,2], **Wenhao Chai**[5],
**Yuzhe Gu**[2], **Weiyun Wang**[2], **Kai Chen**[2], **Gen Luo**[2], **Wenwei Zhang**[2], **Junchi Yan**[1],
**Hua Yang**[1], **Haodong Duan**[2✉], **Xue Yang**[1✉]

[1]ICISEE & SAIS & SCS & SAI, Shanghai Jiao Tong University, [2]Shanghai AI Laboratory,
[3]Beijing University of Posts and Telecommunications, [4]Zhejiang University, [5]Princeton University
*Equal contribution   ✉Corresponding author
https://mm-helix.github.io/

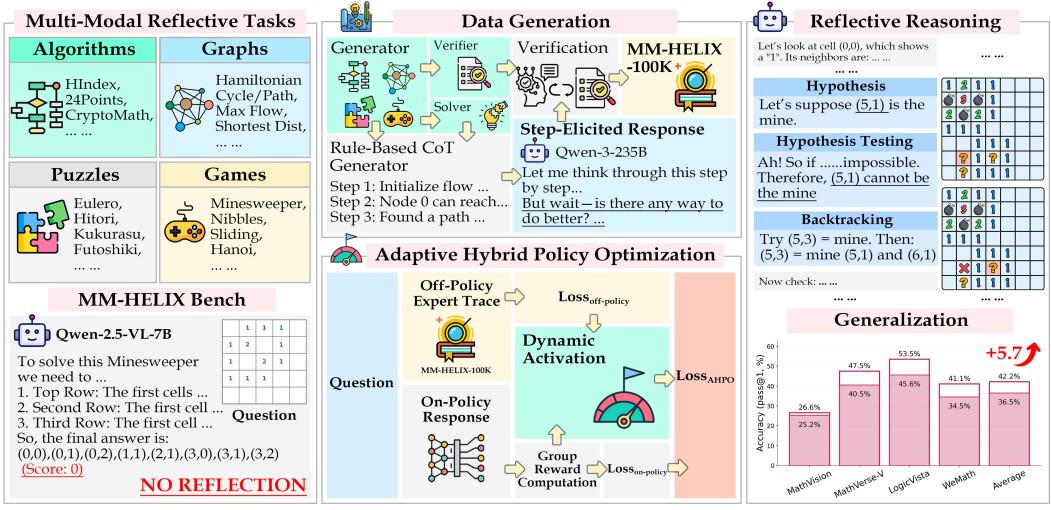

Figure 1: Overview of proposed framework. Our framework comprises two core components: (1) MM-HELIX benchmark to evaluate the reflective capabilities of MLLM, and (2) AHPO method to boost reflection capability and transfer enhanced skills to general reasoning tasks.

## Abstract

While current Multimodal Large Language Models (MLLMs) have demonstrated proficiency in reasoning tasks such as mathematics and logic, their capacity for **long-chain reflective reasoning**, a prerequisite for solving complex real-world problems, remains largely underexplored. In this work, we first conduct an extensive empirical investigation to evaluate this capability. Leveraging a carefully designed data synthesis engine, we construct MM-HELIX, a multimodal benchmark consisting 1,260 samples of 42 challenging synthetic tasks that require iterative thinking and backtracking. Empirical results on this benchmark reveal that existing MLLMs exhibit significant performance deficits in long-chain reflective reasoning. To address this limitation, we generate post-training data and further explore learning paradigms for exploiting such data. We first develop the Step-Elicited Response Generation pipeline to create MM-HELIX-100K, a large-scale dataset of 100k high-quality, reflective reasoning traces for instruction-tuning stage. Given that standard Reinforcement Learning fails on complex tasks due to sparse reward signals and catastrophic forgetting after Supervised Fine-Tuning, we propose Adaptive Hybrid Policy Optimization (AHPO), a novel training strategy that dynamically unifies offline supervision and online optimization into a single stage. This strategy enables the model to learn from expert data when rewards are sparse and conduct independent exploration once proficient. When applied to the Qwen2.5-VL-7B baseline, our method achieves a +18.6% accuracy improve-

ment on MM-HELIX benchmark and demonstrates strong generalization with a +5.7% average performance gain on general mathematic and logic tasks. Our work demonstrate that reflective reasoning in MLLMs can be effectively learned and generalized, paving the way for developing more capable MLLMs.

# 1 INTRODUCTION

Human cognition is fundamentally characterized by the processes of reflection and backtracking. This iterative cycle of trial, error, and correction allows individuals to adapt to novel environments and progressively refine their decisions for greater accuracy. Inspired by this cognitive process, recent advancements in Large Language Models (LLMs) have integrated reflective and multi-step thinking strategies (Guo et al., 2025a), unlocking significant improvements in their reasoning abilities. Concurrently, Multimodal Large Language Models (MLLMs) have undergone rapid development, achieving impressive performance across a spectrum of downstream tasks, from perception (e.g., recognition) to reasoning (e.g., mathematic). Despite these advances, a significant limitation persists in current MLLMs. The majority of these models are designed to generate outputs in a single, direct pass, lacking the intrinsic mechanisms for self-correction and iterative refinement. Consequently, their capacity for end-to-end, multi-step reflective reasoning within rich multimodal contexts remains largely unexplored and underevaluated.

Existing research, e.g. Enigmata (Chen et al., 2025), VGRP-Bench (Ren et al., 2025), and Code2Logic (Tong et al., 2025), have primarily concentrated on text-only problems or puzzle-like challenges that are often constrained to multiple-choice or fill-in-the-blank formats, thereby failing to adequately evaluate the end-to-end reflective reasoning capabilities of MLLMs. To address this critical research gap, we introduce MM-HELIX, a comprehensive benchmark designed to evaluate the long-chain iterative reasoning capabilities of MLLMs. MM-HELIX contains 42 meticulously curated challenging tasks from diverse online sources, categorized into four domains: *Algorithm*, *Graph*, *Puzzle*, and *Game*. Each task requires the model to perform careful visual observation, develop a deep understanding of complex rules, and generate an extended chain-of-thought that necessitates reflection and backtracking. We constructed a versatile procedural generation pipeline to systematically generate samples. This pipeline features: (1) a rule-based code *Generator* programmatically creates multimodal questions with tunable parameters, spanning five hierarchical difficulty levels, which indicates very easy, easy, middle, hard, very hard. (2) a *Solver* module engineered to produce ground-truth solutions; and (3) a *Verifier* module to algorithmically validate answers for tasks with non-unique solutions. This Verifier also functions as a reward oracle in our reinforcement learning environment. Following a rigorous filtering process, our benchmark consists of 1,260 high-quality samples. Our comprehensive evaluation reveals that state-of-the-art MLLMs struggle significantly on MM-HELIX. For instance, even a leading model like Qwen-2.5-VL-72B achieves a mere 13.9% accuracy, underscoring a profound deficit in their reflective reasoning capabilities.

Based on the observation, we wonder if we can boost the reflection of MLLMs within MM-HELIX and generalize to general reasoning tasks like mathematics and logic. We then propose the Step-Elicited Response Generation (SERG) pipeline, a method for efficiently generating high-quality, reflective CoT traces by integrating rule-based, key-step knowledge. Leveraging SERG, we construct MM-HELIX-100K, a large-scale dataset comprising 100k high-quality samples that span 42 tasks across a full spectrum of difficulty levels.

Our initial experiments reveal the limitations of standard training paradigms: instruction-tuning on MM-HELIX-100K caused catastrophic forgetting, while on-policy reinforcement learning failed due to extreme reward sparsity since base model lacks the foundational ability to solve the tasks. To this end, we introduce Adaptive Hybrid Policy Optimization (AHPO), a framework that dynamically integrates off-policy expert guidance with on-policy exploration. AHPO implements an explore-with-supervision strategy by dynamically modulating the off-policy loss via a reward-based gating mechanism. Specifically, when the rewards within a group are sparse, indicating the model is struggling, the off-policy expert data is integrated to guide the model toward correct trajectories. Conversely, once the model demonstrates proficiency and rewards become dense, the off-policy loss is attenuated, encouraging the policy to explore and discover novel solutions. Training with AHPO on a combination of MM-HELIX-100K and a general mathematics RL dataset yielded substantial gains. The model not only demonstrated mastery on in-domain tasks, achieving a +18.6% accuracy

improvement on MM-HELIX, but also successfully **generalized its enhanced reflective skills to general math and logic tasks, with a +5.7% average increase in accuracy.** These results validate that our approach effectively cultivates reflective capabilities that are both robust and transferable.

Our contributions can be summarized as follows:

1. We introduce MM-HELIX, a benchmark comprising 42 challenging multimodal tasks specifically designed to assess long-chain, iterative, and reflective reasoning. Through systematically evaluation, we reveal the critical deficiencies of state-of-the-art MLLMs in these complex reasoning domains.

2. We propose the Step-Elicited Response Generation (SERG) pipeline, a novel and efficient method for generating high-quality demonstration data. We leverage SERG to construct MM-HELIX-100K, a large-scale dataset of 100k multimodal high-quality reflective Chain-of-Thought (CoT) traces.

3. We introduce Adaptive Hybrid Policy Optimization (AHPO), a novel training algorithm that dynamically integrates off-policy expert data with on-policy exploration in a single stage. This hybrid approach is specifically designed to overcome the challenges of sparse rewards, fostering the acquisition of complex reasoning skills that demonstrate substantial generalization.

## 2 RELATED WORK

**Multimodal Large Language Models.** In recent years, MLLMs have rapidly advanced in both general multimodal capabilities and specialized reasoning. Representative models such as Gemini 2.5 (Comanici et al., 2025), Qwen-2.5-VL (Bai et al., 2025b), and InternVL3 (Wang et al., 2025) establish strong general multimodal capabilities. More recently, reasoning-oriented models such as GLM-4.5V-Thinking (Team et al., 2025b), Seed1.5-VL (Seed et al., 2025), and Kimi-VL-A3B-Thinking (Team et al., 2025a) explicitly emphasize structured thinking. Together, these works indicate a growing consensus that reasoning is the next frontier for MLLMs.

**Exploration of Long-chain Reasoning.** Chain-of-Thought prompting (CoT) (Wei et al., 2022) and Tree-of-Thoughts (ToT) (Yao et al., 2023) demonstrate the value of intermediate reasoning traces. Procedural generation has emerged as a solution: Enigmata (Chen et al., 2025) creates logic puzzles, Code2Logic (Tong et al., 2025) synthesizes multimodal QA from game logic, and benchmarks such as VGRP-Bench (Ren et al., 2025) reveal persistent weaknesses in algorithmic reasoning. However, these works have primarily concentrated on text-only problems or puzzle-like challenges that are often constrained to multiple-choice or fill-in-the-blank formats.

**Reinforcement Learning Method.** On-policy algorithms, such as PPO (Schulman et al., 2017), stabilize training by clipping updates, but this is computationally expensive. Variants such as GRPO (Shao et al., 2024) improve stability through within-group advantage, while DAPO (Yu et al., 2025a) dynamically adjusts policy optimization, and GSPO (Zheng et al., 2025) emphasizes gradient scaling for more efficient updates. Off-policy methods reduce training costs by reusing data. Besides, LUFFY (Yan et al., 2025) applies sequence-level optimization to exploit offline preference datasets. Those RL methods all meet problems of inefficient training when facing hard tasks, thus, we propose AHPO to simplify training and enhance multimodal reasoning.

## 3 METHOD

### 3.1 MM-HELIX: BENCHMARKING MULTIMODAL REFLECTIVE END-TO-END REASONING

Recent advancements in Multimodal Large Language Models (MLLMs) (Comanici et al., 2025; Bai et al., 2025b; Wang et al., 2025; Luo et al., 2025b;a) have demonstrated remarkable capabilities, yet a significant limitation persists in their capacity for complex, multi-step reflective reasoning. Existing benchmarks often focus on direct inference tasks, such as mathematical and logical problem-solving, overlooking the evaluation of long-chain visual reasoning processes in an end-to-end manner. To address this gap, we introduce MM-HELIX, a novel benchmark specifically designed to assess and challenge the limits of multimodal reflective reasoning in MLLMs. The construction of this benchmark is guided by four core principles: Multimodal, Long-Chain Reasoning, Reflection, and End-to-End. To instantiate these principles, we have curated 42 diverse tasks from public web resources and existing academic datasets, which are organized into four categories: algorithms,

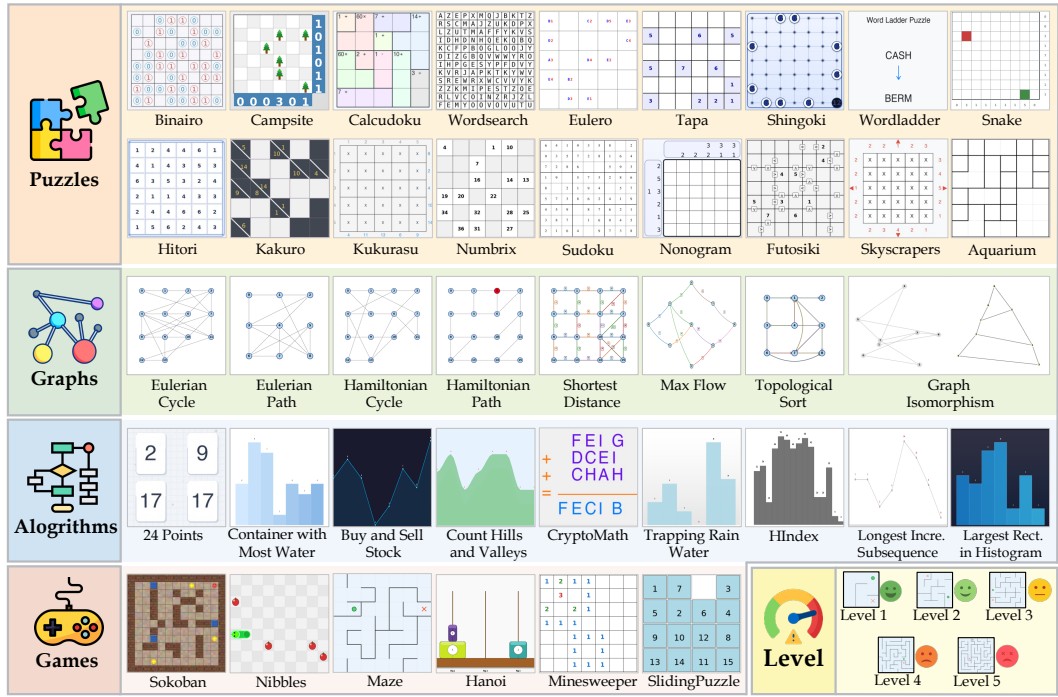

Figure 2: Overview of tasks in MM-HELIX benchmark. MM-HELIX contains 42 challenging tasks designed to evaluate long-chain reflective reasoning across five progressive levels of difficulty.

graphs, puzzles, and games, as shown in Fig. 2. Each task necessitates that the model comprehend complex rules, recognize states within a visual context, and engage in a sequential process of thought, reflection, and backtracking to reach a solution, presenting a substantial challenge to current MLLMs.

To ensure the scalability, diversity, and controlled difficulty of our benchmark, we develop a procedural generation framework. This framework is architected around three core components: an Instance Generator, a deterministic Solver, and an automated Verifier. The Instance Generator produces problem instances based on task-specific rules and scalable parameters. Each generated instance comprises three elements: *Question Description*: A textual prompt outlining the task with its corresponding detailed rules. *Visual Input*: An image presenting the initial problem scenario (e.g., a game board). *Initial State*: A structured data representation of the visual input to facilitate post-evaluation verification. An example is shown in Fig. 3; see Appx. A.7 for all cases.

For each generated instance, the Solver first analyzes the initial state using a rule-based algorithm to determine the instance's solvability. If a solution is deemed to exist, the Solver produces a feasible solution to serve as the ground truth. While tasks in the algorithm and graph categories typically have a unique solution, game and puzzle tasks often permit multiple valid solutions. To facilitate objective and accurate evaluation, we construct an automated Verifier to assess model outputs. This component employs two distinct validation strategies based on the complexity of the required response. For tasks with simple, discrete answers (e.g., a boolean or a numerical value), it performs a direct exact-match comparison against the ground truth. For tasks requiring complex, multi-step solutions, the Verifier first standardizes the model's output

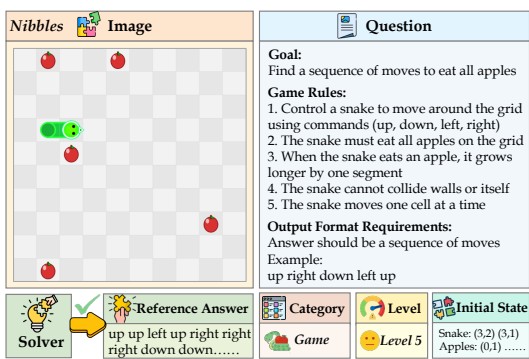

Figure 3: Example of Nibbles task (Level 5) in MM-HELIX benchmark. The snake must eat all apples on the grid by executing a sequence of moves, demanding long-term reflection.

and then simulates the proposed sequence of actions from the Initial State, leveraging the problem's intrinsic rules to confirm the solution's validity.

A key feature of MM-HELIX is its hierarchical difficulty system, designed for the fine-grained evaluation of model capabilities. We scale task difficulty by programmatically adjusting task-specific parameters within the generation framework, primarily by controlling the number of reasoning steps required for a correct solution. By modulating these parameters, we generate tasks across five distinct difficulty levels ranging from Level 1 (very easy) to Level 5 (very hard), where both the problem's scale and reasoning complexity increase with each level. This tiered structure enables a precise identification of the performance degradation threshold for a given model, thereby revealing the limitations of its reasoning capacity. The final evaluation set comprises 1,260 unique instances, a corpus size selected to ensure statistical robustness while maintaining computational tractability. The dataset is balanced across both tasks and difficulty levels: for each of the 42 tasks, we generated 30 instances by sampling 6 instances from each of the 5 difficulty levels. This composition facilitates a reliable and granular assessment of model performance across a wide spectrum of complexity.

## 3.2 MM-HELIX-100K: GUIDING MULTIMODAL REFLECTIVE REASONING

The capability for long-chain reflection is crucial in various advanced applications. However, our evaluation results on MM-HELIX reveal that even state-of-the-art MLLMs, such as Qwen-2.5-VL-72B, struggle significantly with these challenging reflective tasks, achieving an accuracy of only approximately 10%. This performance gap motivates our investigation into whether targeted instruction tuning can enhance this reflective capability and if such improvements can generalize to other complex reasoning domains, such as mathematics and logic.

To effectively train models for such complex, long-chain reasoning, a large-scale, high-quality dataset of reasoning trajectories is indispensable. To this end, we introduce MM-HELIX-100K, a meticulously curated dataset for instruction-tuning comprising 100k instances. The dataset spans 42 distinct tasks and incorporates high-quality responses with reflection.

Generating high-quality Chain-of-Thought (CoT) trajectories at this scale presents a formidable challenge. Conventional methods, such as prompting a large model to generate reasoning steps from scratch in an unconstrained manner, are often inefficient and yield low-quality results. To overcome these challenges, we develop a hybrid and highly efficient data generation pipeline, which we term Step-Elicited Response Generation (SERG), as the pipeline shown in Fig. 4. The process begins with our task-specific generators

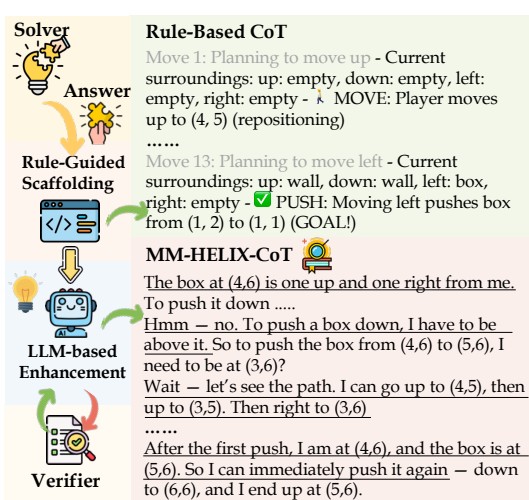

Figure 4: Demonstration of our Step-Elicited Response Generation pipeline.

creating a base set of 150k problem instances. We first employ a programmatic, rule-based CoT constructor to generate a deterministic, skeletal reasoning path by strategically embedding anchors—critical intermediate states or calculations—and connecting them with template-based natural language descriptions. This initial step produces a logically sound but often mechanical and rigid reasoning trace, which is suboptimal for training a nuanced language model.

This rule-based trajectory then serves as a high-quality scaffold for a powerful model, in this work Qwen3-235B. We provide the model with the original question and the rule-based reasoning path, prompting it to refine this scaffold into a more natural, comprehensive, and human-like reasoning process that includes reflective steps. This enhancement phase enriches the dataset with linguistic diversity and more detailed explanations. To guarantee the final dataset's integrity, each generated trajectory is only accepted if its final answer passes the corresponding automated verifier. This stringent filtering mechanism is crucial for eliminating any errors introduced during the LLM enhancement phase and ensures the high fidelity of the training data.

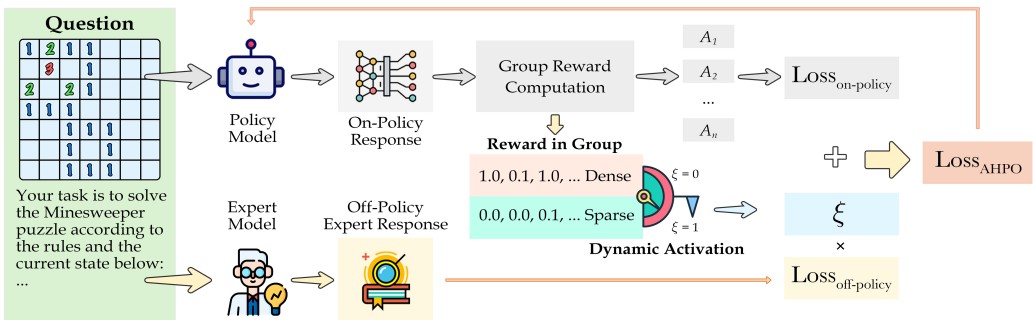

Figure 5: Demonstration of Adaptive Hybrid Policy Optimization (AHPO). AHPO dynamically integrates off-policy expert guidance with on-policy exploration, leading to performance generalization.

### 3.3 AHPO: ADAPTIVE HYBRID ALGORITHM FOR GENERALIZING REFLECTION

On-policy reinforcement learning algorithm, such as GRPO, update its model exclusively from data generated by the current policy. In complex task domains like MM-HELIX, the policy seldom generates successful trajectories, leading to severe reward sparsity that renders the training process inefficient and often ineffective (see GRPO in Fig. 6). A common strategy to mitigate this is to initialize the policy via Supervised Fine-Tuning (SFT) on an offline dataset of expert demonstrations. However, this methodology can induce a significant distributional shift, biasing the policy towards the SFT data distribution and constraining its ability to generalize and adapt during the subsequent RL phase.

To overcome these limitations, we introduce **Adaptive Hybrid Policy Optimization (AHPO)**, a novel algorithm that integrates off-policy and on-policy learning into a unified training framework shown in Fig. 5. The cornerstone of our method is an adaptive mechanism that modulates the influence of offline expert data based on the policy's real-time performance. This allows the model to leverage expert guidance when needed and to rely on its own exploration as it improves. We show the comparison of integrating off-policy loss to GRPO (called static-AHPO) in Fig. 6.

The AHPO objective function dynamically combines a standard off-policy loss with an on-policy GRPO-style objective. The off-policy component is negative log-likelihood loss on expert data $y^*$ :

$$\mathcal{L}_{\text{off}-\text{policy}}(\theta) = -\frac{1}{|y^*|} \sum_{t=1}^{|y^*|} \log \pi_\theta(y_t^*|x, y_{<t}^*).$$ (1)

The on-policy component is a clipped policy gradient objective:

$$\mathcal{L}_{\text{on}-\text{policy}}(\theta) = -\frac{1}{\sum_{i=1}^{N}|\tau_i|} \sum_{i=1}^{N} \sum_{t=1}^{|\tau_i|} \text{CLIP}(r_{i,t}(\theta), A_i, \epsilon),$$ (2)

$$A_i = \frac{R(\tau_i) - \text{mean}(\{R(\tau_i) \mid \tau_i \sim \pi_{\theta_{\text{old}}}(\tau), i = 1, 2, \ldots, N\})}{\text{std}(\{R(\tau_i) \mid \tau_i \sim \pi_{\theta_{\text{old}}}(\tau), i = 1, 2, \ldots, N\})},$$ (3)

where $A_i$ represents the estimated advantage for trajectory $\tau_i$, and $r_{i,t}(\theta) = \pi_\theta(\tau_{i,t}|q, \tau_{i,<t})/\pi_{\theta_{\text{old}}}(\tau_{i,t}|q, \tau_{i,<t})$ is the probability ratio for importance sampling. Following (Yu et al., 2025a), we omit the KL divergence term from the original GRPO formulation to reduce constraints on policy exploration and decrease computational overhead; we also remove the CLIP module for more efficient training.

AHPO unifies these objectives into a single loss function, where the influence of the off-policy term is governed by an adaptive coefficient $\xi$:

$$\mathcal{L}_{\text{AHPO}}(\theta) = \xi \mathcal{L}_{\text{off}-\text{policy}}(\theta) + \mathcal{L}_{\text{on}-\text{policy}}(\theta)$$ (4)

$$= -\frac{1}{Z}(\underbrace{\sum_{i=1}^{N_{\text{off}}} \sum_{t=1}^{|y_i^*|} \xi \log \pi_\theta(y_{i,t}^*|x_i, y_{i,<t}^*)}_{\text{Off-policy objective}} + \underbrace{\sum_{i=1}^{N_{\text{on}}} \sum_{t=1}^{|\tau_i|} \text{CLIP}(r_{i,t}(\theta), A_i, \epsilon))}_{\text{On-policy objective}},$$ (5)

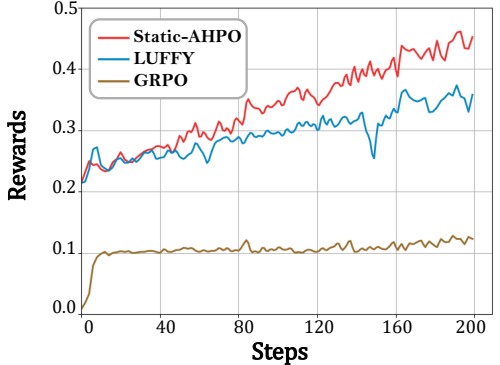
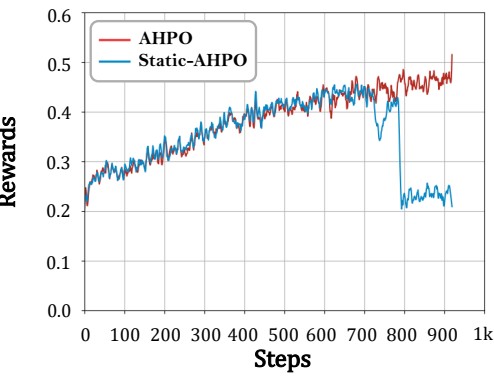

Figure 6: Comparison of GRPO, LUFFY and Static-AHPO. Static-AHPO achieves best performance on challenging tasks.

Figure 7: Comparison of Static-AHPO and AHPO. AHPO dynamically integrates expert data to ensure a robust training.

the activation coefficient $\xi$ is controlled by the following adaptive rule:

$$\xi = \mathbf{1}\left(\sum_{i=1}^{N_{\text{on}}} \mathbb{I}(R(\tau_i) = 1) < \hat{R}\right). \tag{6}$$

Here, $\mathbb{I}(\cdot)$ is the indicator function and $\hat{R}$ is a predefined success rate threshold. This mechanism conditionally applies supervision from expert data: it provides dense guidance when the model's on-policy success rate is below the threshold $\hat{R}$, preventing the agent from getting stuck or hacking in early training. Conversely, as the policy improves and consistently achieves high rewards, the off-policy supervision is deactivated ($\xi = 0$). This adaptive strategy ensures that expert guidance is present during the crucial initial stages of learning but fades out to allow the model to refine its policy through pure exploration. This prevents the model from merely memorizing the expert distribution and encourages the discovery of more robust solutions.

While prior work, such as LUFFY (Yan et al., 2025), has used expert data as positive examples in a preference-based RL framework, our empirical results demonstrate that this approach is less effective than our adaptive loss formulation for the complex tasks. Furthermore, the activation coefficient $\xi$ plays a key role in making robust training. Although a static coefficient provides strong initial guidance, it creates a persistent conflict between the off-policy expert distribution and the models's evolving on-policy distribution. This mismatch can destabilize training and even lead to performance degradation once the model has surpassed the proficiency of the expert data, as shown in Fig. 7.

For our off-policy expert data, we utilize the high-quality CoT trajectories from the MM-HELIX-100K dataset. By dynamically balancing the exploitation of this expert data with on-policy exploration, AHPO effectively learns the reflective reasoning capabilities required by MM-HELIX benchmark and successfully generalizes these skills to broader reasoning domains, leading to significant performance enhancements in tasks involving mathematics and logic.

## 4 EXPERIMENT

### 4.1 EVALUATION RESULTS ON MM-HELIX

Our comprehensive evaluation of 23 leading MLLMs on MM-HELIX benchmark, with full results detailed in Tab. 1, reveals critical limitations in the reasoning capabilities of current models. The evaluation settings are detailed in Appx. A.3. The analysis yields three primary findings:

First, a profound deficit exists in multimodal reflective reasoning. Even the most advanced proprietary model, GPT-5 can only achieve 58.1% accuracy, with no other model surpassing the 50% threshold. This performance gap is even more pronounced for open-source models; the leading contender, Intern-S1-241B, reaches just 33.3% accuracy. The importance of this targeted capability is also underscored by the fact that models capable of iterative reflection systematically outperform their non-reflective

Table 1: Evaluation results on MM-HELIX across both multimodal and text-only settings. These results underscore the ongoing difficulty MLLMs face with complex, long-chain reflective tasks. Thinking models with reflective reasoning capabilities generally achieve higher scores than those without. Furthermore, a significant modality gap is observed where text-only inputs are superior.

| Model | Thinking | Breakdown by Category | | | | | | | | Overall | |
| | | Algorithms | | Graphs | | Puzzles | | Games | | | |
| | | Txt | Img | Txt | Img | Txt | Img | Txt | Img | Txt | Img |
| --- | --- | --- | --- | --- | --- | --- | --- | --- | --- | --- | --- |
| *Proprietary Models* | | | | | | | | | | | |
| GPT-5 (OpenAI, 2025b) | ✓ | 83.0 | 88.5 | 98.3 | 50.4 | 80.9 | 52.6 | 80.0 | 40.0 | 84.5 | 58.1 |
| Seed-1.5-VL (Guo et al., 2025b) | ✓ | 89.3 | 78.9 | 86.7 | 40.4 | 51.6 | 41.9 | 55.6 | 33.3 | 66.9 | 48.3 |
| o4-mini (OpenAI, 2025c) | ✓ | 76.3 | 50.7 | 95.0 | 42.1 | 69.1 | 45.8 | 66.7 | 35.6 | 75.2 | 44.7 |
| Gemini-2.5-Flash (Comanici et al., 2025) | ✓ | 92.6 | 66.7 | 88.3 | 40.8 | 52.1 | 36.7 | 49.4 | 28.3 | 67.3 | 42.7 |
| GPT-4.1 (OpenAI, 2025a) | ✗ | 61.9 | 44.4 | 73.8 | 35.0 | 30.9 | 16.8 | 13.9 | 8.9 | 43.3 | 25.1 |
| GPT-4o (OpenAI, 2024) | ✗ | 33.7 | 18.9 | 44.6 | 25.4 | 10.2 | 4.2 | 10.6 | 6.7 | 21.8 | 11.7 |
| *Open-Source Models* | | | | | | | | | | | |
| Intern-S1-241B-A28B (Bai et al., 2025a) | ✓ | 75.2 | 69.3 | 76.7 | 30.0 | 35.3 | 23.5 | 26.1 | 15.0 | 50.4 | 33.3 |
| GLM-4.5V-106B-A12B-Thinking (Team et al., 2025b) | ✓ | 49.6 | 29.3 | 40.4 | 11.3 | 15.3 | 20.2 | 12.2 | 13.9 | 27.0 | 19.5 |
| Kimi-VL-16B-A3B-Thinking-2506 (Team et al., 2025a) | ✓ | 45.9 | 36.3 | 49.6 | 23.3 | 9.6 | 10.4 | 10.6 | 7.2 | 28.9 | 19.3 |
| GLM-4.1V-9B-Thinking (Team et al., 2025b) | ✓ | 38.1 | 30.7 | 50.4 | 29.2 | 11.6 | 7.4 | 5.0 | 6.1 | 23.7 | 16.3 |
| Qwen-2.5-VL-72B (Bai et al., 2025b) | ✗ | 24.4 | 18.5 | 42.1 | 25.8 | 8.2 | 3.9 | 5.6 | 7.2 | 20.1 | 13.9 |
| Qwen-2.5-VL-32B (Bai et al., 2025b) | ✗ | 22.2 | 15.2 | 46.3 | 22.5 | 8.1 | 4.7 | 5.6 | 6.7 | 20.6 | 12.3 |
| QVQ-72B-Preview (Team, 2024) | ✓ | 22.6 | 21.1 | 36.7 | 16.7 | 4.9 | 3.3 | 6.7 | 3.3 | 17.7 | 11.1 |
| MiniCPM-V-4.5-8B (Yu et al., 2025b) | ✓ | 20.0 | 20.0 | 32.1 | 20.8 | 5.8 | 3.7 | 0.0 | 3.3 | 13.0 | 10.4 |
| InternVL3-78B (Zhu et al., 2025) | ✗ | 20.0 | 14.4 | 43.3 | 25.4 | 10.2 | 4.0 | 10.0 | 1.1 | 18.6 | 9.9 |
| InternVL3-38B (Zhu et al., 2025) | ✗ | 19.3 | 14.1 | 40.8 | 22.5 | 8.2 | 3.5 | 7.8 | 5.6 | 16.7 | 9.7 |
| Llama-4-Scout-109B-A17B-16E (Meta, 2025) | ✗ | 24.1 | 16.3 | 40.8 | 21.3 | 4.4 | 4.2 | 2.2 | 1.7 | 15.2 | 9.7 |
| Ovis2-34B (Lu et al., 2024) | ✗ | 14.4 | 10.4 | 33.8 | 22.1 | 3.9 | 1.2 | 5.0 | 1.7 | 12.0 | 7.2 |
| Gemma-3-27B-IT (Team, 2025) | ✗ | 20.7 | 10.4 | 44.2 | 22.1 | 6.5 | 0.5 | 5.6 | 1.7 | 16.6 | 6.9 |
| Qwen-2.5-VL-7B (Bai et al., 2025b) | ✗ | 5.6 | 5.9 | 25.4 | 17.9 | 0.4 | 0.4 | 0.6 | 1.1 | 8.0 | 6.3 |
| InternVL3-8B (Zhu et al., 2025) | ✗ | 8.1 | 5.9 | 28.8 | 16.7 | 1.6 | 0.7 | 1.1 | 1.1 | 8.1 | 4.9 |
| Ovis2-8B (Lu et al., 2024) | ✗ | 7.8 | 3.3 | 24.2 | 15.4 | 0.5 | 0.2 | 1.1 | 0.6 | 6.7 | 3.8 |
| *Ours* | | | | | | | | | | | |
| MM-HELIX-7B-Thinking | ✓ | 32.2 | 34.8 | 27.5 | 19.2 | 16.3 | 25.3 | 16.1 | 16.7 | 21.8 | 24.9 |

counterparts. For instance, a powerful model like InternVL-3-78B, despite its strong performance on general benchmarks, scores a mere 9.9%. This stark contrast validates that MM-HELIX successfully isolates and measures this critical reasoning skill.

Second, models excel at structured computation but falter in tasks requiring dynamic state tracking. Models demonstrated the highest proficiency on Algorithm tasks, which primarily involve mathematical computation. Performance was moderate on Graph and Puzzle tasks, while the weakest results were observed in Game task. This trend suggests that while current MLLMs are adept at executing well-defined calculations, they lack robustness in adhering to complex instructions and performing the iterative state-tracking inherent to strict rules.

Besides, significant modality gap still exists between text and vision. When problems were presented in a text-only format, performance improved dramatically. GPT-5's accuracy, for example, surged from 58.1% on the multimodal tasks to 84.5% on their text-only equivalents. This significant performance drop highlights a persistent gap between language and visual inputs.

## 4.2 MAIN RESULTS OF AHPO

We benchmark our proposed AHPO against a comprehensive set of baselines( based on Qwen-2.5VL-7B), including pure RL (GRPO), SFT, a sequential SFT+RL pipeline, and an alternative hybrid algorithm LUFF, with training settings detailed in Appx. A.3. As shown in Tab. 2, the results strongly demonstrate the superiority of our method. **On MM-HELIX, AHPO achieves the highest accuracy of 24.9% among all methods, representing a substantial +18.6% point improvement** over the base model Qwen2.5-VL-7B. Notably, this performance also exceeds that of significantly larger, state-of-the-art models such as Qwen2.5-VL-72B and GLM-4.5V-106B. More importantly, AHPO demonstrates remarkable generalization of its learned reflective reasoning capabilities. When trained on a mix including the MMK12 RL dataset which lacks explicit CoT traces, the model still

Table 2: Comparison of AHPO and other training strategies. AHPO achieves significant improvement on MM-HELIX while also showing great performance transfer to general mathematics and logic tasks, indicating a robust enhancement of both specialized and generalized reasoning abilities.

| Method | Type | In-Domain | General Reasoning | | | | |
|---|---|---|---|---|---|---|---|
| | | MM-HELIX | MathVision | MathVerse-V | LogicVista | WeMath | Average |
| Qwen2.5VL-7B | Baseline | 6.3 | 25.2 | 40.5 | 45.6 | 34.5 | 36.5 |
| +GRPO | On-policy | 9.0(+2.7) | 25.8 | 41.0 | 43.6 | 36.4 | 36.7(+0.2) |
| +SFT | Off-policy | 23.8(+17.5) | 21.7 | 33.0 | 38.7 | 26.2 | 29.9(-6.6) |
| +SFT&GRPO | Sequential | 23.3(+17.0) | 25.9 | 39.1 | 45.9 | 35.7 | 36.7(+0.2) |
| +LUFFY | Hybrid | 9.1(+2.8) | 26.0 | 37.9 | 42.7 | 34.8 | 35.4(-1.1) |
| **+AHPO (Ours)** | Hybrid | **24.9(+18.6)** | **26.6** | **47.5** | **53.5** | **41.1** | **42.2(+5.7)** |

Table 3: Comparison of CoT generation methods cost. Our hybrid approach significantly save the generation cost and make less redundancy.

| Method | Pass@16 (%) | Inf. Time (hrs) | Avg. Len. (tokens) |
|---|---|---|---|
| Model Rollout | 25.00 | ~311.96 | 7140.59 |
| **SERG** | **99.80** | **~27.78** | **5500.53** |

Table 4: Efficiency of our dataset in SFT stage. Our method outperforms Rule-Based CoT, indicates great quality of our generation method.

| Method | Puzzle | Game | Algorithm | Graph | Overall |
|---|---|---|---|---|---|
| Rule-Based | 19.3 | 11.1 | 22.6 | 19.6 | 18.9 |
| **Ours** | **23.5** | **16.7** | **32.6** | **20.4** | **23.8** |

learns to apply reflective inference on out-of-domain tasks. This is attributed to AHPO's explore-with-supervision mechanism, which fosters intrinsic reasoning skills rather than mere mimicry. Consequently, **AHPO achieves an average performance gain of +5.7% points across general mathematics and logic tasks**, validating its ability to transfer complex reasoning skills to entirely new domains.

In contrast, the baseline methods reveal critical limitations. The RL-only approach (GRPO) shows negligible improvement on both task sets, failing to learn effectively from the sparse reward signals. While SFT significantly boosts in-domain performance, achieving comparable performance with AHPO, it induces catastrophic forgetting, leading to a substantial performance degradation on the general reasoning tasks. Sequentially applying GRPO after SFT fails to recover this deficit, indicating that the reflective skills learned via fine-tuning do not effectively transfer to out-of-domain problems within this paradigm. LUFFY, which mitigates sparse rewards by substituting policy rollouts with expert data, shows minor gains but remains significantly less effective than AHPO in both performance and generalization. These findings underscore the superior efficacy and generalization capacity of AHPO's unified training strategy compared to sequential pipelines and other hybrid methods.

## 4.3 COMPARISON OF GENERATION PIPELINE

To validate the effectiveness of our SERG pipeline, we conducted a comparative analysis against two baselines. First, we compared SERG's efficiency and output quality against direct roll-outs from a powerful LLM (Qwen3-235B). For this, we prompted the model to generate reasoning trajectories for 1,000 samples without guidance. As detailed in Tab. 3, this unconstrained approach was computationally prohibitive, incurring substantial time costs and producing highly redundant responses. In contrast, SERG demonstrated vastly superior performance, reducing the generation time by 90% while yielding significantly more concise and structured reasoning traces. econd, we evaluated the downstream utility of SERG-generated data. We fine-tuned a model using 22k samples generated by SERG and compared its performance against a model trained on an equivalent amount of data generated by a purely rule-based method. As shown in Tab. 4, the model trained on SERG data outperformed the rule-based baseline by 4.9%. This result confirms that SERG produces data of a higher quality, leading to more effective downstream model training. Collectively, these experiments validate that SERG strikes an optimal balance between generation efficiency and data quality.

Table 5: Performance of each data component in training data. By utilizing AHPO with MM-HELIX dataset which has no overlap with the mathematical content MMK12, the model achieves much better performance on both in-domain and general benchmark.

| Method | MM-HELIX | MathVision | MathVerse-V | LogicVista | WeMath |
|---|---|---|---|---|---|
| Qwen2.5VL-7B | 6.3 | 25.2 | 40.5 | 45.6 | 34.5 |
| +MMK12 Only | 5.5 | 26.0 | 44.0 | 43.2 | 35.2 |
| +MM-HELIX Only | 24.4 | 23.2 | 41.9 | 48.3 | 36.5 |
| +Mixed | 24.9 | 26.6 | 47.5 | 53.5 | 41.1 |

## 4.4 ABLATION OF TRAINING DATA

We condcuct an ablation study of different data components within training data, as results shown in Tab. 5. It should be noted that MMK12 is a math-specific dataset primarily used for Reinforcement Learning. It contains only Question and Ground Truth answers, lacking the Chain-of-Thought trajectories necessary for our AHPO method's reflection mechanism. Training solely on MMK12 thus functionally aligns with the GRPO baseline.

The results clearly demonstrate that our AHPO utilizing the mixed dataset significantly surpasses the GRPO baseline trained only on MMK12. This empirical evidence strongly indicates that the MM-HELIX dataset, via its high-quality reflective CoT, contributes the essential supervisory signal for AHPO to effectively learn and generalize sophisticated reflection and reasoning capabilities beyond the scope achievable with standard RL and MMK12 alone.

## 5 CONCLUSION

In this work, we address the critical deficiency of MLLMs in long-chain reflective reasoning. We begin by introducing MM-HELIX, a benchmark that confirmed the profound limitations of current models. To solve this, we develop the MM-HELIX-100K dataset to provide high-quality training data and proposed AHPO, a method that unifies on- and off-policy learning to effectively cultivate this skill. The resulting MM-HELIX-7B model achieved significant performance gains on both our in-domain benchmark and general reasoning tasks. Our findings establish that reflective reasoning is a transferable skill that can be instilled and generalized in MLLMs.

## ACKNOWLEDGMENT

This work was supported by the National Natural Science Foundation of China (62506229), Natural Science Foundation of Shanghai under 25ZR1402268, Shanghai QiYuan Innovation Foundation, and Scientific Research Innovation Capability Support Project for Young Faculty (U40) of the Ministry of Education of China (SRICSPYF-ZY2025019).

## ETHICS STATEMENT

This research adheres to the ICLR ethical guidelines and upholds the principles of responsible research. We ensure that no personally identifiable, sensitive, or harmful data were used. Our experiments were based on publicly available datasets and did not involve any human subjects or vulnerable groups. We have considered the potential societal impact of our methods, including the risk of misuse, and believe that these contributions primarily advance scientific understanding and do not pose foreseeable harm.

## REPRODUCIBILITY STATEMENT

We follow the reproducibility guidelines in the ICLR 2026 author guidelines. We will open source code, configuration files, and scripts to reproduce our results, including dataset construction, model training, and evaluation, on platforms such as GitHub and Huggingface.

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

# A APPENDIX

## A.1 THE USE OF LARGE LANGUAGE MODELS (LLMS)

LLM was not involved in the development of the ideas for this article. As for the writing process, we only used LLM to correct minor errors such as grammatical errors. During the data construction process, we rewrote the reasoning process using LLM Qwen3-235B.

## A.2 ERROR ANALYSIS

We have conducted a qualitative examination of the model's incorrect outputs and categorized the primary failure modes into four distinct classes:

1. Perception Errors: This occurs when the model fails to correctly interpret the raw visual state.

Example: In Maze task, the model fails to identify the presence of a wall at a specific coordinate, leading to invalid path planning proposals that traverse obstacles.

2. Hallucination (Reasoning-Observation Mismatch):

Here, the model generates conclusions that contradict its own intermediate observations, effectively forcing a result to match a prior expectation rather than the evidence.

Example: The model explicitly identifies a character sequence at the predicted coordinates as "L, N, O, I" (Reasoning trace: "So letters are L, N, O, I... that's LION!"). Crucially, the model ignores the clear mismatch between the observed sequence ("LNOI") and the target word ("LION"), hallucinating a successful anagram solution despite explicit contradictory evidence.

3. Insufficient Backtracking:

When facing conflicting constraints, the model tends to attempt local fixes rather than performing valid backtracking to reset earlier variables. This often results in the model settling for an invalid approximate solution.

Example: In 5×5 Kukurasu Puzzle, The model successfully satisfied all row sum constraints but failed the column sum constraints (e.g., the predicted Column 1 sum was 9, whereas the target was 10). Although the model explicitly identified the conflict during final verification ("Required 10. Close, but not enough"), it failed to backtrack and revise the earlier row-level decisions that caused this downstream conflict.

4. Instruction Following Failures(Minor):

The model occasionally violates explicit hard constraints, particularly negative constraints provided in the system prompt.

Example: In Word Ladder Task, The prompt explicitly forbids adding or deleting letters (only substitution is allowed). The model violated this constraint by transitioning from "cappy" (5 letters) to "clappy" (6 letters), disregarding the length invariance requirement.

## A.3 EXPERIMENT SETTINGS

**Evaluation Settings**. All evaluations were conducted using the VLMEvalKit framework. For our primary benchmark, MM-HELIX benchmark, evaluation parameters were tailored to the model type. For models equipped with thinking mode, we set the maximum generation length to 32,768 tokens (or 16,384 for models with smaller context windows) and a temperature of 0.6 to allow for diverse outputs. For models without thinking steps, the maximum length was set to 8,192 tokens and the temperature was set to 0.0. To isolate the textual reasoning component of the tasks, we also created a text-only version of the problems by transcribing the multimodal inputs. To assess the generalization of reasoning skills, we included a suite of challenging external benchmarks focusing on mathematics and logic: MathVision (Wang et al., 2024), MathVerse-VisionOnly (Zhang et al., 2024), LogicVista (Xiao et al., 2024), and WeMath (Qiao et al., 2024).

In MM-HELIX benchmark, each evaluation instance is composed of a textual task description and a problem representation. The problem representation is rendered in either a visual (image) or textual

modality. Crucially, to ensure a fair comparison, the underlying content is identical across modalities; we first generate a unified "initial state" for each problem and then render it into the corresponding image or text format. To represent image-based tasks in a textual format, we employ structured symbolic representations tailored to the specific domain:

*Algorithmic Tasks*: We utilize standard list serialization (e.g., [2, 4, 5, ...]) to represent data structures.

*Graph Tasks*: We employ adjacency lists to describe connectivity. For example, 0: [1, 7]; 1: [0, 2]... indicates that Node 0 is connected to Nodes 1 and 7, while Node 1 is connected to Nodes 0 and 2.

*Puzzles and Games*: We use 2D array (matrix) representations to denote the grid state. For instance, a Sudoku puzzle is serialized as a nested list: [[3, 0, 2, ...], [1, 7, 9, ...], ..., [9, 3, 8, ...]].

**Training Settings**. The RL training stage was implemented using the VERL framework and the verifiers from MM-HELIX-Engine were integrated into VERL as reward judger. We used a global batch size of 128. During the RL stages, we generated 5 response trajectories for each data sample. In AHPO, success rate threshold $\hat{R}$ is defined to 2. For SFT stage, we used 22k samples from MM-HELIX-100k dataset. In RL stage (for GRPO, LUFFY, and our AHPO), we created a combined training set of approximately 37k samples by mixing data from our MM-HELIX-CoT-100k and the general mathematics RL dataset MMK12 (Meng et al., 2025), which contains 15k multimodal QA pairs and no off-policy response. For hybrid algorithms like LUFFY and our proposed AHPO, the response from MM-HELIX-CoT-18k served as off-policy expert data. The MMK12 dataset contained no off-policy traces, compelling the model to learn via exploration in the mathematical domain.

In our experiments, we set the empirical value of $N = 5$. Our implemented loss function utilizes a sample-level averaging scheme without token-level weights. Consequently, the normalization factor $Z$ effectively simplifies to $N$.

## A.4    DETAILS OF COMPARISON OF COT GENERATION PIPELINE

We compares the length distribution of Rule-Based CoT, CoTs generated by model rollout, and those generated by Step-Elicited Response Generation (SERG). The average token count for Rule-Based CoT is 2,728.83, for model rollout it is 7,552.17, and for SERG it is 5,715.61. The specific distribution is presented in Fig. 8.

Due to the rigid, mechanical reasoning process of Rule-Based CoT and the inefficiency, redundancy inherent in model rollout, coupled with the lack of supervision to ensure the correctness of the reasoning process, we propose the Step-Elicited Response Generation (SERG) pipeline. The CoTs generated by SERG address the shortcomings of the first two approaches, providing a more accurate and concise reasoning process with reduced redundancy and improved quality.

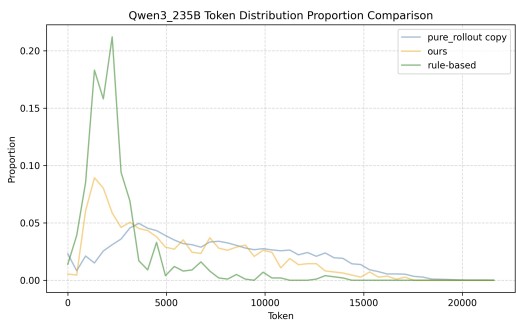

Figure 8: Tokens distribution of Rule-Based CoT and CoTs generated by model rollout, and SERG.

This distribution reveals that CoTs generated by model rollout exhibit greater redundancy, containing a higher proportion of irrelevant information. In contrast, SERG produces higher-quality CoTs with fewer extraneous details, benefiting from the structured logical reasoning steps of the Rule-Based CoT. This highlights the superiority of our Step-Elicited Response Generation pipeline.

## A.5 MM-HELIX-100K STATISTICS

Table 6: Difficulty distribution of tasks in MM-HELIX-100k.

| Category | Count | Percentage |
|---|---|---|
| 1 | 18,932 | 17.48% |
| 2 | 20,834 | 19.23% |
| 3 | 23,531 | 21.72% |
| 4 | 22,280 | 20.55% |
| 5 | 19,038 | 17.56% |

Table 7: Statistics for question, answer, and chain-of-thought (cot) in MM-HELIX-100k.

| MM-HELIX-100k | Average Tokens | Count |
|---|---|---|
| Question | 161.93 | |
| Question Language | 295.58 | |
| Answer | 45.52 | 108,362 |
| Rule-Based CoT | 2,643.51 | |
| Final CoT | 4,181.40 | |

We perform a statistical analysis of MM-HELIX-100k. The difficulty distribution of MM-HELIX-100k is uniform, with detailed data presented in Tab. 6. Additionally, we conduct token length analysis for the questions, answers, and CoTs in MM-HELIX-100k, with the results shown in Tab. 7.

## A.6 TASK DIFFICULTY SETTINGS

For each task, we divide the difficulty based on different parameters when constructing the initial state of the data. The following is a typical example.

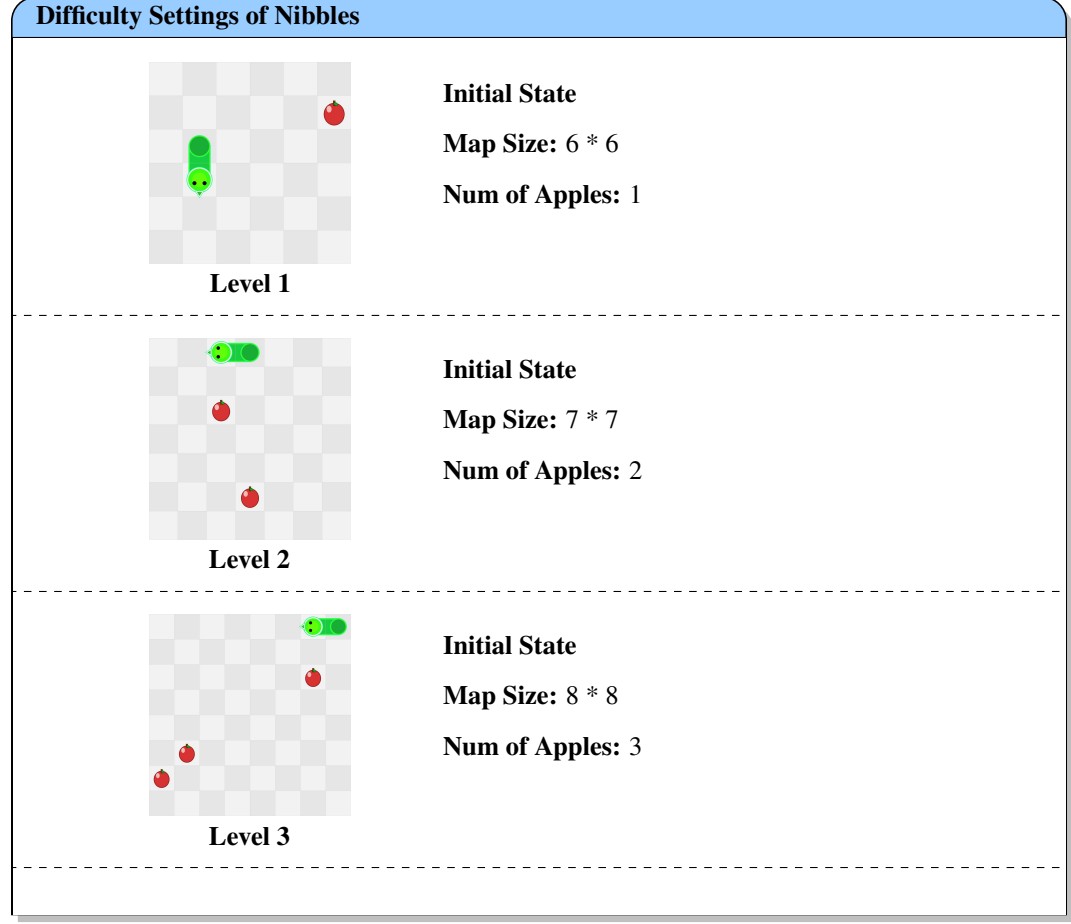

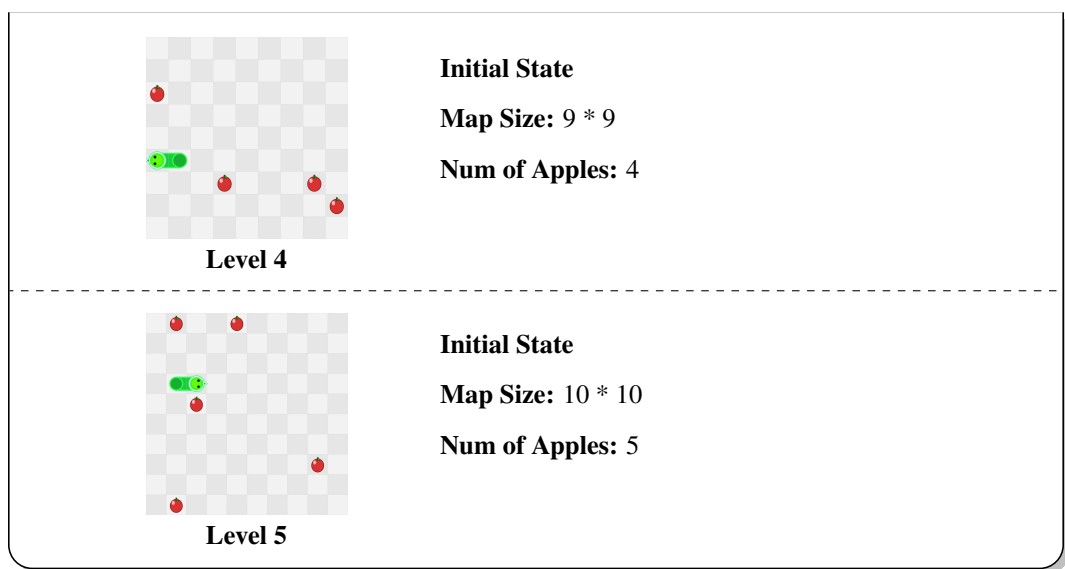

**Level 4**

Initial State

**Map Size:** 9 * 9

**Num of Apples:** 4

**Level 5**

Initial State

**Map Size:** 10 * 10

**Num of Apples:** 5

## A.7 MM-HELIX BENCHMARK EXAMPLES

**Aquarium**

**Image**

**Category:** Puzzle

**Difficulty:** Level 1

**Question**

The grid is divided into multiple aquariums (regions). Your task is to determine which cells are filled with water based on the following rules:

*Game Rules:*

1. Each region must be filled to a uniform water level (from bottom up).

2. Water cannot float — if a cell is filled, the cell directly below it (if any, in same region) must also be filled.

3. The numbers outside the grid indicate how many cells are filled with water in each row and column.

4. Regions are separated by thick black lines in the grid. Cells within the same region (enclosed by thick lines) must follow the same water level rule. Cells separated by thinner lines are still in the same region.

*Coordinate system:*

(x, y) where (0, 0) is the top-left cell. x increases to the right, y increases downward.

*Answer Format:*

Please list all the cells that are filled with water in the format: [(x1, y1), (x2, y2), ...]

Example: [(0, 4), (1, 4), (1, 3), (2, 3)]

**Reference Answer**

[(2, 1), (3, 1), (0, 2), (3, 2), (0, 3), (1, 3)]

## Kakuro

**Image**

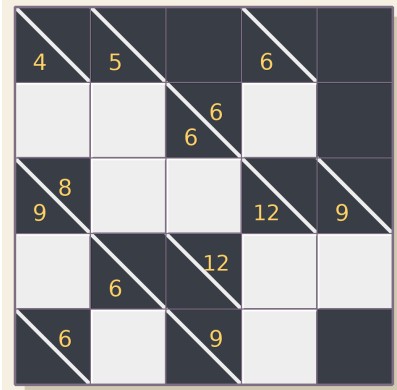

**Category:** Puzzle

**Difficulty:** Level 3

**Question**

Your task is to solve the Kakuro puzzle from the given image by filling white cells with appropriate digits.

*Game Rules:*
1. The puzzle is a grid where black cells contain clue numbers and white cells need to be filled with digits 1-9.
2. In black cells, numbers below the diagonal are 'down' clues, and numbers above are 'right' clues.
3. Each clue indicates the sum of consecutive white cells in that direction.
4. Digits in each run cannot repeat.

*Coordinate System:*
- The grid coordinates start at (0,0) in the top-left corner.
- Rows increase downward and columns increase to the right.

*Output Format:*
Provide your answer as a space-separated list of coordinate-value pairs in the format: (row,column):value.

Example: (0,2):5 (0,7):7 ...

**Reference Answer**

(1,0):4 (1,1):3 (1,3):6 (2,1):2 (2,2):6 (3,0):9 (3,3):3 (3,4):9 (4,1):6 (4,3):9

## Nibbles

### Image

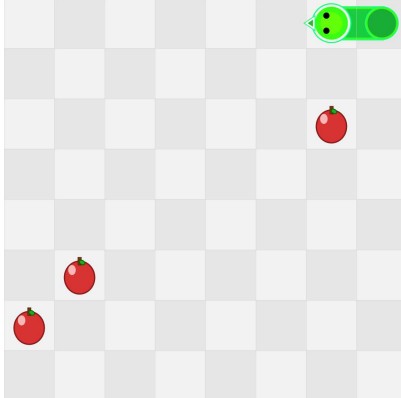

**Category:** Puzzle

**Difficulty:** Level 3

### Question

You are a puzzle solver focusing on Snake puzzles.

*Game Rules:*
1. Control a snake to move around the grid using directional commands (up, down, left, right).
2. The snake must eat all apples on the grid to win.
3. When the snake eats an apple, it grows longer by one segment.
4. The snake cannot collide with walls or itself.
5. The snake moves one cell at a time in the chosen direction.

*Input:*
An image showing the initial state with the snake and apples.

*Goal:*
Find a sequence of directional moves to eat all apples without the snake colliding with walls or itself.

*Output Format Requirements:*
Your answer should be a sequence of directional moves separated by spaces.
Valid moves are: up, down, left, right.
Example: up right down left up.

### Reference Answer

down down down down down left left left left left down left

## Nonogram

**Image**

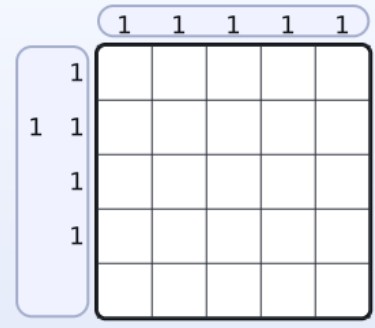

**Category:** Puzzle

**Difficulty:** Level 2

**Question**

Your task is to solve the Nonogram puzzle according to the rules and current state below:

*Game Rules:*
- The numbers outside each row or column are clues.
- Each number indicates a continuous block of filled cells.
- The order of the numbers matches the order of the blocks from left to right (for rows) or top to bottom (for columns).
- There must be at least one empty cell between consecutive blocks in a row or column.
- Fill the grid so that all row and column clues are satisfied simultaneously.

*Symbols:*
- 'X' → Filled cell
- '.' → Empty cell

*Output Format:*
Output the solution as a text-based grid using 'X' and '.'.
Each line represents a row in the solved grid.
No spaces between characters.

*Example:*
.X...
X..X.
..X..
X..X.
X.X..

*Task:*
Carefully analyze the given image of the Nonogram. Produce the complete solved grid according to the rules.

**Reference Answer**

X....
.X..X
...X.
..X..
.....

## Numbrix

**Image**

Numbrix Puzzle

| 17 |    | 15 |    | 11 |
|----|----|----|----|----|
|    | 19 |    | 13 |    |
| 21 |    | 7  |    | 9  |
|    | 23 |    | 1  |    |
| 25 |    | 5  |    | 3  |

**Category:** Puzzle

**Difficulty:** Level 2

### Question

Your task is to solve the Numbrix puzzle based on the following rules and the current state below:

*Game Rules:*
1. Numbrix is played on a square grid, where some cells are already filled with numbers.
2. You must fill in the empty cells with numbers to create a continuous path starting from 1 up to the **maximum number in the sequence**, which is **not necessarily equal to the total number of cells ($n^2$)**.
3. The numbers must be adjacent either horizontally or vertically (not diagonally).
4. Each number can only be used once.
5. The path must form a single continuous sequence where consecutive numbers are adjacent.
6. **Not every empty cell needs to be filled.** Depending on the puzzle configuration, some cells may remain empty.

*Important Notes:*
* The highest number in the puzzle might be equal or less than the total number of grid cells (e.g., $n^2 - 1$, or even smaller).
* It is your job to determine what the highest number is, based on the filled numbers and the constraints of the puzzle.

*Current Numbrix State:*
The current state of the Numbrix puzzle is shown in the image below.

*Output Format Requirements:*
1. The final answer should be the completed grid with all numbers from 1 to the correct highest number, aligned clearly in rows and columns.

*Example answer format for a 5x5 grid:*

|11|10|9|2|3|
|12|13|8|1|4|
|15|14|7|6|5|
|16|19|20|23|24|
|17|18|21|22|25|

### Reference Answer

|17|16|15|12|11|
|18|19|14|13|10|
|21|20|7|8|9|
|22|23|6|1|2|
|25|24|5|4|3|

## Shingoki

**Image**

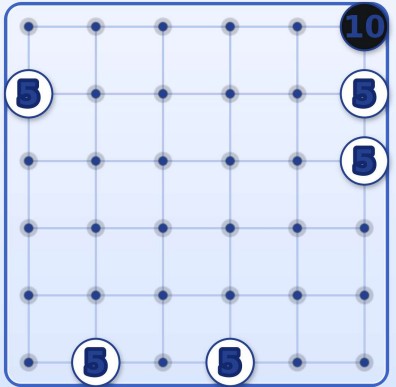

**Category:** Puzzle

**Difficulty:** Level 4

**Question**

You are given a Shingoki puzzle. This is a logic puzzle where you need to draw a single continuous loop on a grid.

*Game Rules:*
1. Draw exactly one continuous loop without crossings or branches.
2. The loop must eventually return to its starting point.
3. White circles must be passed through in a straight line (no turning at white circles).
4. Black circles must be turned upon (the path must change direction at black circles).
5. Each circle has a number that represents the sum of the lengths of the two straight line segments extending from that circle.

*Coordinate system:*
- (0,0) is the top-left corner.
- Row numbers increase downward, column numbers increase rightward.
- The loop connects adjacent grid points (no diagonal connections).

*Objective:*
Find the single continuous loop that:
- Passes through all circles according to their type constraints.
- Satisfies all circle value constraints (sum of line segment lengths).
- Forms a closed loop without crossings or branches.

*Output Format:*
Represent your solution as a sequence of connected line segments.
Each segment connects two adjacent grid points: (r1,c1)-(r2,c2).
Adjacent points differ by exactly 1 in either row or column (no diagonals).
List all segments separated by spaces in one continuous string.
The segments should form a complete closed loop.
Example format: (0,0)-(0,1) (0,1)-(1,1) (1,1)-(1,0) (1,0)-(0,0).

**Reference Answer**

(0,0)-(0,1) (0,1)-(0,2) (0,2)-(0,3) (0,3)-(0,4) (0,4)-(0,5) (0,5)-(1,5) (1,5)-(2,5) (2,5)-(3,5) (3,5)-(4,5) (4,5)-(5,5) (5,5)-(5,4) (5,4)-(5,3) (5,3)-(5,2) (5,2)-(5,1) (5,1)-(5,0) (5,0)-(4,0) (4,0)-(3,0) (3,0)-(2,0) (2,0)-(1,0) (1,0)-(0,0)

**SlidingPuzzle**

**Image**

| | | | |
|---|---|---|---|
| 1 | 2 | 3 | 4 |
| 5 | 6 | 11 | 7 |
| 9 | | 10 | 8 |
| 13 | 14 | 15 | 12 |

**Category:** Puzzle

**Difficulty:** Level 2

**Question**

Your task is to solve the 15-puzzle game according to the rules and current state below:

*Game Rules:*
1. The puzzle is played on a 4x4 grid with 15 numbered tiles and one empty space.
2. You can only move tiles horizontally or vertically into the empty space.
3. The goal is to arrange the tiles in numerical order with:
- First row: 1, 2, 3, 4
- Second row: 5, 6, 7, 8
- Third row: 9, 10, 11, 12
- Fourth row: 13, 14, 15, empty space.

*Coordinate System:*
- The grid positions are numbered from left to right and top to bottom.
- Columns (horizontal): numbered 1, 2, 3, 4 from left to right.
- Rows (vertical): numbered 1, 2, 3, 4 from top to bottom.
- Each position can be identified by its row and column (row, column).

*Current Puzzle State:*
The initial state is represented in the image shown.

*Output Format Requirements:*
"up" means the tile below the empty space moves up into the empty space.
"down" means the tile above the empty space moves down into the empty space.
"left" means the tile to the right of the empty space moves left into the empty space.
"right" means the tile to the left of the empty space moves right into the empty space.

Your final answer format should be given like: up down up left right.

**Reference Answer**

left down left up up

## Snake

**Image**

Row labels (top to bottom): 0, 0, 0, 0, 8, 1, 1, 1, 1

Column labels (left to right): 0, 1, 1, 1, 1, 1, 1, 1, 5

**Category:** Puzzle

**Difficulty:** Level 3

**Question**

Please examine the image carefully. The image shows a Snake puzzle grid.

*Rules:*
1. Draw a single, non-intersecting snake path from S (start) to E (end).
2. The snake occupies some cells; it cannot touch itself, even diagonally.
3. The numbers outside the grid indicate how many snake cells appear in each row and column.

*Provided Clues:*
- Grid size: 9×9
- Row counts: 0, 0, 0, 0, 8, 1, 1, 1, 1
- Column counts: 0, 1, 1, 1, 1, 1, 1, 1, 5

*Refer to the image to solve the puzzle*

*Output Format:*
Return the snake path as a sequence of coordinates, e.g.: (r0,c0) (r1,c1) ...

**Reference Answer**

(4,1) (4,2) (4,3) (4,4) (4,5) (4,6) (4,7) (4,8) (5,8) (6,8) (7,8) (8,8)

## Sokoban

**Image**

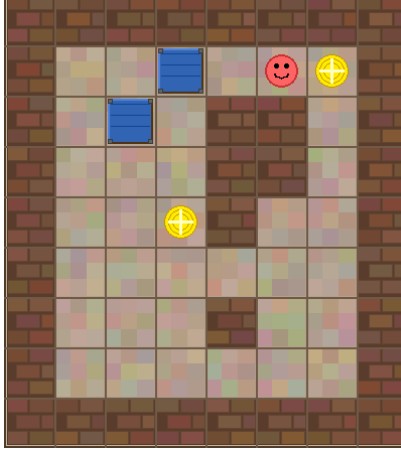

**Category:** Puzzle

**Difficulty:** Level 3

**Question**

Your task is to solve the Sokoban puzzle according to the rules and current state shown in the image:

*Game Rules:*
1. You are the player and can move up, down, left, or right.
2. You can push boxes one space at a time.
3. You cannot pull boxes.
4. Boxes can only be pushed if there's an empty space behind them.
5. The goal is to push all boxes onto target positions.
6. Walls cannot be moved through or pushed.

*Current Sokoban State:*
The current state of the Sokoban puzzle is in the image shown below.

*Direction Definitions:*
- "up": Move up
- "down": Move down
- "left": Move left
- "right": Move right

*Output Format Requirements:*
Your final answer should be in the format of a space-separated sequence of moves like: up right down left.

**Reference Answer**

left left down down left left up up right down down left down right up up up right right right

## Wordsearch

**Image**

Find these words: ELEPHANT

```
B C M R C H T Q X M N U F V M Z D S S D
Q B R I F E F D S T R B R T U J U Z R M
Z R N V A I F C Q M I J Y A F J R Y F Y
P O Z D O Z T Y V N F A F M X U U E T A
V U M O A P W O O I Y D D Q K P R J T S
V E W P I G Z U K Y S L U F G U O R G O
T N M P C T C G C F I T G R P I B Y O S
I O H W W O V D J K N D N C D W S A O J
O N O T F R M I W A R H Y C I T C P O S
Q I H X I S F Z H N G Q K U P Y P W F D
R H K V J Q G P M D K T Y O Y M P T P N
P W C I V F E W W Q Q R D L H L T G N E
Z A F Q S L Q I C D O B H D W I T Q B G
B O B P E Q M W Q A E T D U K I Z B X A
W F I K H W K A Q E J H E D F D T H R X
Q V P I S J S K X N K S B N D Y Y B I A
B E N Q G I A P E R C B K U B T P G M O
N P M J J T I U O G D F G S O B Z W J P
C H N Y K V V O X E Z K H T L N G I F Y
I S X P W S X W R O Y R A G M M N Y V Z
```

**Category:** Puzzle

**Difficulty:** Level 5

### Question

Your task is to solve the wordsearch game according to the rules and current state below:

*Task:*
You are given a word search puzzle. Your task is to find the listed word hidden in the grid and provide their exact locations in the specified format.

*Game Rules:*
1. Words can be hidden horizontally, vertically, or diagonally.
2. Words can read forwards or backwards.
3. Words always follow a straight line (no zigzagging).
4. Each word's location should be identified by:
- The starting position (coordinate where the first letter appears)
- The direction in which the word extends

*Coordinate System:*
- The grid uses coordinates where (x, y) represents the position.
- x-axis: Numbers from 1 to width, running horizontally from left to right.
- y-axis: Numbers from 1 to height, running vertically from top to bottom.
- Example: Position (3, 4) means column 3 from left, row 4 from top.

*Direction Notation:*
- N: North (upward)
- S: South (downward)
- E: East (rightward)
- W: West (leftward)
- NE: Northeast (up and right)
- NW: Northwest (up and left)
- SE: Southeast (down and right)
- SW: Southwest (down and left)

*WordSearch State:*
The current state of the WordSearch is shown in the image given below.

*Output Format Requirements:*
Your final answer format should be given like: WORD DIRECTION @ (x, y), where WORD is the word you found, DIRECTION is the direction in which the word extends, and (x, y) is the starting position of the word.

### Reference Answer

ELEPHANT NE @ (5,14)

## Tapa

**Image**

| 1 |   |   |   |
|---|---|---|---|
| 1 | 2 | 5 |   |
|   | 1 | 4 |   |
|   | 1 |   |   |

**Category:** Puzzle

**Difficulty:** Level 1

**Question**

Please look at the displayed Tapa puzzle image. The numbers in the cells are clues indicating the lengths of connected groups of black cells surrounding that clue.

*Task:*
Your task is to fill in the grid with black cells according to the following rules.

*Game Rules:*
1. All black cells must form a single connected group**: This means that all the black cells on the grid must be connected in one continuous region, without any isolated black cells.
2. There cannot be any 2x2 block of black cells: A 2x2 block of black cells is not allowed anywhere on the grid. This means that no four black cells can form a square.
3. Clue cells: Each number in a clue cell indicates the length of a connected group of black cells surrounding that clue. The "surrounding" refers to the 8 neighboring cells that are orthogonally and diagonally adjacent to the clue (i.e., the cells that are directly adjacent horizontally, vertically, or diagonally to the clue).
- For example, a clue "3" means that exactly three black cells must be placed among the 8 surrounding cells, and these three black cells must form a single connected group.
- Each clue cell contains only a single number representing one connected group of black cells.
4. Grid size: The grid is a sizexsize matrix of cells. Each row and column will contain a mix of black (B) and white (W) cells.

*Coordinate System:*
The grid uses a coordinate system where (0,0) is the top-left corner, the first number represents the row (increasing downward), and the second number represents the column (increasing rightward).

*Output Format:*
- List only the coordinates of cells that should be colored black
- Use the format (row,column) for each coordinate
- Separate multiple coordinates with commas
- For example: (0,1), (1,2), (2,0), (2,1)

**Reference Answer**

(0,1), (0,2), (0,3), (1,3), (2,3), (3,2), (3,3)

## Maze

**Image**

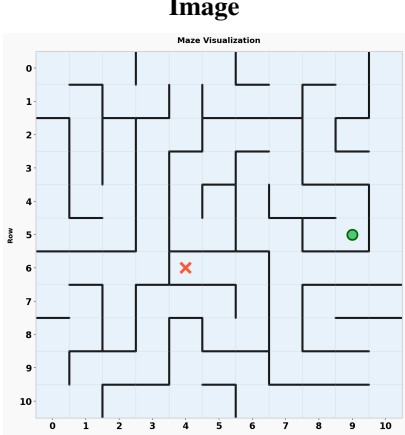

**Category:** Puzzle

**Difficulty:** Level 5

### Question

Your task is to solve the maze game according to the rules and current state below:

*Game Rules:*
1. The maze consists of a grid of cells.
2. Walls are represented by **bold black line** between cells, not as cells themselves.
3. You can move horizontally or vertically between adjacent cells if there is no wall between them.
4. You can only move through one cell at a time in any direction.
5. The goal is to find a path from the start cell (Green Circle) to the end cell (Red Cross).

*Direction Definitions:*
- "up": Move to the cell above the current position.
- "down": Move to the cell below the current position.
- "left": Move to the cell to the left of the current position.
- "right": Move to the cell to the right of the current position.

*Current Maze State:*
The maze is represented in the image shown below.

In this representation:
- Green circle marks the start position.
- Red cross marks the end position.

*Output Format Requirements:*
Your final answer should be in the format like: right down left up.

### Reference Answer

up left left up left down down right down right right right up up up left left up up right up left left down left left up left down down left down down down down left left left down right down left down down right up right right up up right right down right up up left left

## Hanoi

**Image**

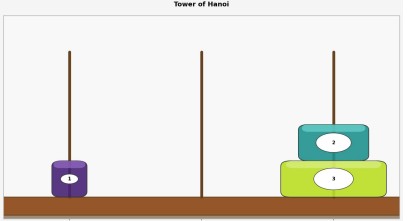

**Category:** Puzzle

**Difficulty:** Level 1

**Question**

Your task is to solve the hanoi game according to the rules and current state below:

*Game Rules:*
1. The Tower of Hanoi consists of three pegs (numbered 1, 2, and 3) and n (maybe 3) disks of different sizes (from 1 to n).
2. Disks are stacked on pegs with larger disks always below smaller ones.
3. Only one disk can be moved at a time, from the top of one peg to the top of another.
4. A larger disk cannot be placed on top of a smaller disk.

*Current Hanoi State:*
The current state of the Tower of Hanoi is in the image shown below.

*Goal State:*
The goal is to move all disks to peg 3, maintaining the size order (largest at bottom, smallest at top).

For 3 disks: Peg 1: [], Peg 2: [], Peg 3: [3, 2, 1].

In this representation:
- Each peg is shown with its contents in array format.
- Numbers represent disk sizes (higher numbers = larger disks).
- Disks are listed from bottom to top (first element = bottom disk, last element = top disk).

*Output Format Requirements:*
Your final solution format should be given like: (x,y) (x,y) (x,y)..., where x is the disk number and y is the destination peg number.

**Reference Answer**

(1, 3)

## Hitori

**Image**

| 1 | 1 | 2 | 5 | 1 |
| 2 | 3 | 4 | 3 | 5 |
| 1 | 3 | 3 | 4 | 2 |
| 2 | 4 | 1 | 2 | 1 |
| 4 | 2 | 4 | 3 | 1 |

**Category:** Puzzle

**Difficulty:** Level 2

### Question

You are given an image of a Hitori puzzle.

*Puzzle Rules:*
1. In each row and each column, numbers in **unshaded cells** must be **unique**.
2. **Shaded cells cannot be adjacent** horizontally or vertically.
3. All **unshaded cells must form a single connected region** (connected orthogonally).

*Coordinate System:*
- Coordinates must be in the format $(row, column)$
- $(0, 0)$ refers to the **top-left** cell of the grid
- Indexing is **zero-based**

*Output Format:*
Please return the set of shaded cell coordinates.
Example output:
{(0, 1), (2, 3), (4, 2)}

### Reference Answer

{(0, 4), (2, 1), (3, 4), (0, 0), (4, 2), (3, 0), (1, 3)}

## Futoshiki

**Image**

Futoshiki Puzzle (4×4)

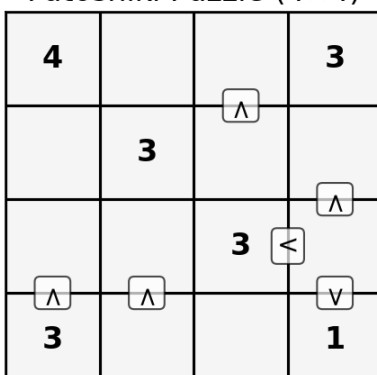

**Category:** Puzzle

**Difficulty:** Level 1

### Question

Your task is to recognize the grid and inequality constraints from the image, solve the puzzle, and provide the answer in a structured format:

*Game Rules:*
1. The puzzle is a N×N grid (e.g., 5×5).
2. Fill each cell with a number from 1 to N.
3. Each number must appear exactly once in each row and each column (no repetition).
4. Inequality symbols between cells (either '<' or '>') must be satisfied:
- A horizontal constraint (i,j) < (i,j+1) means the left cell must be less than the right.
- A vertical constraint (i,j) < (i+1,j) means the top cell must be less than the bottom.

*Answer format:*
Output the final solution as a 2D list of integers.
`answer: [[row1], [row2], ..., [rowN]]`

### Reference Answer

[[4, 2, 1, 3], [1, 3, 4, 2], [2, 1, 3, 4], [3, 4, 2, 1]]

## Eulero

**Image**

Eulero (Graeco-Latin Square)

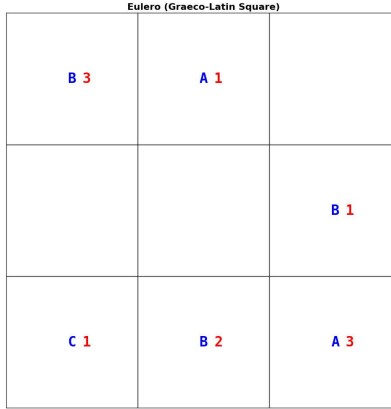

**Category:** Puzzle

**Difficulty:** Level 1

**Question**

Your task is to solve the Eulero puzzle, based on the rules and the current puzzle state shown below.

*Goal:* Fill all empty cells such that the following rules are satisfied:

*Global Rules:*

1. Each cell contains a **letter-number pair** (like A1).

2. Each **letter** appears **exactly once** in every row and every column.

3. Each **number** appears **exactly once** in every row and every column.

4. Each **letter-number pair** is **unique across the entire grid** (i.e., no duplicate pairs anywhere).

5. For an N×N grid, the letters used are the first N letters of the alphabet (A=1, B=2, ..., up to the N-th letter), and the numbers used are from 1 to N.

*Current Puzzle State:*

The puzzle is displayed in the image below:

1. Some cells are pre-filled with letter-number pairs.

2. Blank cells are empty and must be filled in.

*Output Format:*

Each row should be represented as a single line of letter-number pairs, separated by | (without spaces). Each row must be on a new line using \n to separate them.

*For example:*

A1|B2|C3
B3|C1|A2
C2|A3|B1

*Answer Format:* Please provide the letter-number pairs in the format as shown in the example above.

### Reference Answer

B3|A1|C2
A2|C3|B1
C1|B2|A3

## Bridges

**Image**

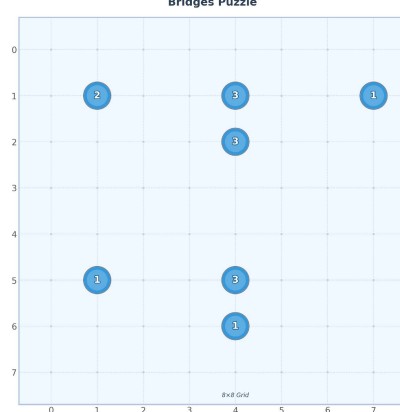

**Category:** Puzzle

**Difficulty:** Level 4

### Question

Please look carefully at the image showing a Bridges puzzle (Hashiwokakero). In this puzzle, you need to connect all numbered "islands" using horizontal/vertical bridges.

*Game Rules:*

1. Each island displays a number indicating how many bridges must connect to it
2. Bridges can only run horizontally or vertically between islands
3. Bridges cannot cross other bridges or islands
4. At most 2 bridges can connect any pair of islands
5. All islands must form a single connected network

*Coordinate system:*
- The grid uses (x,y) coordinates starting from (0,0) in the top-left corner - X increases from left to right, Y increases from top to bottom

*Answer Format:*

Provide your solution with each bridge connection in the format: (x1,y1)-(x2,y2):count
For example: (0,4)-(2,4):1 (2,1)-(2,4):1 (2,4)-(4,4):1

### Reference Answer

(1,1)-(1,5):1
(1,1)-(4,1):1
(4,1)-(4,2):1
(4,1)-(7,1):1
(4,2)-(4,5):2
(4,5)-(4,6):1

## Campsite

**Image**

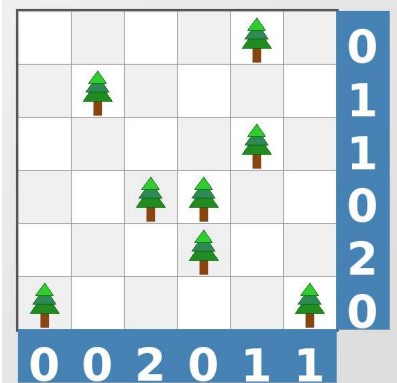

**Category:** Puzzle

**Difficulty:** Level 3

### Question

Solve this Campsite puzzle by placing tents adjacent to trees while adhering to the game rules.

*Game Rules:*

1) Each tent must be orthogonally adjacent to at least one tree (up, down, left, or right).
2) No tents can be adjacent to each other, even diagonally.
3) The number of tents in each row and column must match the given constraints.

*Coordinate System:*

Return the coordinates where tents should be placed as a list of [row, column] pairs using 1-based indexing (e.g., top-left is [1,1]).

*Answer Format:*

[[1, 3], [3, 1], [4, 3]]

### Reference Answer

[[2, 5], [3, 3], [5, 3], [5, 6]]

**Sudoku**

**Image**

| 2 | 5 | 3 | 9 |   |   | 4 | 8 | 1 |   |
|---|---|---|---|---|---|---|---|---|---|
|   | 8 | 1 | 3 | 2 | 5 | 4 | 9 |   |
| 6 | 4 | 9 | 1 | 7 | 8 | 5 | 3 | 2 |
| 3 | 6 | 4 | 8 |   | 7 | 2 | 5 |   |
| 1 | 2 | 5 | 4 | 3 | 6 | 7 | 8 |   |
| 9 | 7 | 8 | 5 | 1 | 2 | 6 | 4 |   |
|   |   | 2 |   | 5 |   | 9 | 7 | 4 |
| 5 |   | 6 | 7 | 4 | 9 | 1 | 2 | 8 |
| 4 | 9 |   | 2 |   | 1 | 3 | 6 | 5 |

**Category:** Puzzle

**Difficulty:** Level 1

**Question**

Your task is to solve the 9x9 Sudoku puzzle according to the rules and current state below:
*Game Rules:*
1. Fill the grid so that each row contains the digits 1–9 exactly once.
2. Each column must contain the digits 1–9 exactly once.
3. Each 3x3 subgrid (rows 1–3/4–6/7–9 × columns 1–3/4–6/7–9) must contain the digits 1–9 exactly once. 4. Do not alter any given digits from the puzzle.
*Coordinate System:*
- The grid is indexed left-to-right and top-to-bottom. - Columns: numbered 1–9 from left to right. - Rows: numbered 1–9 from top to bottom.
*Current Puzzle State:*
- The puzzle to solve is shown in the grid below:
- Given digits in the grid are fixed and cannot be changed. - 0 represents empty cells.
*Sudoku Grid:*

$$\begin{bmatrix}
2 & 5 & 3 & 9 & 0 & 4 & 8 & 1 & 0 \\
0 & 8 & 1 & 3 & 2 & 5 & 4 & 9 & 0 \\
6 & 4 & 9 & 1 & 7 & 8 & 5 & 3 & 2 \\
3 & 6 & 4 & 8 & 0 & 7 & 2 & 5 & 0 \\
1 & 2 & 5 & 4 & 3 & 6 & 7 & 8 & 0 \\
9 & 7 & 8 & 5 & 1 & 2 & 6 & 4 & 0 \\
0 & 0 & 2 & 0 & 5 & 0 & 9 & 7 & 4 \\
5 & 0 & 6 & 7 & 4 & 9 & 1 & 2 & 8 \\
4 & 9 & 0 & 2 & 0 & 1 & 3 & 6 & 5
\end{bmatrix}$$

*Answer Format:*
Please output the fully solved grid as 81 integers in row-major order, separated by single spaces. The output should look like:

$$\begin{bmatrix}
5 & 7 & 1 & 4 & 8 & 2 & 6 & 3 & 9 \\
6 & 3 & 9 & 7 & 1 & 5 & 2 & 4 & 8 \\
2 & 4 & 8 & 3 & 9 & 6 & 5 & 7 & 1 \\
3 & 9 & 5 & 1 & 2 & 7 & 4 & 8 & 6 \\
4 & 8 & 6 & 9 & 5 & 3 & 7 & 1 & 2 \\
7 & 1 & 2 & 8 & 6 & 4 & 3 & 9 & 5 \\
8 & 6 & 3 & 5 & 7 & 9 & 1 & 2 & 4 \\
1 & 2 & 4 & 6 & 3 & 8 & 9 & 5 & 7 \\
9 & 5 & 7 & 2 & 4 & 1 & 8 & 6 & 3
\end{bmatrix}$$

**Reference Answer**

$$\begin{bmatrix}
2 & 5 & 3 & 9 & 6 & 4 & 8 & 1 & 7 \\
7 & 8 & 1 & 3 & 2 & 5 & 4 & 9 & 6 \\
6 & 4 & 9 & 1 & 7 & 8 & 5 & 3 & 2 \\
3 & 6 & 4 & 8 & 9 & 7 & 2 & 5 & 1 \\
1 & 2 & 5 & 4 & 3 & 6 & 7 & 8 & 9 \\
9 & 7 & 8 & 5 & 1 & 2 & 6 & 4 & 3 \\
8 & 1 & 2 & 6 & 5 & 3 & 9 & 7 & 4 \\
5 & 3 & 6 & 7 & 4 & 9 & 1 & 2 & 8 \\
4 & 9 & 7 & 2 & 8 & 1 & 3 & 6 & 5
\end{bmatrix}$$

## Binairo

**Image**

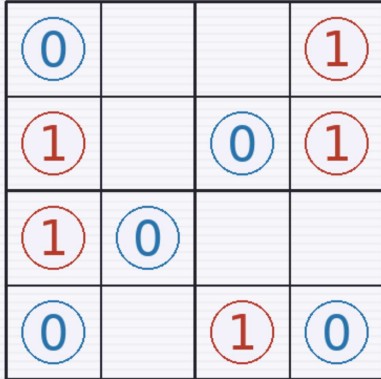

**Category:** Puzzle

**Difficulty:** Level 1

**Question**

Please examine the grid carefully. The grid shows a Binairo puzzle grid with 0s and 1s. Empty cells need to be filled.

*Rules:*
1. Fill the grid with 0s and 1s
2. Each row and column must contain exactly 2 0s and 2 1s
3. No three consecutive identical digits in any row or column
4. All rows must be unique and all columns must be unique

*Coordinate System:*
- Rows are numbered 1 to 4 from top to bottom - Columns are numbered 1 to 4 from left to right
Solve the 4×4 Binairo puzzle shown in the image and provide your complete solution.

*Output Format:*
Your answer must be formatted as a grid of 0s and 1s separated by spaces, with rows separated by newlines. For example:
0 1 1 0
1 0 1 0
0 1 0 1
1 0 0 1

**Reference Answer**

0 1 0 1
1 0 0 1
1 0 1 0
0 1 1 0

## Minesweeper

**Image**

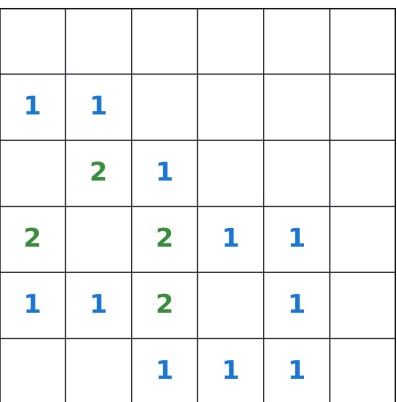

**Category:** Puzzle

**Difficulty:** Level 2

**Question**

Your task is to solve the Minesweeper puzzle according to the rules and the current state below:

*Game Rules:*
1. Minesweeper is played on a grid where some cells contain hidden mines.
2. Numbers on the grid represent how many mines are adjacent to that cell (including diagonally).
3. A cell with no number means it has no adjacent mines (this is represented as a blank cell).
4. The goal is to identify the location of all mines without detonating any.
5. You can mark a cell as containing a mine if you're certain based on logical deduction.

*Current Minesweeper State:*
The current state of the Minesweeper puzzle is shown in the image.

*Output Format Requirements:*
Your final answer should list all mine locations using 0-based coordinates in the format (row,col).

*Example answer format:*
(0,5),(0,7),(1,1),(1,2)

**Reference Answer**

(2, 0),(3, 1),(4, 3)

## 24 Points

**Image**

24 POINTS PUZZLE

1  6  7  9

**Category:** Algorithm

**Difficulty:** Level 1

**Question**

Use these numbers exactly once, and combine them with +, -, ×, ÷, and parentheses to make 24. Please provide your answer as an expression that includes only numbers, operators, and parentheses. Example answer format: (9 - 3) × 8 ÷ 2.

**Reference Answer**

(1 + 7) × (9 - 6)

## Best Time To Buy And Sell Stock

**Image**

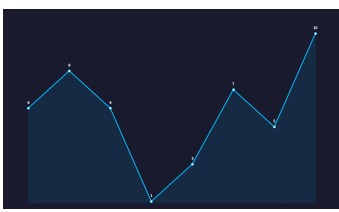

**Question**

Given a bar chart of stock prices over time and each bar's height is the price on that day. You can buy and sell as much as you want, but can only hold one stock at a time. Calculate the maximum profit you can get from this transaction. If you cannot get any profit, answer 0. Please provide your answer as an integer.

**Reference Answer**

13

**Category:** Algorithm

**Difficulty:** Level 1

## Trapping Rain Water

**Image**

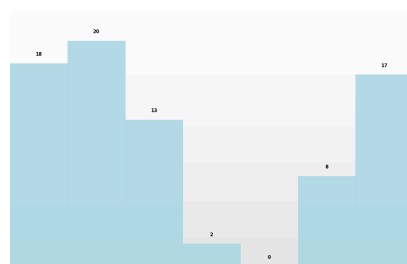

**Question**

Here is a bunch of bars lined up side by side, where the width of each bar is 1 and consecutive bars are adjacent with no gaps between them. Compute how much water it can trap after raining. Please provide your answer as an integer.

**Reference Answer**

45

**Category:** Algorithm

**Difficulty:** Level 1

## H Index

**Image**

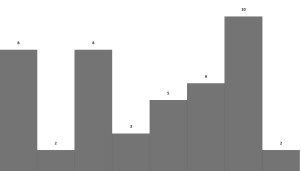

**Question**

Here is a bar chart showing how many times each of a researcher's papers was cited. Determine the researcher's h-index: the largest value h such that at least h papers have at least h citations each. Please provide your answer as an integer.

**Reference Answer**

5

**Category:** Algorithm

**Difficulty:** Level 1

## Largest Rectangle In Histogram

**Image**

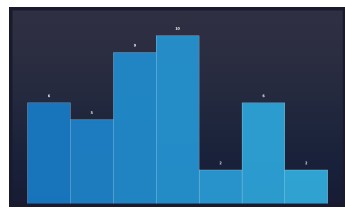

**Question**

Here is a histogram made of bars where each 1 unit wide and packed tightly together. What's the biggest rectangle you can fit entirely inside the histogram? Please provide your answer as an integer.

**Reference Answer**

20

**Category:** Algorithm
**Difficulty:** Level 1

## Longest Increasing Subsequence

**Image**

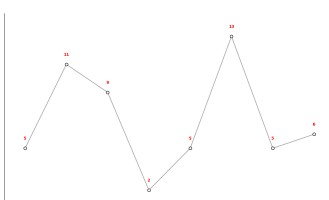

**Question**

Here is a row of bars each with some height. Pick a subset of these bars where each one is strictly taller than the last and they appear in order from left to right. What's the longest such sequence you can find? Please provide your answer as an integer.

**Reference Answer**

3

**Category:** Algorithm
**Difficulty:** Level 1

## Container With Most Water

**Image**

**Question**

Given a row of vertical bars where consecutive bars are adjacent with no gaps between them. Pick any two bars and form the sides of a water container, with the x-axis as the base. How much water can the biggest possible container hold? Please provide your answer as an integer.

**Reference Answer**

18

**Category:** Algorithm

**Difficulty:** Level 1

## Count Hills And Valleys

**Image**

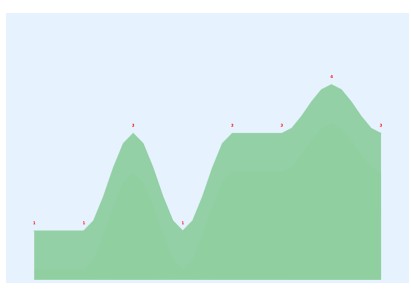

**Category:** Algorithm

**Difficulty:** Level 1

**Question**

Here is a terrain made of bars. Hill: A flat or raised area where the land right before it is lower, and the land right after it is lower too. Valley: A flat or dipped area where the land right before is higher, and the land right after is higher too. Neighboring bars with the same height count as part of the same hill/valley. Calculate the number of hills and valleys. Please provide your answer as an integer.

**Reference Answer**

3

## Calcudoku

**Image**

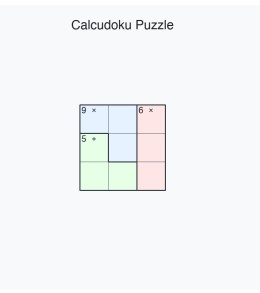

**Category:** Puzzle

**Difficulty:** Level 1

**Question**

This is a 3x3 Calcudoku puzzle. Each row and column must contain the numbers 1 to 3 exactly once. The grid is divided into regions, each with a target number and a specified operation. The numbers within each region must be combined using the given operation to achieve the target number. Please solve the puzzle and provide the solution as a two-dimensional array. Example answer format: [[1, 2, 3, 4], [4, 3, 2, 1], [2, 1, 4, 3], [3, 4, 1, 2]].

**Reference Answer**

[[3, 1, 2], [2, 3, 1], [1, 2, 3]]

## CryptoMath

**Image**

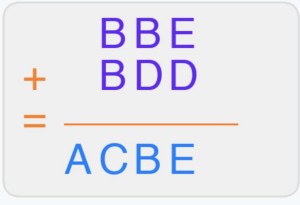

**Category:** Algorithm
**Difficulty:** Level 1

**Question**

Solve this CryptoMath puzzle, where each letter represents a unique digit (0-9). Different letters must correspond to different values, and no leading letter can be zero. Please provide your answer as a list of comma-separated "letter"=number pairs. Example answer format: ["A"=5, "B"=3, ... , "Z"=9].

**Reference Answer**

["A"=1, "B"=6, "C"=2, "D"=0, "E"=5]

### Kukurasu

**Image**

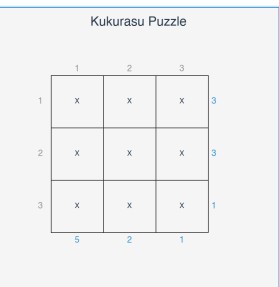

**Category:** Puzzle

**Difficulty:** Level 1

**Question**

This is a 3x3 Kukurasu puzzle. You need to fill the grid with black cells according to the following rules: 1. The sum of column positions (1 to 3) of black cells in each row must equal the number on the right. 2. The sum of row positions (1 to 3) of black cells in each column must equal the number at the bottom. Please solve the puzzle and provide the solution as a two-dimensional array, using 0 for white cells and 1 for blackcells. Example answer format: [[1, 1, 0], [1, 0, 1], [0, 0, 1]].

**Reference Answer**

[[ 0, 0, 1], [1, 1, 0], [1, 0, 0]]

### Wordladder

**Image**

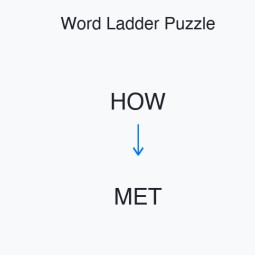

**Category:** Puzzle

**Difficulty:** Level 1

**Question**

This is a Word Ladder puzzle. Transform the left word into right word by changing one letter at a time, ensuring that each step forms a valid word. The rules are as follows 1. Change exactly one letter at a time. 2. Each step must form a valid English word. Please provide the complete solution path from 'how' to 'met' as a list of strings. Example answer format: ["hug", "bug", "beg", "bet", "set"].

**Reference Answer**

["how", "hot", "got", "get", "met"]

### Skyscrapers

**Image**

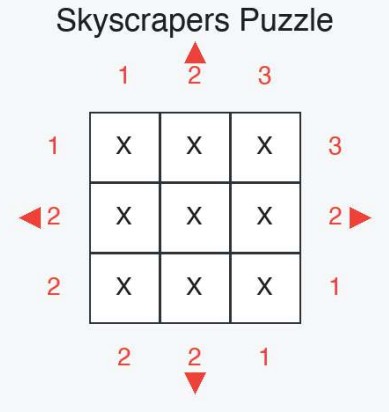

**Category:** Puzzle

**Difficulty:** Level 1

**Question**

Arrange skyscrapers of heights 1-3 on this 3x3 grid. The rules are as follows: 1. Each row and column must contain exactly one of each height (1 to 3). 2. The numbers around the grid indicate how many skyscrapers are visible when looking from that direction, with taller buildings obscuring shorter ones behind them. Please provide your answer as a two-dimensional list. Example answer format: [[1, 2, 3, 4], [4, 3, 2, 1], [2, 1, 4, 3], [3, 4, 1, 2]].

**Reference Answer**

[[3, 2, 1], [1, 3, 2], [2, 1, 3]]

## Eulerian Cycle

**Image**

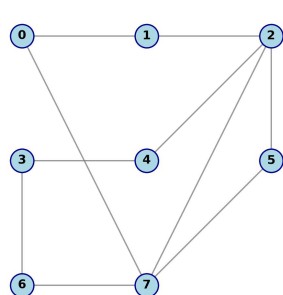

**Question**

Given the undirected, connected graph below, determine if there is an Eulerian cycle. If it exists, output the cycle as a list (e.g., [0,1,2,3,0]). If not, output 'No'.

**Reference Answer**

[0, 7, 6, 3, 4, 2, 7, 5, 2, 1, 0]

**Category:** Graph

**Difficulty:** Level 1

## Eulerian Path

**Image**

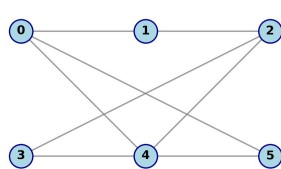

**Question**

Given the undirected, connected graph below, determine if there is an Eulerian path. If it exists, output the path as a list (e.g., [0,1,2,3]). If not, output 'No'.

**Reference Answer**

[0, 5, 4, 3, 2, 4, 0, 1, 2]

**Category:** Graph

**Difficulty:** Level 1

## Graph Isomorphism

**Image G1**          **Image G2**

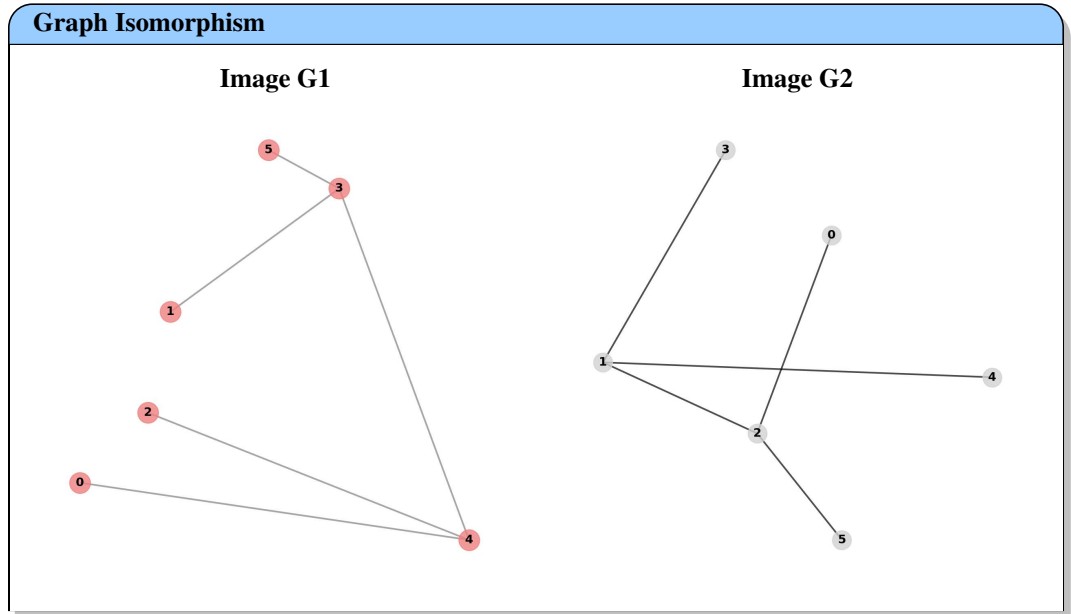

### Question

Given two connected undirected planar graphs G1 and G2 shown below, determine if they are isomorphic by analyzing their planar structure. Answer with 'Yes' or 'No'.

### Reference Answer

Yes

## Hamiltonian Cycle

**Image**

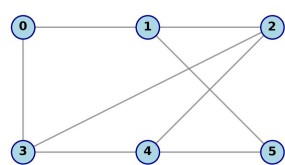

**Question**

Given the undirected, connected graph below, determine if there is a Hamiltonian cycle. If it exists, output the cycle as a list (e.g., [0,1,2,3]). If not, output 'No'.

**Reference Answer**

[0, 1, 5, 4, 2, 3]

**Category:** Graph

**Difficulty:** Level 1

## Hamiltonian Path

**Image**

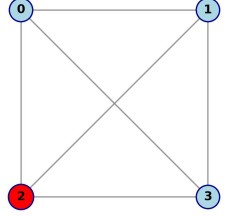

**Question**

Given an undirected graph below, determine whether a Hamiltonian path starting from vertex 2 (marked in red) exists. If it exists, output the path as a list (e.g., [0,1,2,3]). If not, output 'No'.

**Reference Answer**

[2, 0, 1, 3]

**Category:** Graph

**Difficulty:** Level 1

**Max Flow**

**Image**

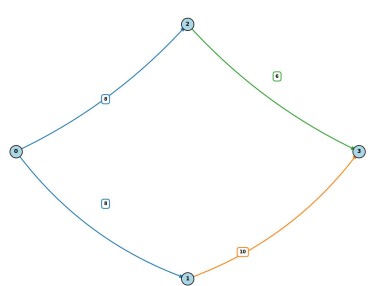

**Question**

Below is a layered directed acyclic graph (DAG) with capacities on each edge. Compute the maximum flow from node 0 to node 3. Answer with the maximum flow value (an integer).

**Reference Answer**

14

**Category:** Graph

**Difficulty:** Level 1

---

**Shortest Distance**

**Image**

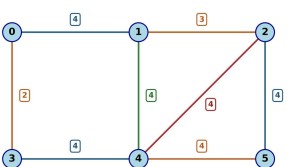

**Question**

Given a weighted undirected graph below, what is the shortest distance from node 0 to node 5? Answer with a number (can be integer or decimal).

**Reference Answer**

10

**Category:** Graph

**Difficulty:** Level 1

---

**Topological Sort**

**Image**

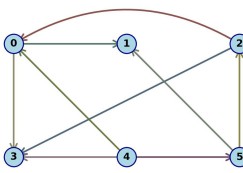

**Question**

Given the directed acyclic graph (DAG) shown in the image below, please list ONE possible valid topological orders. Answer with a list of numbers. For example: [0, 1, 2, 3].

**Reference Answer**

[4, 5, 2, 0, 1, 3]

**Category:** Graph

**Difficulty:** Level 1

## A.8   EXAMPLES IN MM-HELIX-100K

---

**Nibbles**

**Image**

---

**Question**

You are a puzzle solver focusing on Snake puzzles.

*Game Rules*: 1. Control a snake to move around the grid using directional commands (up, down, left, right) 2. The snake must eat all apples on the grid to win 3. When the snake eats an apple, it grows longer by one segment 4. The snake cannot collide with walls or itself 5. The snake moves one cell at a time in the chosen direction

*Input*: - An image showing the initial state with the snake and apples

*Goal*: Find a sequence of directional moves to eat all apples without the snake colliding with walls or itself.

*Output Format Requirements*: Your answer should be a sequence of directional moves separated by spaces. Valid moves are: up, down, left, right Example: up right down left up

---

**Rule-based CoT**

I need to solve this Snake puzzle by carefully analyzing the image and applying strategic reasoning. Let me work through this step by step.

### Step 1: Understanding the Game Rules and Mechanics

Let me first clarify the fundamental rules of this Snake puzzle:

**Core Movement Rules:** - The snake moves one cell at a time in four directions: up, down, left, right - The snake cannot immediately reverse direction (e.g., if moving right, cannot move left next) - Each move, the snake's head advances to the next cell in the chosen direction

**Collision Rules:** - The snake dies if it hits any wall (moves outside the grid boundaries) - The snake dies if its head collides with any part of its own body - These are the only two failure conditions

**Apple and Growth Mechanics:** - There are 3 apple(s) placed on the 8×8 grid - When the snake's head reaches an apple's position, the apple is consumed - Upon eating an apple, the snake's length increases by exactly 1 segment - The tail doesn't move for one turn when an apple is eaten (snake grows)

**Victory Condition:** - **Primary Goal:** Consume all 3 apple(s) without any collisions - The puzzle is solved when no apples remain on the board

**Strategic Implications:** - As the snake grows longer, navigation becomes more constrained - Path planning must account for future snake length - Early moves must preserve space for later maneuvers

### Step 2: Careful Image Analysis and Initial State Reading

Now I will read the image carefully to extract the precise initial game state:

**Grid Layout Analysis:** - Grid dimensions: 8 rows × 8 columns - Total cells available: 64 - Coordinate system: (row, col) with (0,0) at top-left

**Visual Grid Representation:** "' ........ ........ ...A.... ..S..... ..H..... .......A ....A... ........ "'
Legend: H = Snake Head, S = Snake Body, A = Apple, . = Empty Cell

**Snake Initial Configuration:** - Snake body positions: [(4, 2), (3, 2)] - Snake length: 2 segments - Head position: (4, 2) - Current facing direction: down - Tail position: (3, 2)

**Apple Locations Analysis:** - Apple positions: [(2, 3), (6, 4), (5, 7)] - Number of apples to collect: 3 - Apples sorted by Manhattan distance from head: [(2, 3), (6, 4), (5, 7)]

**Distance Analysis:** - Apple at (2, 3): 3 moves minimum (Manhattan distance) - Apple at (6, 4): 4 moves minimum (Manhattan distance) - Apple at (5, 7): 6 moves minimum (Manhattan distance)

**State Reading Verification:** Let me double-check my reading of the initial state: ✓ Grid size correctly identified as 8×8 ✓ Snake head at (4, 2) facing down ✓ Snake body has 2 segments total ✓ Found all 3 apples on the grid ✓ No overlapping positions between snake and apples

**Spatial Constraints:** - Available empty cells: 59 - Grid boundaries: rows [0, 7], columns [0, 7] - Movement constraints: cannot reverse direction, cannot hit walls or self

### Step 3: Strategic Exploration and Detailed Reasoning Process

Now I'll develop a comprehensive strategy to solve this puzzle through systematic exploration:

**Initial Strategic Assessment:**

**Apple-by-Apple Strategic Analysis:**

**Target 1: Apple at (2, 3)** - Manhattan distance: 3 moves - Direct path analysis: Need to move -2 vertically, 1 horizontally - Path considerations: * Check if snake body blocks direct routes * Evaluate alternative paths if direct route blocked * Consider space requirements for future growth * Ensure tail clearance for return paths

**Target 2: Apple at (6, 4)** - Manhattan distance: 4 moves - Direct path analysis: Need to move 2 vertically, 2 horizontally

**Target 3: Apple at (5, 7)** - Manhattan distance: 6 moves - Direct path analysis: Need to move 1 vertically, 5 horizontally

**Move-by-Move Strategic Reasoning:**

Let me trace through the solution logic step by step:

**Move 1: right** - Current head: (4, 2) → New head: (4, 3) - Direction change: down → right - No apple at (4, 3), snake maintains length 2 - Safety check: position (4, 3) is within bounds and collision-free

**Move 2: up** - Current head: (4, 3) → New head: (3, 3) - Direction change: right → up - No apple at (3, 3), snake maintains length 2 - Safety check: position (3, 3) is within bounds and collision-free

**Move 3: up** - Current head: (3, 3) → New head: (2, 3) - Direction change: up → up - ✓ Apple consumed at (2, 3)! Snake grows to length 3 - Safety check: position (2, 3) is within bounds and collision-free

**Move 4: right** - Current head: (2, 3) → New head: (2, 4) - Direction change: up → right - No apple at (2, 4), snake maintains length 3 - Safety check: position (2, 4) is within bounds and collision-free

**Move 5: down** - Current head: (2, 4) → New head: (3, 4) - Direction change: right → down - No apple at (3, 4), snake maintains length 3 - Safety check: position (3, 4) is within bounds and collision-free

**Move 6: down** - Current head: (3, 4) → New head: (4, 4) - Direction change: down → down - No apple at (4, 4), snake maintains length 3 - Safety check: position (4, 4) is within bounds and collision-free

**Move 7: down** - Current head: (4, 4) → New head: (5, 4) - Direction change: down → down - No apple at (5, 4), snake maintains length 3 - Safety check: position (5, 4) is within bounds and collision-free

**Move 8: down** - Current head: (5, 4) → New head: (6, 4) - Direction change: down → down - ✓ Apple consumed at (6, 4)! Snake grows to length 4 - Safety check: position (6, 4) is within bounds and collision-free

... (continuing for 4 more moves)

**Strategic Heuristics Applied:** - **Nearest-First Strategy:** Prioritize closest apples to minimize travel distance - **Space Preservation:** Avoid moves that create dead-ends or trap the snake - **Growth Planning:** Account for snake length increase when planning paths - **Boundary Awareness:** Use grid edges strategically to guide movement - **Tail Management:** Ensure the tail has clearance for complex maneuvers

**Alternative Path Consideration:** - If direct path to nearest apple is blocked, consider perimeter approach - Evaluate trade-offs between immediate apple collection vs. strategic positioning - Plan escape routes before entering confined spaces

**Risk Assessment:** - Monitor available space as snake grows longer - Avoid creating situations where the snake blocks its own path to remaining apples - Maintain flexibility for direction changes throughout the solution

### Step 4: Solution Validation and Comprehensive Reflection

Now I'll thoroughly validate the solution and reflect on the reasoning process:

**Solution Completeness Verification:** ✓All 3 apples are successfully collected ✓Solution contains 12 moves ✓Snake length progression: 2 → 5 (growth of 3)

**Collision Safety Verification:** Let me verify each move is safe: ✓All moves verified safe - no wall or self-collisions detected ✓Final snake length: 5 (expected: 5) ✓Remaining apples: 0 (expected: 0)

**Strategic Reflection:** - The solution demonstrates effective application of nearest-first strategy - Space management was crucial for avoiding self-trapping scenarios - Growth planning ensured the snake maintained maneuverability throughout

**Algorithm Efficiency Analysis:** - Solution efficiency: 12 moves for 3 apples - Average moves per apple: 4.0 - Path optimality: Solution balances directness with safety constraints

**Final Answer Confidence:** Based on thorough analysis and verification, the solution is: **right up up right down down down down right up right right**

This solution successfully guides the snake to collect all apples while avoiding all collision risks. The strategic approach demonstrates systematic planning and careful execution of the Snake puzzle mechanics.

--------------------------------------------------------------------------------

**MM-HELIX-100k**

<think> Alright, let's dive into this Snake puzzle. I'm looking at the grid and trying to get a feel for the layout — not just the positions, but the *space*, the flow, how the snake might move through it as it grows. This isn't just about getting from point A to B; it's about doing so in a way that doesn't paint the snake into a corner later.

First, let's orient ourselves. The grid is 8×8, and from what I can see, the snake starts with its head at (4,2), and there's a body segment just above it at (3,2). So it's pointing downward. There are three apples: one up at (2,3), another down at (6,4), and one way over to the right at (5,7).

Hmm. The closest apple is (2,3) — just two rows up and one column to the right from the head. That feels like a natural first target. But I can't just rush up — the body is directly above the head, so if I go up immediately, I'd be moving into where the body just was. Wait, no — the snake moves forward, and the tail follows. So as long as I don't try to reverse direction, I should be fine. But right now, the snake is facing down. If I go *up*, that's a 180-degree turn — which isn't allowed. So I can't go up first. That's a key constraint.

So, options from (4,2): I can't go down (would hit the wall? No — row 4 to 5 is fine), but down is the current direction. Wait — I *can* continue down, or go left or right. But not up, because that would be reversing.

Wait — actually, the rule is you can't reverse into your own body. Since the body is at (3,2), above the head, moving up would immediately collide with the body. So yes, up is invalid. So first move must be either right, left, or down.

Left from (4,2) is (4,1) — empty. Right is (4,3) — also empty. Down is (5,2) — also clear. So three options.

Now, the apple at (2,3) is the closest, but I can't go straight up. So I need to approach it from a different angle, or create space to maneuver.

Maybe I can go right first, then up. That might work. Let's try that in my mind.

Start: head at (4,2), body at (3,2).

Move 1: **right** → head to (4,3). Body shifts — now body is at (4,2), tail was at (3,2) but now moves to (4,2)? Wait no — when the head moves, the entire snake shifts forward: the head

moves to (4,3), the previous head (4,2) becomes the new body segment, and the old body (3,2) disappears? No — actually, in Snake, the tail moves forward too unless you eat an apple.

Wait — correction: in standard Snake, when you move, the head advances, and the tail retracts — so the body follows the path. Only when you eat an apple does the tail *not* retract, so the snake grows.

So after moving right: head at (4,3), body at (4,2), tail at (3,2)? No — the body should be the previous head position, and the tail moves up.

Let's track it properly:

- Initial: head (4,2), body (3,2) — so the snake occupies (3,2) and (4,2), with head at bottom. - Move **right**: head goes to (4,3). The body follows: the segment that was at (4,2) is now the body, and the tail that was at (3,2) moves to (4,2). So now snake is: head (4,3), body (4,2), tail (4,2)? No — two segments: head and one body. After move, positions are: (4,3) and (4,2). The old body (3,2) is gone — it moved down to (4,2)? No.

Actually, in a two-segment snake, when the head moves, the body takes the head's old position, and the tail (which is the body) leaves its old position.

So: start — positions: (3,2) and (4,2), with (4,2) as head.

After **right**: head moves to (4,3). The body moves to where the head was: (4,2). So now snake is at (4,2) and (4,3), head at (4,3), body at (4,2). The old tail (3,2) is no longer part of the snake.

Yes, that makes sense.

Now, from (4,3), can I go **up**? To (3,3). Is that safe? (3,3) is empty — no apple, no wall. And I'm not reversing — I was going right, now up — that's a 90-degree turn, allowed. So yes.

Move 2: **up** → head to (3,3), body to (4,3), tail to (4,2)? No: body was at (4,2), now moves to (4,3)? Let's track:

After move 1: head (4,3), body (4,2)

Move 2: head to (3,3), body takes (4,3), so now snake is (4,2), (4,3), (3,3)? No — only two segments. After move, positions: head (3,3), body (4,3). The old body (4,2) is now gone — the snake shifted.

Wait — no: when the head moves to (3,3), the body moves to the head's previous position (4,3), and the tail (which was at (4,2)) is left behind. So yes, snake is now at (4,3) and (3,3).

Now, from (3,3), I can go **up** to (2,3) — and there's an apple! And I'm not reversing — I was going up, continue up — that's fine.

Move 3: **up** to (2,3) — eat the apple! Snake grows. So now, when the head moves to (2,3), the body follows to (3,3), and the tail (which was at (4,3)) moves to (3,3)? No — because it grew, the tail doesn't retract. So the body segments are: new head (2,3), then (3,3), then (4,3). So now length is 3.

Great — one apple down.

Now, where to next? The remaining apples are at (6,4) and (5,7). (6,4) is closer — from (2,3) to (6,4) is 4 down, 1 right — Manhattan distance 5. (5,7) is farther.

But I need to get down there. From (2,3), I'm facing up, but I just moved up. I can go left, right, or down — not up (would be reversing if I were going up, but I just came from down? Wait — my last move was **up**, so I'm facing up. So I can go left, right, or up — but not down (reverse).

But I want to go down eventually. So I need to turn left or right first.

Right to (2,4)? That's empty. Let's try **right**.

Move 4: **right** to (2,4). Now facing right. Body: (2,3), (3,3). Tail still at (4,3)? Let's see: after eating, snake was at (2,3), (3,3), (4,3). After moving head to (2,4), body follows: (2,3) → (2,4)? No — head moves to (2,4), the first body segment moves to (2,3), the second moves to (3,3), and the tail (4,3) moves to (3,3)? No — with growth, when you move *after* eating, the tail *does* move — only during the eat move does the tail not retract.

Clarify: when you eat an apple, that move causes the snake to grow — so the tail does *not* move that turn. But on subsequent moves, the tail moves normally.

So after move 3 (eating), snake occupies: (2,3), (3,3), (4,3) — three segments.

Now move 4: **right** → head to (2,4). The body shifts: (2,3) → (2,4)? No — the head moves to (2,4), the segment that was at (2,3) moves to where the head was — but the head was at (2,3)? No — head was at (2,3), moves to (2,4). So the first body segment (at (3,3)) moves to (2,3), and the tail (at (4,3)) moves to (3,3). So now snake is at (2,4), (2,3), (3,3).

Yes.

Now, I want to go down toward (6,4). From (2,4), I can go **down** — that's allowed, since I'm facing right, down is a turn.

Move 5: **down** to (3,4). Is (3,4) safe? Yes, empty. Snake now: head (3,4), body (2,4), (2,3)? No — after move: head (3,4), first body (2,4), second body (2,3)? No — the body segments shift: (2,4) → (3,4)? No.

Head was at (2,4), moves to (3,4). The segment that was at (2,3) moves to (2,4), and the one at (3,3) moves to (2,3). So now: (3,4), (2,4), (2,3).

Wait — the snake was at (2,4), (2,3), (3,3). After moving head to (3,4), the body follows: (2,4) becomes the new body position, but it moves to where the head was — (2,4)? I'm getting tangled.

Standard rule: each segment moves to the position of the one ahead of it.

So: - Head moves from (2,4) to (3,4) - The first body segment (at (2,3)) moves to (2,4) - The second body segment (at (3,3)) moves to (2,3)

So snake now occupies: (3,4), (2,4), (2,3)

(3,3) is now empty.

Now, continue **down** to (4,4)

Move 6: **down** → head to (4,4). Then body: (3,4) → (4,4)? No — head (3,4) → (4,4), body (2,4) → (3,4), tail (2,3) → (2,4). So snake: (4,4), (3,4), (2,4)

Still safe.

Move 7: **down** → to (5,4). Body: (4,4) → (5,4)? Head (4,4) → (5,4), body (3,4) → (4,4), tail (2,4) → (3,4). So (5,4), (4,4), (3,4)

Move 8: **down** → to (6,4). And there's an apple! Eat it. Snake grows to length 4.

So now, after eating: head at (6,4), and the body follows: previous head (5,4) becomes body, (4,4) becomes next, (3,4) becomes tail — but since it grew, the tail doesn't retract. So segments: (6,4), (5,4), (4,4), (3,4)

Now, one apple left at (5,7). From (6,4), I need to go right — but I'm facing down (last move was down), so I can go left, right, or down — not up.

(5,7) is at row 5, column 7. I'm at row 6, column 4. So I need to go up one and right three.

But I can't go up directly — that would be reversing. So I need to turn right first.

Move 9: **right** → to (6,5). Now facing right. Snake: head (6,5), body (6,4), (5,4), (4,4) — tail at (4,4)? Let's track: after eating, snake at (6,4), (5,4), (4,4), (3,4). Move right: head to (6,5), body to (6,4), then (5,4), then (4,4) — tail moves to (5,4)? No — each segment shifts forward: (6,4) → (6,5)? No.

Head was at (6,4), moves to (6,5). The first body segment (5,4) moves to (6,4), the next (4,4) moves to (5,4), the tail (3,4) moves to (4,4). So now: (6,5), (6,4), (5,4), (4,4)

Good.

Move 10: **right** → to (6,6). Head (6,6), body (6,5), (6,4), (5,4)

Move 11: **right** → to (6,7). Head (6,7), body (6,6), (6,5), (6,4)

Now, I need to go to (5,7). So **up**. Am I allowed? I'm facing right, up is a 90-degree turn — yes.

Move 12: **up** → to (5,7). Apple! Eat it. Snake grows to length 5.

Now all apples are collected.

Let me double-check for collisions:

- All moves stayed within 0–7 rows and columns. - No move went into a wall. - Did the snake ever hit itself? Let's see the path:

After move 3: snake at (2,3), (3,3), (4,3) — fine.

After move 4: (2,4), (2,3), (3,3) — no overlap.

After move 5: (3,4), (2,4), (2,3) — all distinct.

After move 6: (4,4), (3,4), (2,4)

After move 7: (5,4), (4,4), (3,4)

After move 8: eat at (6,4) → (6,4), (5,4), (4,4), (3,4)

After move 9: (6,5), (6,4), (5,4), (4,4) — (6,4) is adjacent, not overlapping.

After move 10: (6,6), (6,5), (6,4), (5,4)

After move 11: (6,7), (6,6), (6,5), (6,4)

After move 12: (5,7), (6,7), (6,6), (6,5), (6,4) — all good.

No self-collisions, no walls, all apples eaten.

The path feels natural: start by going right to avoid reversing, then up to get the first apple. Then snake down the right side, collecting the second apple on the way down. Then move right across the bottom, and finally up to the last apple.

I could have tried to go for (5,7) first, but that would have been longer and riskier. This way, I kept the snake compact early on, and used the open space on the right to move freely.

So the final answer is: right up up right down down down down right up right right </think>

Final Answer: right up up right down down down down right up right right

## A.9 COMPLETE BREAKDOWN RESULTS

This section provides the full breakdown results over the four task types of MM-HELIX.

### A.9.1 BREAKDOWN RESULTS ON THE ALGORITHMS CATEGORY

Table 8: Breakdown results on the **Algorithms** category.

| Model | 24 | BuySell | Container | Hills | Crypto | HIndex | Rect | LIS | Rain |
|---|---|---|---|---|---|---|---|---|---|
| *Proprietary Models* | | | | | | | | | |
| GPT-5 (OpenAI, 2025b) | 96.7 | 80.0 | 93.3 | 73.3 | 100.0 | 96.7 | 90.0 | 93.3 | 73.3 |
| Seed-1.5-VL (Guo et al., 2025b) | 100.0 | 80.0 | 83.3 | 60.0 | 86.7 | 83.3 | 73.3 | 73.3 | 70.0 |
| o4-mini (OpenAI, 2025c) | 86.7 | 10.0 | 36.7 | 43.3 | 60.0 | 66.7 | 50.0 | 63.3 | 40.0 |
| Gemini-2.5-Flash (Comanici et al., 2025) | 96.7 | 43.3 | 66.7 | 56.7 | 83.3 | 76.7 | 56.7 | 70.0 | 50.0 |
| GPT-4.1 (OpenAI, 2025a) | 63.3 | 46.7 | 56.7 | 16.7 | 26.7 | 60.0 | 33.3 | 43.3 | 53.3 |
| GPT-4o (OpenAI, 2024) | 10.0 | 30.0 | 23.3 | 0.0 | 0.0 | 30.0 | 23.3 | 33.3 | 20.0 |
| *Open-Source Models* | | | | | | | | | |
| Intern-S1-241B-A28B (Bai et al., 2025a) | 86.7 | 80.0 | 70.0 | 83.3 | 63.3 | 46.7 | 66.7 | 83.3 | 43.3 |
| GLM-4.5V-106B-A12B-Thinking (Team et al., 2025b) | 56.7 | 16.7 | 40.0 | 3.3 | 23.3 | 23.3 | 33.3 | 53.3 | 13.3 |
| Kimi-VL-16B-A3B-Thinking-2506 (Team et al., 2025a) | 90.0 | 36.7 | 33.3 | 10.0 | 16.7 | 43.3 | 26.7 | 43.3 | 26.7 |
| GLM-4.1V-9B-Thinking (Team et al., 2025b) | 76.7 | 10.0 | 43.3 | 13.3 | 20.0 | 30.0 | 16.7 | 30.0 | 36.7 |
| Qwen-2.5-VL-72B (Bai et al., 2025b) | 13.3 | 20.0 | 26.7 | 16.7 | 0.0 | 43.3 | 6.7 | 30.0 | 10.0 |
| Qwen-2.5-VL-32B (Bai et al., 2025b) | 33.3 | 26.7 | 16.7 | 0.0 | 3.3 | 16.7 | 3.3 | 26.7 | 10.0 |
| InternVL3-78B (Zhu et al., 2025) | 46.7 | 20.0 | 20.0 | 6.7 | 6.7 | 10.0 | 10.0 | 10.0 | 0.0 |
| InternVL3-38B (Zhu et al., 2025) | 43.3 | 3.3 | 23.3 | 3.3 | 3.3 | 13.3 | 3.3 | 26.7 | 6.7 |
| Llama-4-Scout-109B-A17B-16E (Meta, 2025) | 66.7 | 30.0 | 3.3 | 10.0 | 0.0 | 6.7 | 3.3 | 20.0 | 6.7 |
| MiniCPM-V-4.5-8B (Yu et al., 2025b) | 53.3 | 6.7 | 20.0 | 13.3 | 6.7 | 30.0 | 13.3 | 33.3 | 3.3 |
| QVQ-72B-Preview (Team, 2024) | 76.7 | 20.0 | 26.7 | 3.3 | 0.0 | 20.0 | 3.3 | 33.3 | 6.7 |
| Ovis2-34B (Lu et al., 2024) | 23.3 | 0.0 | 3.3 | 6.7 | 0.0 | 20.0 | 13.3 | 26.7 | 0.0 |
| Gemma-3-27B-IT (Team, 2025) | 10.0 | 0.0 | 13.3 | 3.3 | 0.0 | 23.3 | 10.0 | 30.0 | 3.3 |
| Qwen-2.5-VL-7B (Bai et al., 2025b) | 10.0 | 0.0 | 6.7 | 0.0 | 0.0 | 10.0 | 3.3 | 23.3 | 0.0 |
| InternVL3-8B (Zhu et al., 2025) | 10.0 | 0.0 | 6.7 | 3.3 | 0.0 | 10.0 | 0.0 | 23.3 | 0.0 |
| Ovis2-8B (Lu et al., 2024) | 13.3 | 0.0 | 0.0 | 0.0 | 0.0 | 10.0 | 0.0 | 6.7 | 0.0 |
| *Ours* | | | | | | | | | |
| MM-HELIX-7B-Thinking | 56.7 | 30.0 | 46.7 | 40.0 | 10.0 | 46.7 | 26.7 | 43.3 | 13.3 |

The abbreviations used in the table above are explained in the following table:

Table 9: Abbreviation list of the keywords in the **Algorithms** category.

| Abbreviation | Task |
|---|---|
| 24 | 24 Points |
| BuySell | Best Time to Buy and Sell Stock |
| Container | Container With Most Water |
| Hills | Counting Hills and Valleys |
| Crypt | CryptoMath |
| HIndex | H-Index |
| Rect | Largest Rectangle in Histogram |
| LIS | Longest Increasing Subsequence |
| Rain | Trapping Rain Water |

### A.9.2 BREAKDOWN RESULTS ON THE GRAPHS CATEGORY

Table 10: Breakdown results on the **Graphs** category.

| Model | EulerCyc | EulerPath | GraphIso | HamilCyc | HamilPath | MaxFlow | ShortDist | TopoSort |
|---|---|---|---|---|---|---|---|---|
| *Proprietary Models* | | | | | | | | |
| GPT-5 (OpenAI, 2025b) | 33.3 | 33.3 | 53.3 | 40.0 | 60.0 | 80.0 | 90.0 | 13.3 |
| Seed-1.5-VL (Guo et al., 2025b) | 23.3 | 30.0 | 56.7 | 23.3 | 46.7 | 70.0 | 60.0 | 13.3 |
| o4-mini (OpenAI, 2025c) | 33.3 | 33.3 | 53.3 | 33.3 | 50.0 | 66.7 | 56.7 | 10.0 |
| Gemini-2.5-Flash (Comanici et al., 2025) | 30.0 | 36.7 | 43.3 | 26.7 | 46.7 | 63.3 | 66.7 | 13.3 |
| GPT-4.1 (OpenAI, 2025a) | 10.0 | 20.0 | 63.3 | 20.0 | 33.3 | 70.0 | 60.0 | 3.3 |
| GPT-4o (OpenAI, 2024) | 6.7 | 26.7 | 56.7 | 16.7 | 20.0 | 33.3 | 43.3 | 0.0 |
| *Open-Source Models* | | | | | | | | |
| Intern-S1-241B-A28B (Bai et al., 2025a) | 16.7 | 26.7 | 50.0 | 16.7 | 23.3 | 50.0 | 56.7 | 0.0 |
| GLM-4.5V-106B-A12B-Thinking (Team et al., 2025b) | 0.0 | 10.0 | 6.7 | 10.0 | 20.0 | 30.0 | 13.3 | 0.0 |
| Kimi-VL-16B-A3B-Thinking-2506 (Team et al., 2025a) | 16.7 | 20.0 | 46.7 | 16.7 | 26.7 | 40.0 | 20.0 | 0.0 |
| GLM-4.1V-9B-Thinking (Team et al., 2025b) | 16.7 | 23.3 | 46.7 | 16.7 | 33.3 | 50.0 | 43.3 | 3.3 |
| Qwen-2.5-VL-72B (Bai et al., 2025b) | 16.7 | 23.3 | 56.7 | 10.0 | 20.0 | 43.3 | 36.7 | 0.0 |
| Qwen-2.5-VL-32B (Bai et al., 2025b) | 13.3 | 20.0 | 30.0 | 16.7 | 23.3 | 40.0 | 36.7 | 0.0 |
| InternVL3-78B (Zhu et al., 2025) | 10.0 | 20.0 | 46.7 | 16.7 | 26.7 | 40.0 | 40.0 | 3.3 |
| InternVL3-38B (Zhu et al., 2025) | 10.0 | 23.3 | 46.7 | 16.7 | 13.3 | 33.3 | 36.7 | 0.0 |
| Llama-4-Scout-109B-A17B-16E (Meta, 2025) | 16.7 | 26.7 | 43.3 | 10.0 | 23.3 | 26.7 | 20.0 | 3.3 |
| MiniCPM-V-4.5-8B (Yu et al., 2025b) | 6.7 | 23.3 | 40.0 | 20.0 | 16.7 | 26.7 | 30.0 | 3.3 |
| QVQ-72B-Preview (Team, 2024) | 16.7 | 16.7 | 36.7 | 6.7 | 13.3 | 20.0 | 20.0 | 3.3 |
| Ovis2-34B (Lu et al., 2024) | 16.7 | 23.3 | 53.3 | 23.3 | 16.7 | 23.3 | 13.3 | 6.7 |
| Gemma-3-27B-IT (Team, 2025) | 16.7 | 26.7 | 33.3 | 16.7 | 23.3 | 36.7 | 20.0 | 3.3 |
| Qwen-2.5-VL-7B (Bai et al., 2025b) | 10.0 | 23.3 | 53.3 | 0.0 | 13.3 | 23.3 | 20.0 | 0.0 |
| InternVL3-8B (Zhu et al., 2025) | 13.3 | 26.7 | 33.3 | 16.7 | 23.3 | 6.7 | 13.3 | 0.0 |
| Ovis2-8B (Lu et al., 2024) | 16.7 | 10.0 | 26.7 | 23.3 | 13.3 | 16.7 | 16.7 | 0.0 |
| *Ours* | | | | | | | | |
| MM-HELIX-7B-Thinking | 16.7 | 23.3 | 20.0 | 10.0 | 26.7 | 26.7 | 30.0 | 3.3 |

The abbreviations used in the table above are explained in the following table:

Table 11: Abbreviation list of the keywords in the **Graphs** category.

| Abbreviation | Task |
|---|---|
| EulerCyc | Eulerian Cycle |
| EulerPath | Eulerian Path |
| GraphIso | Graph Isomorphism |
| HamilCyc | Hamiltonian Cycle |
| HamilPath | Hamiltonian Path |
| MaxFlow | Max Flow |
| ShortDist | Shortest Distance (Weighted) |
| TopoSort | Topological Sort |

### A.9.3 Breakdown results on the Puzzles category

Table 12: Breakdown results on the **Puzzles** category (Part 1).

| Model | Aqua | Bina | Brid | Calcu | Camp | Eule | Futo | Hito | Kaku | Kuku |
|---|---|---|---|---|---|---|---|---|---|---|
| *Proprietary Models* | | | | | | | | | | |
| GPT-5 (OpenAI, 2025b) | 33.3 | 23.3 | 83.3 | 30.0 | 63.3 | 53.3 | 33.3 | 83.3 | 26.7 | 100.0 |
| Seed-1.5-VL (Guo et al., 2025b) | 10.0 | 30.0 | 50.0 | 16.7 | 86.7 | 60.0 | 20.0 | 40.0 | 36.7 | 63.3 |
| o4-mini (OpenAI, 2025c) | 26.7 | 13.3 | 73.3 | 23.3 | 53.3 | 50.0 | 30.0 | 43.3 | 43.3 | 76.7 |
| Gemini-2.5-Flash (Comanici et al., 2025) | 3.3 | 20.0 | 60.0 | 3.3 | 46.7 | 46.7 | 16.7 | 63.3 | 36.7 | 40.0 |
| GPT-4.1 (OpenAI, 2025a) | 3.3 | 0.0 | 46.7 | 13.3 | 13.3 | 33.3 | 10.0 | 40.0 | 16.7 | 60.0 |
| GPT-4o (OpenAI, 2024) | 0.0 | 20.0 | 0.0 | 3.3 | 16.7 | 20.0 | 10.0 | 16.7 | 13.3 | 33.3 |
| *Open-Source Models* | | | | | | | | | | |
| Intern-S1-241B-A28B (Bai et al., 2025a) | 3.3 | 23.3 | 60.0 | 26.7 | 20.0 | 16.7 | 20.0 | 20.0 | 30.0 | 0.0 |
| GLM-4.5V-106B-A12B-Thinking (Team et al., 2025b) | 13.3 | 30.0 | 13.3 | 6.7 | 60.0 | 6.7 | 6.7 | 30.0 | 0.0 | 33.3 |
| Kimi-VL-16B-A3B-Thinking-2506 (Team et al., 2025a) | 3.3 | 16.7 | 20.0 | 6.7 | 16.7 | 13.3 | 26.7 | 13.3 | 16.7 | 10.0 |
| GLM-4.1V-9B-Thinking (Team et al., 2025b) | 6.7 | 16.7 | 6.7 | 13.3 | 40.0 | 10.0 | 3.3 | 16.7 | 13.3 | 20.0 |
| Qwen-2.5-VL-72B (Bai et al., 2025b) | 0.0 | 6.7 | 13.3 | 10.0 | 23.3 | 16.7 | 10.0 | 6.7 | 0.0 | 6.7 |
| Qwen-2.5-VL-32B (Bai et al., 2025b) | 3.3 | 0.0 | 10.0 | 3.3 | 6.7 | 3.3 | 16.7 | 0.0 | 6.7 | 13.3 |
| InternVL3-78B (Zhu et al., 2025) | 0.0 | 0.0 | 30.0 | 26.7 | 3.3 | 3.3 | 6.7 | 3.3 | 0.0 | 13.3 |
| InternVL3-38B (Zhu et al., 2025) | 3.3 | 3.3 | 16.7 | 20.0 | 0.0 | 3.3 | 13.3 | 10.0 | 6.7 | 10.0 |
| Llama-4-Scout-109B-A17B-16E (Meta, 2025) | 6.7 | 10.0 | 13.3 | 3.3 | 30.0 | 16.7 | 3.3 | 23.3 | 10.0 | 3.3 |
| MiniCPM-V-4.5-8B (Yu et al., 2025b) | 6.7 | 3.3 | 10.0 | 10.0 | 20.0 | 10.0 | 20.0 | 6.7 | 6.7 | 6.7 |
| QVQ-72B-Preview (Team, 2024) | 10.0 | 13.3 | 6.7 | 6.7 | 6.7 | 6.7 | 16.7 | 10.0 | 13.3 | 0.0 |
| Ovis2-34B (Lu et al., 2024) | 13.3 | 30.0 | 6.7 | 13.3 | 6.7 | 13.3 | 30.0 | 10.0 | 0.0 | 10.0 |
| Gemma-3-27B-IT (Team, 2025) | 6.7 | 3.3 | 10.0 | 3.3 | 0.0 | 0.0 | 6.7 | 13.3 | 10.0 | 3.3 |
| Qwen-2.5-VL-7B (Bai et al., 2025b) | 13.3 | 0.0 | 3.3 | 0.0 | 6.7 | 0.0 | 10.0 | 6.7 | 10.0 | 16.7 |
| InternVL3-8B (Zhu et al., 2025) | 6.7 | 0.0 | 3.3 | 16.7 | 10.0 | 0.0 | 10.0 | 3.3 | 6.7 | 0.0 |
| Ovis2-8B (Lu et al., 2024) | 0.0 | 10.0 | 0.0 | 0.0 | 0.0 | 6.7 | 3.3 | 10.0 | 0.0 | 3.3 |
| *Ours* | | | | | | | | | | |
| MM-HELIX-7B-Thinking | 3.3 | 23.3 | 50.0 | 6.7 | 46.7 | 43.3 | 20.0 | 13.3 | 30.0 | 26.7 |

Table 13: Breakdown results on the **Puzzles** category (Part 2).

| Model | Nono | Num | Shin | Sky | Snak | Sudo | Tapa | WLad | WSch |
|---|---|---|---|---|---|---|---|---|---|
| *Proprietary Models* | | | | | | | | | |
| GPT-5 (OpenAI, 2025b) | 26.7 | 86.7 | 80.0 | 53.3 | 10.0 | 26.7 | 50.0 | 36.7 | 100.0 |
| Seed-1.5-VL (Guo et al., 2025b) | 3.3 | 6.7 | 70.0 | 40.0 | 100.0 | 50.0 | 33.3 | 20.0 | 60.0 |
| o4-mini (OpenAI, 2025c) | 13.3 | 50.0 | 43.3 | 43.3 | 96.7 | 3.3 | 43.3 | 30.0 | 100.0 |
| Gemini-2.5-Flash (Comanici et al., 2025) | 0.0 | 40.0 | 40.0 | 40.0 | 83.3 | 40.0 | 36.7 | 10.0 | 70.0 |
| GPT-4.1 (OpenAI, 2025a) | 0.0 | 16.7 | 53.3 | 20.0 | 43.3 | 0.0 | 3.3 | 10.0 | 33.3 |
| GPT-4o (OpenAI, 2024) | 0.0 | 0.0 | 6.7 | 3.3 | 33.3 | 0.0 | 0.0 | 13.3 | 13.3 |
| *Open-Source Models* | | | | | | | | | |
| Intern-S1-241B-A28B (Bai et al., 2025a) | 0.0 | 26.7 | 13.3 | 23.3 | 53.3 | 53.3 | 16.7 | 0.0 | 43.3 |
| GLM-4.5V-106B-A12B-Thinking (Team et al., 2025b) | 0.0 | 0.0 | 0.0 | 6.7 | 40.0 | 20.0 | 6.7 | 6.7 | 50.0 |
| Kimi-VL-16B-A3B-Thinking-2506 (Team et al., 2025a) | 0.0 | 10.0 | 3.3 | 6.7 | 50.0 | 10.0 | 0.0 | 0.0 | 26.7 |
| GLM-4.1V-9B-Thinking (Team et al., 2025b) | 0.0 | 10.0 | 0.0 | 3.3 | 30.0 | 3.3 | 6.7 | 0.0 | 0.0 |
| Qwen-2.5-VL-72B (Bai et al., 2025b) | 0.0 | 6.7 | 16.7 | 3.3 | 13.3 | 6.7 | 0.0 | 0.0 | 10.0 |
| Qwen-2.5-VL-32B (Bai et al., 2025b) | 0.0 | 6.7 | 3.3 | 3.3 | 16.7 | 3.3 | 0.0 | 0.0 | 3.3 |
| InternVL3-78B (Zhu et al., 2025) | 0.0 | 0.0 | 6.7 | 3.3 | 6.7 | 0.0 | 0.0 | 3.3 | 0.0 |
| InternVL3-38B (Zhu et al., 2025) | 0.0 | 0.0 | 3.3 | 3.3 | 3.3 | 0.0 | 0.0 | 0.0 | 3.3 |
| Llama-4-Scout-109B-A17B-16E (Meta, 2025) | 0.0 | 0.0 | 20.0 | 3.3 | 13.3 | 0.0 | 0.0 | 6.7 | 16.7 |
| MiniCPM-V-4.5-8B (Yu et al., 2025b) | 0.0 | 0.0 | 0.0 | 0.0 | 6.7 | 0.0 | 0.0 | 0.0 | 16.7 |
| QVQ-72B-Preview (Team, 2024) | 0.0 | 0.0 | 6.7 | 0.0 | 0.0 | 0.0 | 0.0 | 6.7 | 10.0 |
| Ovis2-34B (Lu et al., 2024) | 0.0 | 0.0 | 0.0 | 0.0 | 0.0 | 3.3 | 3.3 | 0.0 | 0.0 |
| Gemma-3-27B-IT (Team, 2025) | 0.0 | 0.0 | 3.3 | 0.0 | 3.3 | 0.0 | 0.0 | 0.0 | 10.0 |
| Qwen-2.5-VL-7B (Bai et al., 2025b) | 0.0 | 0.0 | 0.0 | 0.0 | 3.3 | 0.0 | 0.0 | 6.7 | 3.3 |
| InternVL3-8B (Zhu et al., 2025) | 0.0 | 0.0 | 0.0 | 3.3 | 0.0 | 0.0 | 6.7 | 0.0 | 3.3 |
| Ovis2-8B (Lu et al., 2024) | 0.0 | 0.0 | 0.0 | 0.0 | 0.0 | 3.3 | 6.7 | 0.0 | 6.7 |
| *Ours* | | | | | | | | | |
| MM-HELIX-7B-Thinking | 13.3 | 23.3 | 60.0 | 20.0 | 16.7 | 30.0 | 6.7 | 6.7 | 40.0 |

The abbreviations used in the table above are explained in the following table:

Table 14: Abbreviation list of the keywords in the **Puzzles** category.

| Abbreviation | Task |
|---|---|
| Aqua | Aquarium |
| Bina | Binairo |
| Brid | Bridges |
| Calcu | Calcudoku |
| Camp | Campsite |
| Eule | Eulero |
| Futo | Futoshiki |
| Hito | Hitori |
| Kaku | Kakuro |
| Kuku | Kukurasu |
| Nono | Nonogram |
| Num | Numbrix |
| Shin | Shingoki |
| Sky | Skyscrapers |
| Snak | Snake |
| Sudo | Sudoku |
| Tapa | Tapa |
| WLad | Word Ladder |
| WSch | Wordsearch |

### A.9.4 BREAKDOWN RESULTS ON THE GAMES CATEGORY

Table 15: Breakdown results on the **Games** category.

| Model | Maze | Mine | Nib | Slide | Soko | Hanoi |
|---|---|---|---|---|---|---|
| *Proprietary Models* | | | | | | |
| GPT-5-2025-08-27(16K) (OpenAI, 2025b) | 10.0 | 23.3 | 10.0 | 86.7 | 16.7 | 93.3 |
| Seed-1.5-VL(16K) (Guo et al., 2025b) | 6.7 | 53.3 | 20.0 | 63.3 | 3.3 | 53.3 |
| o4-mini(16K) (OpenAI, 2025c) | 6.7 | 26.7 | 10.0 | 66.7 | 13.3 | 90.0 |
| Gemini-2.5-flash(16K) (Comanici et al., 2025) | 0.0 | 50.0 | 13.3 | 46.7 | 3.3 | 56.7 |
| GPT-4.1-2025-04-14(32K) (OpenAI, 2025a) | 3.3 | 0.0 | 0.0 | 3.3 | 0.0 | 46.7 |
| GPT-4o (OpenAI, 2024) | 0.0 | 0.0 | 0.0 | 3.3 | 0.0 | 36.7 |
| *Open-Source Models* | | | | | | |
| Intern-S1-241B-A28B (Bai et al., 2025a) | 0.0 | 20.0 | 0.0 | 36.7 | 0.0 | 33.3 |
| GLM-4.5V-106B-A12B-Thinking (Team et al., 2025b) | 0.0 | 16.7 | 3.3 | 10.0 | 3.3 | 50.0 |
| Kimi-VL-16B-A3B-Thinking-2506 (Team et al., 2025a) | 0.0 | 3.3 | 0.0 | 3.3 | 0.0 | 10.0 |
| GLM-4.1V-9B-Thinking (Team et al., 2025b) | 0.0 | 0.0 | 3.3 | 3.3 | 0.0 | 26.7 |
| Qwen-2.5-VL-72B (Bai et al., 2025b) | 0.0 | 20.0 | 0.0 | 36.7 | 3.3 | 26.7 |
| Qwen-2.5-VL-32B (Bai et al., 2025b) | 0.0 | 16.7 | 3.3 | 33.3 | 0.0 | 6.7 |
| InternVL3-78B (Zhu et al., 2025) | 0.0 | 0.0 | 3.3 | 6.7 | 0.0 | 16.7 |
| InternVL3-38B (Zhu et al., 2025) | 0.0 | 3.3 | 3.3 | 10.0 | 6.7 | 13.3 |
| Llama-4-Scout-109B-A17B-16E (Meta, 2025) | 0.0 | 3.3 | 0.0 | 10.0 | 0.0 | 33.3 |
| MiniCPM-V-4.5-8B (Yu et al., 2025b) | 0.0 | 0.0 | 0.0 | 3.3 | 3.3 | 13.3 |
| QVQ-72B-Preview (Team, 2024) | 0.0 | 3.3 | 3.3 | 6.7 | 0.0 | 16.7 |
| Ovis2-34B (Lu et al., 2024) | 0.0 | 0.0 | 0.0 | 3.3 | 0.0 | 6.7 |
| Gemma-3-27B-IT (Team, 2025) | 0.0 | 0.0 | 0.0 | 3.3 | 0.0 | 3.3 |
| Qwen-2.5-VL-7B (Bai et al., 2025b) | 0.0 | 0.0 | 0.0 | 0.0 | 0.0 | 3.3 |
| InternVL3-8B (Zhu et al., 2025) | 0.0 | 0.0 | 0.0 | 0.0 | 0.0 | 6.7 |
| Ovis2-8B (Lu et al., 2024) | 0.0 | 0.0 | 0.0 | 0.0 | 3.3 | 3.3 |
| *Ours* | | | | | | |
| MM-HELIX-7B-Thinking | 3.3 | 16.7 | 23.3 | 26.7 | 3.3 | 26.7 |

The abbreviations used in the table above are explained in the following table:

Table 16: Abbreviation list of the keywords in the **Games** category.

| Abbreviation | Task |
|---|---|
| Maze | Maze |
| Mine | Minesweeper |
| Nib | Nibbles |
| Slide | Sliding Puzzle |
| Soko | Sokoban |
| Hanoi | Tower of Hanoi |

