# OpenReview forum: "MM-HELIX: Boosting Multimodal Long-Chain Reflective Reasoning with Holistic Platform and Adaptive Hybrid Policy Optimization"
_ICLR.cc/2026/Conference — ICLR 2026 Poster_

### Official Review · Reviewer_EG1J · 2025-10-26

**Soundness:** 4
**Presentation:** 3
**Contribution:** 4
**Rating:** 6
**Confidence:** 4

**Summary:**

This paper presents MM-HELIX, a novel multimodal benchmark designed to evaluate MLLMs on long-chain, reflective reasoning tasks that require iteration and backtracking. MM-HELIX consists of 42 diverse, programmatically generated tasks across four domains—algorithms, graphs, puzzles, and games—spanning five difficulty levels. Through evaluation on this benchmark, the authors identify key shortcomings in existing MLLMs and propose a training pipeline that features: 1. SERG for generating large-scale reflective chain-of-thought data, and 2. AHPO for integrating off-policy SFT and on-policy RL. Empirical results demonstrate significant improvements on both MM-HELIX and standard mathematical and logic benchmarks, highlighting strong generalizability.

**Strengths:**

**1. Originality:** The work makes notable contributions through (1) the MM-HELIX benchmark targeting multimodal long reasoning, (2) the SERG pipeline combining rule-based anchors with LLM refinement for large-scale CoT data generation, and (3) AHPO's dynamic integration of off-policy SFT and on-policy RL to address reward sparsity and catastrophic forgetting.

**2. Quality:** The technical execution is strong, with SERG demonstrating efficiency and quality, rigorous AHPO formulation (Section 3.3), and comprehensive experiments showing substantial improvements on MM-HELIX and generalization to math/logic benchmarks (Table 2). The ablations (Figures 6-7) effectively validate the design choices.

**3. Clarity:** The paper is well-structured with clear problem formulation and effective visualizations (Figures 1-3) that illustrate the benchmark design and system architecture. Additionally, the motivation for training methods that elicit extended reasoning from open-source MLLMs is well established.

**Weaknesses:**

**1. Benchmark Synthetic Nature / Coverage:** Although MM-HELIX procedurally generates tasks and spans many categories, all tasks appear synthetic and rule-based (see Figure 2 and example in Figure 3). There is no clear discussion or experiment detailing how well the benchmark reflects the “real-world” complexity or ambiguity of practical reasoning. Similarly, the tasks' diversity outside logic/math settings is not addressed. This may limit the external validity of the results.

**2. Limited Evaluation Metrics:** The evaluation in Table 1 and the Appendix relies almost exclusively on raw accuracy, with limited discussion of more nuanced aspects such as error typologies or qualitative analysis beyond completion rates. Given that the benchmark aims to probe "reflective" capabilities—a term used throughout the paper—a deeper analysis of response length, backtracking frequency, and self-correction behaviors would strengthen the evaluation and provide insights into how models engage in iterative reasoning.

**3. Equation Clarity and Notational Issues:** Although the main equations in Section 3.3 are written out, some aspects (but not limited to) lack precision for reproducibility:
Equation defining $N$ could be more clearly defined as number of roll-outs, and the use of $\operatorname{CLIP}$ could be further detailed.
The loss unification assumes normalization factor $Z$, but this is undefined and never discussed again.
The citation in line 305 should be added.
The off-policy and on-policy learning is uncleared whether they are applied to the same training data or different data point.

**4. Ablations Analysis Limited:**
There is also no ablation analysis on hyperparameters introduced in Equation 5, despite their potential importance.

**Questions:**

1. How are the difficulty levels in the benchmark determined? Is there a principled methodology or validation process?

2. Performance Disparities Across Domains: Open-source models perform substantially worse on Games and Puzzles compared to Algorithms and Paths (Table 1 and Appendix). What are the failure patterns underlying these disparities? Are they primarily due to reasoning capability limitations, context length constraints, instruction-following difficulties, or other factors? A more fine-grained error analysis could provide valuable insights.

3. In SERG, the initial step provides logically sound trajectories. However, beyond oracle trajectories that directly reach the correct solution, could trajectories that incorporate rethinking, reflection, or error handling be more valuable for developing the model's generalizable reasoning capabilities? At the second step, what specific prompts are used to transform rule-based trajectories into more natural, comprehensive reasoning processes with reflective steps? How is diversity in reasoning behaviors (mentioned above) controlled/encouraged during this refinement stage?

4. Are on-policy and off-policy learning applied to the same data points or separate sets? How is the number of samples per training step determined? Are there ablation studies on key hyperparameters, such as the ratio of on-policy to off-policy samples or the success rate threshold?

---

> ### Author Response · Authors · 2025-11-21
> **Official Comment by Authors for Reviewer EG1J (1/6)**
>
> > ***Weakness 1: Benchmark Synthetic Nature / Coverage: Although  MM-HELIX procedurally generates tasks and spans many categories, all  tasks appear synthetic and rule-based (see Figure 2 and example in  Figure 3). There is no clear discussion or experiment detailing how well  the benchmark reflects the “real-world” complexity or ambiguity of  practical reasoning. Similarly, the tasks' diversity outside logic/math  settings is not addressed. This may limit the external validity of the  results.***
>
> Thank you for raising this highly relevant question.
>
> **Real-world Relevance**
>
> Our design philosophy for MM-HELIX was explicitly guided by the goal of cultivating core cognitive capabilities that underpin real-world problem-solving in two aspects:
>
> 1. From the perspective of the problem-solving process, the synthetic nature of the tasks is to systematically enhance the MLLM's long-chain reasoning ability, particularly focusing on the reflective and backtracking thought processes. This is precisely the sophisticated solution methodology employed by humans when facing complex, multi-step tasks in practice—involving hypothesis testing, error correction, and iterative refinement.
> 2. From a task type perspective, the four task types in MM-HELIX are closely related to the reasoning needs in real-world scenarios. Graph tasks reflect everyday route planning and supply chain decisions. Game tasks simulate resource allocation and strategic decision-making. Algorithm tasks are similar to scheduling and multi-step conditional logic. Puzzle tasks encompass trial and error, hypothesis testing and visual cues, and overall planning. Therefore, we believe that MM-HELIX truly reflects the model's essential capabilities in real life and practical reasoning.
>
> **Clarification on Generalization and Task Diversity**
>
> We would like to clarify that MM-HELIX was not specifically constructed for math/logic, and its content has minimal overlap with such traditional datasets. The evaluation of our method on math and logic benchmarks (like MathVision/LogicVista) is therefore viewed as a zero-shot generalization test. The positive results strongly validate that our data and algorithm impart highly generalizable reasoning capabilities.
>
> Regarding task diversity outside of reasoning/logic, our focus is deliberately placed on tasks that benefit from reflective thinking. As established in prior research[1-2], reflection mechanisms offer negligible enhancement for low-level tasks such as perception, OCR, or simple information retrieval. Our benchmark is thus specifically designed to cultivate core skills crucial for real-world complex analysis and decision-making, which are the critical bottlenecks in current MLLMs.
>
> As demonstrated by the zero-shot output comparison on LogicVista (sample id: v1_10, shown below), our model, after training on the synthetic reflection data, successfully generalizes the reflective mechanism to out-of-domain practical reasoning tasks, exhibiting correct self-correction that the base model fails to achieve.
>
> *Response of base model Qwen2.5-VL-7B (wrong):*
> ```
> To determine which option should replace the question mark, let's analyze the pattern in the sequence of images above the separating line.
> ......
> - A: A tall rectangle with a smaller rectangle removed from the bottom right corner.
> - B: A tall rectangle with a smaller rectangle removed from the top left corner.
> ......
> - E: A tall rectangle with a smaller rectangle removed from the bottom right corner.
> The correct option that matches the pattern is B.
> ```
>
> *Response after AHPO training: (correct):*
> ```
> Let’s take a closer look at this puzzle. We’re given a sequence of shapes above a line, and we need to find which of the five options below it (A through E) should come next. The goal is to identify how the pattern progresses.
>
> First, let’s examine the shapes in the top row:
>
> ...
>
> Looking more carefully, each cell seems to have a specific configuration. For example:
> - In the first image: a small square in the top-left, and a diagonal line in the bottom-right.
> - Second: a vertical bar in the top-left, and a small square in the bottom-right.
> ...
> - Fifth: a tall vertical bar in the top-left, and a diagonal line in the bottom-right.
>
> Wait — actually, looking again, the first image has a small square in the top-left and a diagonal line in the bottom-right. But the second one has a vertical bar in the top-left and a small square in the bottom-right. So maybe the pattern isn’t about position, but about *what’s in each cell*?
>
> Alternatively, perhaps it's about the **orientation** or **combination** of elements.
>
> Let’s try to map this out more systematically.
>
> ....
>
> So the shape should be: vertical bar in top-left, small square in bottom-right.
>
> Final Answer: \boxed{E}
> ```
> [1] Internvl3. 5: Advancing open-source multimodal models in versatility, reasoning, and efficiency
>
> [2] Qwen3-VL

---

> ### Author Response · Authors · 2025-11-21
> **Official Comment by Authors for Reviewer EG1J (2/6)**
>
> > ***Weakness 2: Limited Evaluation Metrics: The evaluation in  Table 1 and the Appendix relies almost exclusively on raw accuracy, with  limited discussion of more nuanced aspects such as error typologies or  qualitative analysis beyond completion rates. Given that the benchmark  aims to probe "reflective" capabilities—a term used throughout the  paper—a deeper analysis of response length, backtracking frequency, and  self-correction behaviors would strengthen the evaluation and provide  insights into how models engage in iterative reasoning.***
>
> We thank the reviewer for this insightful suggestion. We have expanded our evaluation to include a detailed qualitative error typology and a quantitative analysis of reasoning behaviors (response length, backtracking, and self-correction).
>
> **Error Analysis**
>
> We have conducted a qualitative examination of the model's incorrect outputs and categorized the primary failure modes into four distinct classes:
>
> 1. Perception Errors:
>
> This occurs when the model fails to correctly interpret the raw visual state.
>
> *Example*: In Maze task, the model fails to identify the presence of a wall at a specific coordinate, leading to invalid path planning proposals that traverse obstacles.
>
> 2. Hallucination (Reasoning-Observation Mismatch):
>
> Here, the model generates conclusions that contradict its own intermediate observations, effectively forcing a result to match a prior expectation rather than the evidence.
>
> *Example*: The model explicitly identifies a character sequence at the predicted coordinates as "L, N, O, I" (Reasoning trace: “So letters are L, N, O, I... that's LION!”). Crucially, the model ignores the clear mismatch between the observed sequence ("LNOI") and the target word ("LION"), hallucinating a successful anagram solution despite explicit contradictory evidence.
>
> 3. Insufficient Backtracking:
>
> When facing conflicting constraints, the model tends to attempt local fixes rather than performing valid backtracking to reset earlier variables. This often results in the model settling for an invalid approximate solution.
>
> *Example*: In 5×5 Kukurasu Puzzle, The model successfully satisfied all row sum constraints but failed the column sum constraints (e.g., the predicted Column 1 sum was 9, whereas the target was 10). Although the model explicitly identified the conflict during final verification (“Required 10. Close, but not enough”), it failed to backtrack and revise the earlier row-level decisions that caused this downstream conflict.
>
> 4. Instruction Following Failures(Minor):
>
> The model occasionally violates explicit hard constraints, particularly negative constraints provided in the system prompt.
>
> *Example*: In Word Ladder Task, The prompt explicitly forbids adding or deleting letters (only substitution is allowed). The model violated this constraint by transitioning from "cappy" (5 letters) to "clappy" (6 letters), disregarding the length invariance requirement.
>
> **Quantitative Analysis of Reflective Behaviors**
>
> We conducted a comparative analysis of our model, MM-HELIX-7B-Thinking, against several others: Intern-S1-241B-A28B, Kimi-VL-16B-A3B-Thinking-2506, GLM-4.1V-9B-Thinking, and MiniCPM-V-4.5-8B. For these models, we calculated the average output word count, and we utilized Gemini-2.5-Flash to extract and quantify their backtracking frequency and the number of self-correction instances. The results are summarized below:
>
> | Model | Intern-S1-241B-A28B | MM-HELIX-7B-Thinking (ours) | Kimi-VL-16B-A3B-Thinking-2506 | GLM-4.1V-9B-Thinking | MiniCPM-V-4.5-8B |
> | --- | --- | --- | --- | --- | --- |
> | avg_word_count | 5413.77 | 1484.7 | 3788.2 | 4230.9 | 6011.57 |
> | backtracking | 61.55 | 14.91 | 34 | 34.09 | 67.94 |
> | self_correction | 6.97 | 1.31 | 7.1 | 4.85 | 5.49 |
> | score | 33.3 | 24.9 | 19.3 | 16.3 | 10.4 |
>
> The results indicate that *there is no strong correlation between a model's score and its backtracking frequency or the number of self-corrections*. Models with the highest backtracking frequency (MiniCPM-V and Intern-S1) occupy the extreme ends of the performance spectrum.
>
> MiniCPM-V and, to some extent, Kimi-VL and GLM, exhibit high frequency without corresponding high performance, indicating the reflection is often redundant, ill-informed, or fails to lead to meaningful recovery from errors. Powerful models that leverage reflection well (like Intern-S1) may use a higher frequency of reflection, but it is executed strategically and accurately, contributing directly to a higher outcome.
>
> Compared to weaker models, MM-HELIX-7B-Thinking engages in fewer unproductive reflections and makes fewer errors in its chain-of-thought, thus requiring significantly less self-correction. This quantitative evidence strongly suggests that it is the quality and efficacy of the reflective step rather than its frequency that determines the final reasoning performance.

---

> ### Author Response · Authors · 2025-11-21
> **Official Comment by Authors for Reviewer EG1J (3/6)**
>
> > ***Weakness 3: Equation Clarity and Notational Issues: Although  the main equations in Section 3.3 are written out, some aspects (but not  limited to) lack precision for reproducibility: Equation defining N could be more clearly defined as number of roll-outs, and the use of CLIP could be further detailed. The loss unification assumes normalization factor Z, but this is undefined and never discussed again. The citation in line 305 should be added. The off-policy and on-policy learning is uncleared whether they are applied to the same training data or different data point.***
>
> Thank you for this highly rigorous and essential review of our notational precision and methodological clarity in Section 3.3. We apologize for the initial ambiguity and fully agree that the formal definitions must be precise for reproducibility.
> We will immediately revise the manuscript based on your comprehensive suggestions:
> 1. Definition of N (Number of Roll-outs):
> We will explicitly define $N$ as the number of roll-out trajectories collected from the current policy. In our experiments, we set the empirical value of $N=5$.
> 2. Clarification Regarding CLIP:
> We appreciate you pointing out the inclusion of the CLIP term. Due to the significant distribution shift between the on-policy data and the expert off-policy data, we found that applying the restrictive CLIP factor to the off-policy ratio was detrimental to effective learning. To prioritize learning from the high-quality expert demonstrations, we removed the CLIP mechanism from the off-policy loss term in the final implementation.
> 3. Normalization Factor Z:
> Our implemented loss function utilizes a sample-level averaging scheme without token-level weights. Consequently, the normalization factor $Z$ effectively simplifies to $N$.
> 4. Citation in Line 305:
> We apologize for the missing citation. The correct reference is indeed to the DAPO paper. We will rectify this omission.
> 5. On-Policy and Off-Policy Data Points:
> We first want to clarify that MMK12 is a math-specific dataset primarily used for Reinforcement Learning. It contains only Question and Ground Truth answers, lacking the Chain-of-Thought trajectories necessary for our AHPO method's reflection mechanism. Thus, training AHPO on MMK12 functionally aligns with the GRPO baseline, which means pure on-policy; only training on MM-HELIX dataset has dynamic off-policy data involved. This separate application is the core mechanism enabling the generalization of reflection capability from MM-HELIX-100K to the target MMK12 domain.
>
> We are committed to making these precise corrections throughout the revised manuscript. Thanks again for your suggestions!
>
> > ***Weakness 4: Ablations Analysis Limited: There is also no ablation analysis on hyperparameters introduced in Equation 5, despite their potential importance.***
>
> Thank you for this excellent suggestion.
> We conducted an ablation study focusing on the impact of different threshold values for $\hat{R}$ while holding the other parameters constant: the roll-out batch size was set to $N=5$, and the training was conducted for 500 AHPO steps on 15k training samples. The results below report the performance:
>
> | Threshold| Algorithm | Graph| Game | Puzzle | Overall |
> |--------|--------|------------|-------|---------|-----------|
> | 0 | 13.0 | 15.0 | 4.4 | 5.4 | 8.7 |
> | 1 | 21.1 | 17.9 | 20.9 | 18.8 | 20.4 |
> | 2 | 23.7 | 18.8 | 12.6 | 20.1 | 20.1 |
> | 3 | 15.1 | 16.2 | 4.4| 6.5 | 8.9 |
> | 4 | 5.6 | 15.8 | 4.4 | 6.8 | 7.9 |
> | 5 | 11.5 | 18.8 | 3.3 | 4.4 | 8.5 |
>
> We have several observations:
> 1. When $\hat{R}=0$, the performance is significantly lower. This confirms that the off-policy data is essential for effective learning, particularly when the underlying policy struggles with sparse rewards in the pure RL baseline.
> 2. Setting the threshold too high($\hat{R}>=3$) leads to a sharp drop in performance. At high $\hat{R}$ values, the dynamic coefficient is rarely activated, which imposes the off-policy expert loss on the policy only when it is already performing extremely well. This effectively forces the model to over-regularize towards the expert data, suppressing necessary exploration and resulting in training instability (manifested as repetitive or degenerate outputs).
> 3. The thresholds $\hat{R}=1$ and $\hat{R}=2$ yield the most robust and highest performance, indicating that a moderate level of on-policy success is required before the expert guidance is dynamically engaged.
>
>  Thank you again for suggesting this important dimension of analysis.

---

> ### Author Response · Authors · 2025-11-21
> **Official Comment by Authors for Reviewer EG1J (4/6)**
>
> > ***Question 1: How are the difficulty levels in the benchmark determined? Is there a principled methodology or validation process?***
>
> Thank you for your suggestions. We would like to provide detailed clarification on this methodological aspect below:
>
> The classification of difficulty levels is not performed manually on a per-sample basis, but is systematically controlled by the specific hyperparameters inherent to the procedural generation of each task category. We manually defined the range of these parameters, after which the task instances are automatically generated and assigned to a difficulty level based on satisfying these pre-defined parameter thresholds. The details are shown below:
>
> For Algorithm tasks, the main factors influencing the difficulty are the length of the input array, the size of the values, and the number of steps related to the overall algorithm complexity.
>
> - For example, in the rainwater collection task in this paper, as the difficulty increases, the input array length ranges from (5, 10), (10, 15), (15, 20), (20, 25) to (25, 30), and the value ranges from (5, 10), (10, 20), (20, 30), (30, 40) to (40, 50). The number of solution steps also increases accordingly to ensure the reasonableness of the difficulty grading.
>
> For Graph theory tasks, the primary criterion for determining difficulty is the scale of the graph, specifically defined by the number of nodes (N).
>
> - For example, for the Hamiltonian Cycle task, we established five distinct difficulty levels corresponding to the graph size: Level 1 (N=6), Level 2 (N=9), Level 3 (N=12), Level 4 (N=14), and Level 5 (N=16).
>
> For Puzzle and Game tasks, the difficulty level (1–5) for each puzzle and game tasks is determined by the joint contribution of two standardized quantitative dimensions: Grid Size and Solution Complexity. The Grid Size controls the magnitude of the state space, while the Solution Complexity dictates the depth of reasoning required, which is controlled by enforcing a minimum number of optimal steps or by adjusting the density of constraints.
>
> While Grid Size expands the state space, it alone does not guarantee difficulty, a large maze could still have a trivial, straight-line solution. Therefore, Solution Complexity here serves as a critical second dimension to enforce the depth of reasoning. This ensures that tasks with large grids also require a correspondingly complex chain of logic (e.g., more steps, tighter constraints) to be solved, preventing "hollow" high-level tasks.
>
> - Taking the Sokoban puzzle as a example, the grid dimensions range from 5×7 (Level 1), 7×9 (Level 2), 7×9 (Level 3), 11×13 (Level 4) to 11×13 (Level 5). The required optimal solution length ranges from [1, 15] (Level 1), (15, 25] (Level 2), (25, 40] (Level 3), (40, 50] (Level 4) to (50, 60] (Level 5), and the number of boxes to move ranges from 1 (Level 1–2), 2 (Level 3–4) to 3 (Level 5).
>
> By coupling these two dimensions, we ensure that higher difficulty levels consistently represent a significant increase in both the computational search space and the required reasoning depth.
>
> > ***Question 2: Performance Disparities Across Domains: Open-source models  perform substantially worse on Games and Puzzles compared to Algorithms  and Paths (Table 1 and Appendix). What are the failure patterns  underlying these disparities?***
>
> We appreciate the reviewer’s insightful observation regarding the performance gap between the Games & Puzzles and Algorithms & Paths domains.
>
> We attribute the performance gap between domains to the dynamic nature of the tasks. Algorithms and Paths problems often present static states that align well with code-heavy training corpora, which models are adept at processing. Conversely, Games and Puzzles are highly dynamic: the state evolves after every action. This necessitates rigorous chain-of-thought reasoning and state tracking, which remains a significant challenge for current open-source models compared to their proprietary counterparts.

---

> ### Author Response · Authors · 2025-11-21
> **Official Comment by Authors for Reviewer EG1J (5/6)**
>
> > ***Question 3: In SERG, the initial step provides logically sound trajectories.  However, beyond oracle trajectories that directly reach the correct  solution, could trajectories that incorporate rethinking, reflection, or  error handling be more valuable for developing the model's  generalizable reasoning capabilities? At the second step, what specific  prompts are used to transform rule-based trajectories into more natural,  comprehensive reasoning processes with reflective steps? How is  diversity in reasoning behaviors (mentioned above) controlled/encouraged  during this refinement stage?***
>
> Many thanks to the reviewers for the insightful observation.
>
> **Rethinking v.s. Direct solution**
>
> Recent studies consistently show that learning from reflective trajectories outperforms training solely on perfect oracle paths:
>
> 1. Error correction and robustness.
> Training only on correct trajectories leads to exposure bias: once the model deviates from the ideal path at inference time, it cannot recover. Recent work such as SCoRe [1] demonstrates that incorporating multi-round self-reflection and self-correction into training significantly improves reasoning robustness compared with SFT on oracle demonstrations. Similar conclusions also appear in Reflexion [2], where explicit reflective feedback enables models to detect and recover from mistakes in sequential decision tasks.
>
> 2. Iterative refinement improves reasoning depth.
> Methods such as Self-Refine [5] and SELF-RAG [6] show that training on trajectories that include proposal, critique, refinement consistently yields better accuracy than learning from single-pass correct outputs.
>
> These findings support this conclusion: trajectories that include rethinking, verification, and error correction contain richer supervision than oracle-only traces.
>
> [1] Training Language Models to Self-Correct via Reinforcement Learning, ICLR 2024.
>
> [2] Reflexion: Language Agents with Verbal Reinforcement Learning, 2023.
>
> [3] Quiet-STaR: Language Models Can Teach Themselves to Think Before Speaking, 2024.
>
> [4] rStar-Math: Small LLMs Can Master Math Reasoning with Self-Evolved Deep Thinking, arXiv 2025.
>
> **Prompt for Refinement in SERG**
>
> The reflection and rethinking behaviors are already embedded in the rule-based stage itself. Our rule-based CoT generator are task-dependent: every task (e.g., Sokoban, Nibbles, Calcudoku, Graph Isomorphism) has its own multi-stage reasoning program that includes explicit backtracking, branch elimination, feasibility checks, and self-correction loops. These behaviors naturally arise from the underlying algorithmic solvers. Within the same task, the rule-based generator also employs multiple heuristic strategies. These heuristics inherently include reflection patterns.
>
> Thus, the reflective reasoning is not fabricated but emerges directly from the solver’s rule-based decision logic.
> In the SERG rewriting stage, the rule-based CoT acts as a scaffold that ensures correctness, while the LLM only rewrites it into more fluent, human-like language. To do this, we use the following prompt:
> ```
>  ## Your Task
>
>  You will be given a problem along with an automatically generated reasoning process. The reasoning process is logically correct, but it is too rigid and mechanical, which makes it less suitable for training purposes. Your task is to rewrite and enhance the reasoning process by introducing natural exploration, thoughtful consideration, and a more human-like flow of ideas, while still arriving at the correct solution. Please put the puzzle's final answer in `<answer></answer>` tags at the end.
>
>  ## Input
>
>  ### Original Problem Statement:
>  [problem_content_placeholder]
>
>  ### Current Reasoning Process:
>  [reasoning_content_placeholder]
>
>  ## Output Requirements
>
>  Please rewrite the reasoning process to be more natural and exploratory, without revealing the final answer until you derive it through your own reasoning.
> ```

---

> ### Author Response · Authors · 2025-11-21
> **Official Comment by Authors for Reviewer EG1J (6/6)**
>
> > ***Question 4: Are on-policy and off-policy learning applied to the same data  points or separate sets? How is the number of samples per training step  determined?***
>
> Thank you for this highly pertinent set of questions concerning our training pipeline's data utilization and hyperparameter sensitivity. We appreciate the opportunity to provide precise clarification.
>
> 1. On-Policy and Off-Policy Data Points
>
> We would like to clarify that during training, the MMK12 dataset serves as the on-policy data, and our MM-HELIX serves as both on-policy and off-policy data.
>
> MMK12 is a math-specific dataset that provides the target domain for the model during the Reinforcement Learning phase. However, as previously clarified, MMK12 strictly contains only Question and Ground Truth Answer pairs; it does not contain the Chain-of-Thought (CoT) trajectories or reflective steps required for the core mechanism of our AHPO approach.
> Therefore, the AHPO training conducted solely on MMK12 is functionally equivalent to the GRPO baseline. Our primary experimental objective for including it was to empirically validate that AHPO can effectively generalize the reflective capabilities learned from the non-mathematical MM-HELIX expert data to the mathematical reasoning domain where no domain-specific reflection data was provided.
>
> 2. Sample Determination per Training Step
>
> The batch size per training step was determined primarily by GPU memory capacity and training cost feasibility. Based on these constraints, we set the total batch size to 128 samples. This value strikes an empirical balance between gradient stability and computational efficiency.
>
> We will ensure these detailed clarifications are integrated into the revised manuscript.

---

### Official Review · Reviewer_onUL · 2025-10-31

**Soundness:** 3
**Presentation:** 3
**Contribution:** 3
**Rating:** 8
**Confidence:** 3

**Summary:**

The paper introduces MM-HELIX, a multimodal benchmark of 42 tasks (algorithms, graphs, puzzles, games) with 1,260 procedurally generated instances designed to test long-chain reflective reasoning in MLLMs. It further proposes a Step-Elicited Response Generation (SERG) pipeline to build MM-HELIX-100K, a dataset of ~100k verified reasoning traces, and a training algorithm Adaptive Hybrid Policy Optimization (AHPO) that adaptively mixes off-policy supervision with on-policy exploration via a reward-gated coefficient. On MM-HELIX, state-of-the-art models underperform (e.g., strong open-source models <35% acc.; large proprietary models ≤58.1% multimodal), while AHPO on Qwen2.5-VL-7B improves accuracy by +18.6 points and yields +5.7 points average gains on external math/logic benchmarks. The work argues that reflective reasoning in MLLMs is learnable and transferable.

**Strengths:**

- Well-posed evaluation target. MM-HELIX isolates long-chain reflective reasoning with verifiable tasks, hierarchical difficulty, and balanced sampling (1,260 instances across 42 tasks), enabling controlled diagnosis
- SERG uses rule-based scaffolds refined by a strong LLM, then filtered by verifiers to produce 100k reflective traces with far lower time/cost than naive rollouts and with better downstream utility
- AHPO’s reward-gated mixing of off-policy expert loss and on-policy GRPO addresses reward sparsity and SFT-forgetting. The authors show the learning curves of AHPO and demonstrate consistent gains against different baseline algorithms (GRPO/LUFFY/SFT)

**Weaknesses:**

- Equation (6) describes ξ’s activation by a success-rate threshold \hat{R}, but the exact estimator (win-dow size, grouping), threshold choice, and sensitivity are not fully ablated. Providing an ablation over \hat{R} and these different parameters would be helpful to get a deeper understanding of the algorithm.

**Questions:**

- I'd love to see some abaltion on the success rate threshold and how would that impact the perfomrance
- For generation pipeline, Did you evaluate sample efficiency when using fewer than 22k traces? Is there a particular reason for the "22k" number? Would be interesting to see the scaling behavior of the generated samples (i.e., if 5k SERG vs. 5k rule-based, how would the performance comparison look like).

---

> ### Author Response · Authors · 2025-11-21
> **Official Comment by Authors for Reviewer onUL**
>
> We sincerely appreciate your constructive feedback and valuable suggestions.
>
> > ***Weakness 1 & Question 1: Equation (6) describes ξ’s activation by a success-rate threshold \hat{R}, but the exact estimator (win-dow size, grouping), threshold choice, and sensitivity are not fully ablated. Providing an ablation over \hat{R} and these different parameters would be helpful to get a deeper understanding of the algorithm. I'd love to see some abaltion on the success rate threshold and how would that impact the perfomrance.***
>
> Thank you for this excellent suggestion.
>
> We conducted an ablation study focusing on the impact of different threshold values for $\hat{R}$ while holding the other parameters constant: the roll-out batch size was set to $N=5$, and the training was conducted for 500 AHPO steps on 15k training samples. The results below report the performance:
>
> | Threshold| Algorithm | Graph| Game | Puzzle | Overall |
> |--------|--------|------------|-------|---------|-----------|
> | 0 | 13.0 | 15.0 | 4.4 | 5.4 | 8.7 |
> | 1 | 21.1 | 17.9 | 20.9 | 18.8 | 20.4 |
> | 2 | 23.7 | 18.8 | 12.6 | 20.1 | 20.1 |
> | 3 | 15.1 | 16.2 | 4.4| 6.5 | 8.9 |
> | 4 | 5.6 | 15.8 | 4.4 | 6.8 | 7.9 |
> | 5 | 11.5 | 18.8 | 3.3 | 4.4 | 8.5 |
>
> We have several observations:
> 1. When $\hat{R}=0$, the performance is significantly lower. This confirms that the off-policy data is essential for effective learning, particularly when the underlying policy struggles with sparse rewards in the pure RL baseline.
> 2. Setting the threshold too high($\hat{R}>=3$) leads to a sharp drop in performance. At high $\hat{R}$ values, the dynamic coefficient is rarely activated, which imposes the off-policy expert loss on the policy only when it is already performing extremely well. This effectively forces the model to over-regularize towards the expert data, suppressing necessary exploration and resulting in training instability (manifested as repetitive or degenerate outputs).
> 3. The thresholds $\hat{R}=1$ and $\hat{R}=2$ yield the most robust and highest performance, indicating that a moderate level of on-policy success is required before the expert guidance is dynamically engaged.
>
> Thank you again for suggesting this important dimension of analysis.
>
> > ***Question 2: For generation pipeline, Did you evaluate sample efficiency when  using fewer than 22k traces? Is there a particular reason for the "22k"  number? Would be interesting to see the scaling behavior of the  generated samples (i.e., if 5k SERG vs. 5k rule-based, how would the  performance comparison look like).***
>
> Thank you for your pertinent questions.
>
> **1. Rationale for 22k training data**
>
> The decision to utilize 22k samples was driven by both computational resource constraints and strategic data prioritization:
> - The primary limiting factor was the disproportionate and computationally prohibitive cost for the Reinforcement Learning stage when handling excessively long Chain-of-Thought trajectories. Consequently, we implemented a practical constraint, filtering the MM-HELIX dataset to include only those samples whose combined input-output sequence length was less than 8192 tokens. This constraint was critical for maintaining a feasible training timeline and cost, particularly during the iterative RL optimization phase.
> - Beyond the length constraint, the final 22k set was also curated based on a principled data balancing strategy. We adjusted the proportional representation of different task categories to align with the observed baseline learning difficulty of the initial model. Tasks that were demonstrably more challenging for the base model were allocated a slightly higher proportion.
>
> **2.  Scaling Behavior(Rule-based vs. SERG)**
>
> We also conduct an ablation study to explore the scaling behavior of two generation method, as shown below.
>
> | Method/Data Scale| 5k | 13k | 22k|
> |--------|--------|------------|------------|
> | Rule-based| 15.8 | 16.7 | 18.9 |
> | SERG | 12.4 | 19.4 | 23.8 |
>
> At the minimal scale, the Rule-based method initially outperforms SERG. We observe that under 5k data, model trained with SERG has a noticeable frequency of repetitive responses. We hypothesize this is because the limited 5k samples are insufficient for the model to fully internalize the complex, long-chain reflective thinking mode generated by SERG.
>
> However, when the data scale is adequate, SERG significantly surpasses the Rule-based method. This finding strongly indicates that while the reflective CoT generated by SERG requires a higher minimum volume of data to stabilize the training process, it offers a substantially higher performance ceiling due to the intrinsic quality and sophistication of the reflective thought processes.
>
> Thank you very much for your insightful questions!

---

### Official Review · Reviewer_cRVP · 2025-11-01

**Soundness:** 3
**Presentation:** 4
**Contribution:** 4
**Rating:** 4
**Confidence:** 4

**Summary:**

This paper addresses the weakness of MLLMs in long-chain reflective reasoning. It introduces MM-HELIX, a benchmark of 42 complex multimodal tasks and 1260 samples across difficulty levels. The authors also build MM-HELIX-100K, a dataset of 100k verified reflective reasoning traces generated via the SERG pipeline. They also propose AHPO, a training algorithm that adaptively balances expert supervision and reinforcement learning to improve reasoning stability and exploration.

**Strengths:**

1. The paper presents a substantial amount of work, including both a benchmark and a dataset, as well as a proposed improvement method. It makes a meaningful contribution to research on MLLM reasoning tasks.
2. Based on the proposed dataset and methodology, the paper achieves significant improvements across various visual reasoning benchmarks. The proposed AHPO effectively alleviates the issue of insufficient high-quality rollouts when the model faces challenging tasks. Compared to previous hybrid method LUFFY, AHPO demonstrates a considerable performance gain.

**Weaknesses:**

My main concern is that the paper lacks many important implementation details, including those related to the benchmark, dataset, and ablation studies of the proposed method. These missing parts may confuse readers and make the work difficult to reproduce.

Benchmark:
1. What are the text and image settings during evaluation, and how are image-based tasks converted into textual form?
2. What are the criteria for difficulty categorization? The paper only briefly mentions this. Is the classification done manually?
3. What are the implementation details of the automated verifier?

Dataset:
1. The paper lacks detailed information on data construction, including prompts and heuristic rules.
2. There is no statistical breakdown of the proportions of different task types within the dataset.

Experiments and Analysis:
1. In Section A.8.2, the improvement on Graph-type tasks is smaller than on other types, but there is no accompanying analysis.
2. The base model used for training should be explicitly stated in the experimental setup, even if readers can infer it from Table 2.
3. In Figures 6 and 7, why is only static-AHPO compared with other algorithms, while Figure 7 also includes a non-static version(AHPO). Does static-AHPO mean the activation coefficient = 0?
4. In Appendix A.2 (Training Setting), why are only 22k samples used for SFT instead of the full dataset? Is there any comparison showing the relationship between data volume and performance, and how were these 22k samples selected?
5. Why is math data included during the RL stage? Was the AHPO model in Table 2 also trained with math data? If so, do benchmarks such as MathVision and MathVerse fall into the in-domain category? If that’s the case, does the claimed generalization ability still hold? Also, is there any comparison between models trained with and without math data?
6. [Minor] The paper lacks ablation studies on different success rate thresholds and activation coefficient.
7. [Minor] Is there any comparison showing how the CoT length in MM-HELIX-100K affects reasoning performance?

Typo:
Line 305–306: “following()”

Others:
[Minor] I suggest that the authors move Sections A.5–A.7 to the end of the appendix, so that readers don’t have to scroll extensively or risk overlooking the subsequent content. Alternatively, at the very least, an appendix table of contents should be added.

If the authors address most of my concerns, I would be willing to raise my score.

**Questions:**

See weaknesses

---

> ### Author Response · Authors · 2025-11-21
> **Official Comment by Authors for Reviewer cRVP (1/7)**
>
> > ***Weakness Benchmark 1: what are the text and image settings during evaluation, and how are image-based tasks converted into textual form?***
>
> We thank the reviewer for this question regarding our multimodal evaluation setup.
>
> **Evaluation Settings**:
>
> Each evaluation instance is composed of a textual task description and a problem representation. The problem representation is rendered in either a visual (image) or textual modality. Crucially, to ensure a fair comparison, the underlying content is identical across modalities; we first generate a unified "initial state" for each problem and then render it into the corresponding image or text format.
>
> **Textual Conversion of Image-based Tasks**:
>
> To represent image-based tasks in a textual format, we employ structured symbolic representations tailored to the specific domain:
>
> - Algorithmic Tasks: We utilize standard list serialization (e.g., [2, 4, 5, ...]) to represent data structures.
>
> - Graph Tasks: We employ adjacency lists to describe connectivity. For example, 0: [1, 7]; 1: [0, 2]... indicates that Node 0 is connected to Nodes 1 and 7, while Node 1 is connected to Nodes 0 and 2.
>
> - Puzzles and Games: We use 2D array (matrix) representations to denote the grid state. For instance, a Sudoku puzzle is serialized as a nested list: [[3, 0, 2, ...], [1, 7, 9, ...], ..., [9, 3, 8, ...]].
>
> > ***Weakness Benchmark 2: What are the criteria for difficulty categorization? The paper only briefly mentions this. Is the classification done manually?***
>
> Thank you for your suggestions. We would like to provide detailed clarification on this methodological aspect below:
>
> The classification of difficulty levels is not performed manually on a per-sample basis, but is systematically controlled by the specific hyperparameters inherent to the procedural generation of each task category. We manually defined the range of these parameters, after which the task instances are automatically generated and assigned to a difficulty level based on satisfying these pre-defined parameter thresholds. The details are shown below:
>
> **For Algorithm tasks,** the main factors influencing the difficulty are the length of the input array, the size of the values, and the number of steps related to the overall algorithm complexity.
>
> - For example, in the rainwater collection task in this paper, as the difficulty increases, the input array length ranges from (5, 10), (10, 15), (15, 20), (20, 25) to (25, 30), and the value ranges from (5, 10), (10, 20), (20, 30), (30, 40) to (40, 50). The number of solution steps also increases accordingly to ensure the reasonableness of the difficulty grading.
>
> **For Graph theory tasks,** the primary criterion for determining difficulty is the scale of the graph, specifically defined by the number of nodes (N).
>
> - For example, for the Hamiltonian Cycle task, we established five distinct difficulty levels corresponding to the graph size: Level 1 (N=6), Level 2 (N=9), Level 3 (N=12), Level 4 (N=14), and Level 5 (N=16).
>
> **For Puzzle and Game tasks**, the difficulty level (1–5) for each puzzle and game tasks is determined by the joint contribution of two standardized quantitative dimensions: Grid Size and Solution Complexity. The Grid Size controls the magnitude of the state space, while the Solution Complexity dictates the depth of reasoning required, which is controlled by enforcing a minimum number of optimal steps or by adjusting the density of constraints.
>
> While Grid Size expands the state space, it alone does not guarantee difficulty, a large maze could still have a trivial, straight-line solution. Therefore, Solution Complexity here serves as a critical second dimension to enforce the depth of reasoning. This ensures that tasks with large grids also require a correspondingly complex chain of logic (e.g., more steps, tighter constraints) to be solved, preventing "hollow" high-level tasks.
>
> - Taking the Sokoban puzzle as a example, the grid dimensions range from 5×7 (Level 1), 7×9 (Level 2), 7×9 (Level 3), 11×13 (Level 4) to 11×13 (Level 5). The required optimal solution length ranges from [1, 15] (Level 1), (15, 25] (Level 2), (25, 40] (Level 3), (40, 50] (Level 4) to (50, 60] (Level 5), and the number of boxes to move ranges from 1 (Level 1–2), 2 (Level 3–4) to 3 (Level 5).
>
> By coupling these two dimensions, we ensure that higher difficulty levels consistently represent a significant increase in both the computational search space and the required reasoning depth.

---

> ### Author Response · Authors · 2025-11-21
> **Official Comment by Authors for Reviewer cRVP (2/7)**
>
> > ***Weakness Benchmark 3: What are the implementation details of the automated verifier?***
>
> Thank you for this highly relevant and practical question.
>
> Our verifier is based on manually crafted, rule-based code tailored to the specific nature of each task category. The verification logic is functionally divided into two distinct approaches:
>
> 1. **Direct Answer Matching**: For task categories where the correct output is a unique, single-point answer (predominantly in the Algorithm and Graph domains), we first parse the final answer from model output. The extracted answer is then directly compared for exact equivalence with the corresponding Ground Truth (GT) answer.
> 2. **Procedural Validation**: For tasks with an inherent procedural or path-finding element where multiple correct solution sequences may exist (primarily in the Game and Puzzle domains), we need to validate whether the output satisfies the rules and targets given by the question. The verifier is initialized with the initial state and the full set of domain-specific rules for the given problem instance.  The verifier then programmatically simulates the execution of the proposed solution sequence (or verifies the final state) against the game rules, ensuring that the final output satisfies all requirements and reaches the target state from the initial state without violating any constraints.
>
> For example, in the Maze task, the verifier performs a step-by-step check to confirm that the path indicated by the model follows the Game Rules and successfully connects the Start (S) to the End (E) cells, as initial states, rules and output shown below:
> ```
> Initial state:
> +-+-+-+
> |0 0 S|
> + +-+ +
> |0|0 0|
> + + +-+
> |0|0 E|
> +-+-+-+
>
> Game Rules:
> 1. The maze consists of a grid of cells
> 2. Walls are represented by the '+', '-', and '|' characters between cells, not as cells themselves
> 3. You can move horizontally or vertically between adjacent cells if there is no wall between them
> 4. You can only move through one cell at a time in any direction
> 5. The goal is to find a path from the start cell (S) to the end cell (E)
>
> Output Answer:
> +-+-+-+
> |0 0 S|
> + +-+ +
> |0|↓ ←|
> + + +-+
> |0|→ E|
> +-+-+-+
> ```

---

> ### Author Response · Authors · 2025-11-21
> **Official Comment by Authors for Reviewer cRVP (3/7)**
>
> > ***Weakness Dataset 1: The paper lacks detailed information on data construction, including prompts and heuristic rules.***
>
> Our data construction pipeline consists of two sequential stages:
>
> **1. Problem generation with deterministic rule-based reasoning.**
>
> For each tasks (e.g., Eulero, Bridges, Campsite, Snake), we instantiate a task-specific generator that creates tasks instances, generates and validate the solution, and produces a structured four-step rule-based CoT trace. These steps always cover (a) game rule comprehension, (b) careful parsing of the initial state, (c) systematic reasoning using puzzle-specific heuristics (such as constraint propagation, forced moves, connectivity analysis, or adjacency checks), and (d) final verification. Every stage is logged in natural language, ensuring the resulting CoT is logically sound and correct.
>
> To make the rule-based heuristics concrete, we take the following pseudo-code of the rule-based cot generator for sudoku as an example:
> ```
> function generate_cot_sudoku(puzzle):
>     # Step 1: Rules & Setup
>     extract initial 9x9 grid and pre-filled numbers
>     describe global rules (unique 1-9 in every row, column, and 3x3 subgrid)
>
>     # Step 2: Initial State
>     render the grid, distinguishing between initial clues and empty cells
>
>     # Step 3: Heuristic Reasoning
>     calculate candidate set for all empty cells
>
>     loop while empty cells exist:
>         progress_made = False
>
>         # Heuristic A: Naked Singles
>         for each cell in empty_cells:
>             if size(candidates[cell]) == 1:
>                 value = candidates[cell][0]
>                 fill cell with value
>                 log f"Cell {cell} can only be {value} (Naked Single)"
>                 update candidates for peers (row, col, subgrid)
>                 progress_made = True
>
>         # Heuristic B: Hidden Singles
>         if not progress_made:
>             for unit in (all_rows, all_cols, all_subgrids):
>                 for digit in 1..9:
>                     possible_positions = find_cells_in_unit_allowing_digit(unit, digit)
>                     if size(possible_positions) == 1:
>                         cell = possible_positions[0]
>                         fill cell with digit
>                         log f"In {unit}, digit {digit} can only go in {cell} (Hidden Single)"
>                         update candidates
>                         progress_made = True
>
>         # Heuristic C: Randomized Search / Backtracking (if stuck)
>         if not progress_made and not is_solved():
>             select cell with minimum candidates (heuristically best pivot)
>             randomly select one candidate value (to simulate trial)
>             log f"No forced moves. Trying {value} at {cell}..."
>             recurse_solve(grid_state)
>
>     # Step 4: Verification
>     check all rows, columns, and subgrids sum to 45 and contain unique digits
>     return the rule-based CoT
> ```
>
> **2. SERG rewriting for human-like reasoning.**
>
> The deterministic CoT is then rewritten by advanced LLM using the following prompt:
>  ```
>  ## Your Task
>
>  You will be given a problem along with an automatically generated reasoning process. The reasoning process is logically correct, but it is too rigid and mechanical, which makes it less suitable for training purposes. Your task is to rewrite and enhance the reasoning process by introducing natural exploration, thoughtful consideration, and a more human-like flow of ideas, while still arriving at the correct solution. Please put the puzzle's final answer in `<answer></answer>` tags at the end.
>
>  ## Input
>
>  ### Original Problem Statement:
>  [problem_content_placeholder]
>
>  ### Current Reasoning Process:
>  [reasoning_content_placeholder]
>
>  ## Output Requirements
>
>  Please rewrite the reasoning process to be more natural and exploratory, without revealing the final answer until you derive it through your own reasoning.
> ```

---

> ### Author Response · Authors · 2025-11-21
> **Official Comment by Authors for Reviewer cRVP (4/7)**
>
> > ***Weakness Dataset 2: There is no statistical breakdown of the proportions of different task types within the dataset.***
>
> Thank you for this valuable feedback.
>
> MM-HELIX-100K encompasses 42 tasks across  games, puzzles, graphs and alogrithms problem.
> Below is a detailed statistical breakdown of the task types:
>
> Task Distribution in MM-HELIX-100K
>
> | Task                       | Count | Task                         | Count | Task                       | Count |
> |----------------------------|-------|------------------------------|-------|----------------------------|-------|
> | slidingpuzzle              | 3497  | ContainerWithMostWater       | 3000  | HIndex                     | 3000  |
> | Kukurasu                   | 3000  | Calcudoku                    | 3000  | LargestRectangleInHistogram| 3000  |
> | LongestIncreasingSubsequence| 3000  | CountHillsAndValleys         | 2998  | 24Points                   | 2996  |
> | CryptoMath                 | 2993  | TrappingRainWater            | 2978  | bridges                    | 2910  |
> | Skyscrapers                | 2908  | BestTimeToBuyAndSellStock    | 2896  | sudoku                     | 2700  |
> | snake                      | 2546  | kakuro                       | 2500  | maze                       | 2495  |
> | hitori                     | 2486  | tapa                         | 2483  | wordsearch                 | 2477  |
> | futoshiki                  | 2476  | aquarium                     | 2462  | binairo                    | 2454  |
> | eulero                     | 2453  | nibbles                      | 2445  | sokoban                    | 2410  |
> | shortest_distance_weighted | 2300  | topological_sort             | 2298  | minesweeper                | 2218  |
> | numbrix                    | 2386  | max_flow                     | 2531  | eulerian_cycle             | 2511  |
> | campsite                   | 2502  | WordLadder                   | 2917  | eulerian_path              | 2902  |
> | nonogram                   | 2857  | hamiltonian_cycle            | 2847  | shingoki                   | 2841  |
> | hamiltonian_path           | 2417  | hanoi                        | 272   |
>
> > ***Weakness Experiment 1: In Section A.8.2, the improvement on Graph-type tasks is smaller than on other types, but there is no accompanying analysis.***
>
> Thank you for pointing out the relatively attenuated performance gain on the Graph-type tasks.
>
> We hypothesize that this marginal enhancement is primarily due to a confluence of inherent challenges specific to graph and  summarize two primary mitigating factors:
>
> 1. Graph visualizations often involve complex, non-standardized layouts where lines and nodes intersect irregularly. This inherent complexity imposes significant demands on the model’s visual perception module. The model may struggle with accurately parsing the topological structure from the visual input, particularly when dealing with large or dense graphs. Consequently, errors at the perception stage propagate and limit the potential benefit of our high-level reasoning algorithm.
> 2. A notable subset of Graph-type problems, such as determining the existence of a Hamiltonian Path, are fundamentally binary decision tasks (Yes/No). While we require the generation of the constructive solution upon a 'Yes' response, the primary reward signal remains anchored to the binary correctness. This structure introduces a higher potential for superficial reward hacking where the model might optimize for the correct classification without achieving a deep, generalizable understanding of the underlying graph algorithms. The resulting policy may thus be less robust than those developed for tasks requiring detailed, multi-step generative reasoning.
>
> For future work, we plan to specifically address these issues by exploring more specialized graph perception mechanisms and developing finer-grained, intermediate reward functions tailored for sequential decision-making in graph traversal and analysis tasks. Thank you again for this valuable observation.
>
> > ***Weakness Experiment 2: The base model used for training should be explicitly stated in the  experimental setup, even if readers can infer it from Table 2.***
>
> Thank you for this constructive feedback regarding the explicit statement of our foundational model(Qwen2.5-VL-7B-Instruct). We will immediately revise the manuscript to explicitly state this detail in the designated section.
>
> Thank you for helping us enhance the precision of our experimental description!

---

> ### Author Response · Authors · 2025-11-21
> **Official Comment by Authors for Reviewer cRVP (5/7)**
>
> > ***Weakness Experiment 3: In Figures 6 and 7, why is only static-AHPO compared with other  algorithms, while Figure 7 also includes a non-static version(AHPO).  Does static-AHPO mean the activation coefficient = 0?***
>
> Thank you for highlighting the potential ambiguity concerning the comparison variants in Figures 6 and 7. We apologize for the lack of explicit definition in the caption and appreciate the opportunity to clarify this aspect of our experimental design.
>
> We confirm your interpretation: Static-AHPO is indeed a special, non-adaptive instance of the full AHPO framework, where the activation coefficient = 0. This configuration ensures that the off-policy expert loss is applied uniformly across all training steps, without the dynamic, uncertainty-based modulation that characterizes the full AHPO algorithm.
>
> Figure 6 is designed to isolate and demonstrate the fundamental benefit of directly incorporating an off-policy expert loss against existing methodologies (LUFFY and GRPO). By using Static-AHPO, we compare the efficacy of our core learning objective against the baselines before introducing the complexity of dynamic adaptation.
>
> Figure 7's primary objective is to perform a critical ablation study to empirically validate the necessity of the dynamic activation mechanism. By comparing the performance of the full AHPO against Static-AHPO, we clearly illustrate the robust training process gained by the adaptive application.
>
> We will ensure that the revised manuscript explicitly defines Static-AHPO in the Methodology section and clarifies its relationship to the full AHPO framework, thanks again for your suggestions!
>
> > ***Weakness Experiment 4: In Appendix A.2 (Training Setting), why are only 22k samples used  for SFT instead of the full dataset? Is there any comparison showing the  relationship between data volume and performance, and how were these  22k samples selected?***
>
> Thank you for your pertinent questions.
>
> **Data Selection**
>
> The decision to utilize 22k samples was driven by both computational resource constraints and strategic data prioritization:
>
> - The primary limiting factor was the disproportionate and computationally prohibitive cost for the Reinforcement Learning stage when handling excessively long Chain-of-Thought (CoT) trajectories. Consequently, we implemented a practical constraint, filtering the MM-HELIX dataset to include only those samples whose combined input-output sequence length was less than 8192 tokens. This constraint was critical for maintaining a feasible training timeline and cost, particularly during the iterative RL optimization phase.
>
> - Beyond the length constraint, the final 22k set was also curated based on a principled data balancing strategy. We adjusted the proportional representation of different task categories to align with the observed baseline learning difficulty of the initial model. Tasks that were demonstrably more challenging for the base model were allocated a slightly higher proportion.
>
> **Data Volume Comparison**
>
> To empirically compare the performance of different data volume, we conducted an ablation study comparing the performance achieved using 5k, 13k and 22k samples. The results summarized in table below clearly demonstrate a positive correlation: the model trained on the larger scale exhibited superior performance.
>
> | Data Scale| MM-HELIX |
> |--------|--------|
> | 5k | 12.4 |
> | 13k | 19.4 |
> | 22k | 23.8 |

---

> ### Author Response · Authors · 2025-11-21
> **Official Comment by Authors for Reviewer cRVP (6/7)**
>
> > ***Weakness Experiment 5: Why is math data included during the RL stage? Was the AHPO model in  Table 2 also trained with math data? If so, do benchmarks such as  MathVision and MathVerse fall into the in-domain category? If that’s the  case, does the claimed generalization ability still hold? Also, is  there any comparison between models trained with and without math data?***
>
> Thank you for the question. We appreciate the opportunity to clarify this aspect of our experimental design.
>
> **1. Rationale for Including math dataset MMK12 in the RL Stage:**
>
> MMK12 is a math-specific dataset for RL stage. MMK12 strictly contains only Question and Ground Truth Answer pairs; it does not have the Chain-of-Thought (CoT) trajectories or reflective steps required for the core mechanism of our AHPO approach.
>
> *Therefore, the AHPO training conducted solely on MMK12 is functionally equivalent to the GRPO baseline.* Our primary experimental objective for including it was to empirically validate that AHPO can effectively generalize the reflective capabilities learned from the non-mathematical MM-HELIX dataset expert data to the mathematical reasoning domain where no domain-specific reflection data was provided.
>
> **2. Generalization Claim and In-Domain Status:**
>
> The AHPO model presented in Table 2 was indeed trained on the mixed dataset (MMK12 + MM-HELIX-100K). However, the crucial distinction for the generalization claim is the source of the reflective CoT signal. MM-HELIX dataset serves as the source domain for reflection guidance.
>
> Furthermore, we meticulously ensured that there is no content overlap between the samples used for training in MM-HELIX dataset and the problems in the general reasoning evaluation benchmarks(MathVision and MathVerse). Thus, these benchmarks remain out-of-distribution with respect to the expert guidance signal, and our claim of generalization remains valid.
>
> **3. Empirical Comparison (With and Without Math Data):**
>
> The comparison between the model trained only on MMK12 only and the AHPO model trained on the mixed dataset (MMK12 + MM-HELIX) is demonstrated in the table below.
>
> The AHPO model utilizing the mixed dataset significantly surpasses the MMK12-Only baseline. This robust performance differential provides strong empirical evidence that the high-quality, reflective CoT knowledge derived from the non-mathematical MM-HELIX-100K dataset is the essential supervisory signal that empowers AHPO to generalize sophisticated reasoning capabilities across different domains, an outcome unachievable by standard RL on MMK12 alone.
>
> | Model                   | MM-HELIX | MathVision | MathVerse-V | LogicVista | WeMath |
> |-------------------------|----------|------------|-------------|------------|--------|
> | Qwen2.5VL-7B            | 6.3      | 25.2       | 40.5        | 45.6       | 34.5   |
> | + MMK12 Only(AHPO=GRPO) | 5.5      | 26.0       | 44.0        | 43.2       | 35.2   |
> | + MM-HELIX Only         | 24.4     | 23.2       | 41.9        | 48.3       | 36.5   |
> | + Mixed                 | 24.9     | 26.6       | 47.5        | 53.5       | 41.1   |

---

> ### Author Response · Authors · 2025-11-21
> **Official Comment by Authors for Reviewer cRVP (7/7)**
>
> > ***Weakness Experiment 6: [Minor] The paper lacks ablation studies on different success rate thresholds and activation coefficient.***
>
> Thank you for this excellent suggestion.
>
> We conducted an ablation study focusing on the impact of different threshold values for $\hat{R}$ while holding the other parameters constant: the roll-out batch size was set to $N=5$, and the training was conducted for 500 AHPO steps on 15k training samples. The results below report the performance:
>
> | Threshold| Algorithm | Graph| Game | Puzzle | Overall |
> |--------|--------|------------|-------|---------|-----------|
> | 0 | 13.0 | 15.0 | 4.4 | 5.4 | 8.7 |
> | 1 | 21.1 | 17.9 | 20.9 | 18.8 | 20.4 |
> | 2 | 23.7 | 18.8 | 12.6 | 20.1 | 20.1 |
> | 3 | 15.1 | 16.2 | 4.4| 6.5 | 8.9 |
> | 4 | 5.6 | 15.8 | 4.4 | 6.8 | 7.9 |
> | 5 | 11.5 | 18.8 | 3.3 | 4.4 | 8.5 |
>
> We have several observations:
> 1. When $\hat{R}=0$, the performance is significantly lower. This confirms that the off-policy data is essential for effective learning, particularly when the underlying policy struggles with sparse rewards in the pure RL baseline.
> 2. Setting the threshold too high($\hat{R}>=3$) leads to a sharp drop in performance. At high $\hat{R}$ values, the dynamic coefficient is rarely activated, which imposes the off-policy expert loss on the policy only when it is already performing extremely well. This effectively forces the model to over-regularize towards the expert data, suppressing necessary exploration and resulting in training instability (manifested as repetitive or degenerate outputs).
> 3. The thresholds $\hat{R}=1$ and $\hat{R}=2$ yield the most robust and highest performance, indicating that a moderate level of on-policy success is required before the expert guidance is dynamically engaged.
>
> Thank you again for suggesting this important dimension of analysis.
>
> > ***Weakness Experiment 7: [Minor] Is there any comparison showing how the CoT length in MM-HELIX-100K affects reasoning performance?***
>
> Thank you for this highly relevant question.
> We conducted an ablation study and compared two key thresholds for the input-output sequence length: 4096 tokens and 8192 tokens using 15k training data on Qwen2.5-VL-7B.
>
> | Context Len| algorithm | graph| game | puzzle | overall |
> |--------|--------|------------|-------|---------|-----------|
> | 4096 | 18.8 | 17.5 | 16.6 | 13.8 | 17.5 |
> | 8192 | 21.1 | 17.9 | 20.9 | 18.8 | 20.4 |
>
> The empirical results, summarized in the table above, clearly indicate that the model trained with the 8192-token sequence length constraint exhibited superior performance compared to the 4096-token constraint.
>
> We propose that this improvement stems from a direct correlation between task complexity and solution length: longer CoT trajectories are typically an inherent characteristic of more complex, multi-step reasoning tasks. By increasing the maximum sequence length to 8192 tokens, we effectively incorporate a larger proportion of these high-difficulty samples into our training distribution. Consequently, the model is exposed to and learns from more intricate reflective steps, which ultimately enhances its overall generalization and reasoning capabilities across the full spectrum of task difficulty.

---

> > ### Author Response · Authors · 2025-11-27
> >
> > Dear Reviewer cRVP,
> >
> > We hope this message finds you well. We wanted to follow up on the rebuttal we submitted for our ICLR submission.
> >
> > We know the review period keeps you busy, and we really appreciate the time you've put into reviewing our paper. We've worked hard to address the concerns from your review. Whenever you have a moment, we'd love to hear what you think about our responses.
> >
> > If you need us to clarify anything else, just let us know. Thanks so much for your time.
> >
> > Best regards,
> >
> > Authors

---

> > > ### Comment · Reviewer_cRVP · 2025-11-28
> > >
> > > Thank you for the response. The additional explanations and experiments have addressed my concerns. Please make sure to add all supplementary information and experiments into the final version of the manuscript. In addition, I suggest that the authors include results on the full 100k dataset in the future work to verify whether performance continues to improve consistently at a larger data scale.
> > >
> > > At the moment, the openreview is experiencing an issue that prevents me from updating the score. Once this issue is solved, I will raise my score to 6.
> > >
> > > Best

---

### Official Review · Reviewer_3idK · 2025-11-03

**Soundness:** 3
**Presentation:** 3
**Contribution:** 3
**Rating:** 6
**Confidence:** 2

**Summary:**

This paper addresses the deficiency of current Multimodal Large Language Models (MLLMs) in long-chain reflective reasoning. The authors make three main contributions: (1) MM-HELIX, a benchmark consisting of 1,260 samples across 42 challenging tasks designed to evaluate multimodal reflective reasoning capabilities; (2) MM-HELIX-100K, a large-scale dataset of 100k high-quality reflective Chain-of-Thought (CoT) traces generated via the Step-Elicited Response Generation (SERG) pipeline; and (3) Adaptive Hybrid Policy Optimization (AHPO), a novel training algorithm that dynamically integrates off-policy expert guidance with on-policy exploration. When applied to Qwen2.5-VL-7B, their method achieves +18.6% accuracy improvement on MM-HELIX and +5.7% average performance gain on general mathematics and logic tasks, demonstrating effective learning and generalization of reflective reasoning capabilities.

**Strengths:**

- Novel focus on end-to-end multimodal reflective reasoning, going beyond existing benchmarks that use multiple-choice formats
- Rigorous benchmark construction with automated verification ensuring correctness
- Well-structured paper with clear methodology sections
- Practical improvements on both in-domain and out-of-domain tasks

**Weaknesses:**

- Missing analysis of under what conditions AHPO would fail or succeed
- Limited error analysis showing what types of mistakes the model makes
- Training mix includes both MM-HELIX-100K and MMK12; unclear how much each contributes
- No analysis of diversity in generated CoT traces (might all be similar)

**Questions:**

- Can you ablate training on MM-HELIX-100K only vs. MMK12 only to understand each dataset's contribution?
- Can AHPO work with smaller base models (e.g., 1B-3B parameters)?

---

> ### Author Response · Authors · 2025-11-21
> **Official Comment by Authors for Reviewer 3idk (1/2)**
>
> We sincerely appreciate your constructive feedback and valuable suggestions.
>
> > ***Weakness 1: Missing analysis of under what conditions AHPO would fail or succeed***
>
> We appreciate the suggestion for a deeper analysis of AHPO.
>
> We posit that AHPO demonstrates significant efficacy and is most impactful in scenarios where the baseline Reinforcement Learning model struggles to converge or generates sub-optimal solutions due to sparse rewards or an expansive, challenging state-action space. The core strength of AHPO lies in its ability to inject expert-derived trajectories and guide the exploration process, thereby effectively bootstrapping the learning of complex, high-quality policies. This mechanism is essential for overcoming inherent difficulties in the initial RL optimization phase.
>
> Conversely, we anticipate that AHPO's incremental benefit diminishes when meeting easily solvable environments. In environments where the base RL algorithm can rapidly and reliably achieve near-optimal performance, the overhead introduced by the AHPO mechanism—namely, the expert policy generation and integration—may not yield a commensurate increase in performance, making it less resource-effective.
>
> We acknowledge the necessity of formally defining and empirically validating these boundary conditions, which we have identified as a key priority for our future work. Thank you again for this valuable suggestion.
>
> > ***Weakness 2: Limited error analysis showing what types of mistakes the model makes***
>
> Thankyou for pointing out the need for a deeper error analysis. We have conducted a qualitative examination of the model's incorrect outputs and categorized the primary failure modes into four distinct classes:
>
> 1. Perception Errors:
> This occurs when the model fails to correctly interpret the raw visual state.
>
> Example: In Maze task, the model fails to identify the presence of a wall at a specific coordinate, leading to invalid path planning proposals that traverse obstacles.
>
> 2. Hallucination (Reasoning-Observation Mismatch):
>
> Here, the model generates conclusions that contradict its own intermediate observations, effectively forcing a result to match a prior expectation rather than the evidence.
>
> Example: The model explicitly identifies a character sequence at the predicted coordinates as "L, N, O, I" (Reasoning trace: “So letters are L, N, O, I... that's LION!”). Crucially, the model ignores the clear mismatch between the observed sequence ("LNOI") and the target word ("LION"), hallucinating a successful anagram solution despite explicit contradictory evidence.
>
> 3. Insufficient Backtracking:
>
> When facing conflicting constraints, the model tends to attempt local fixes rather than performing valid backtracking to reset earlier variables. This often results in the model settling for an invalid approximate solution.
>
> Example: In 5×5 Kukurasu Puzzle, The model successfully satisfied all row sum constraints but failed the column sum constraints (e.g., the predicted Column 1 sum was 9, whereas the target was 10). Although the model explicitly identified the conflict during final verification (“Required 10. Close, but not enough”), it failed to backtrack and revise the earlier row-level decisions that caused this downstream conflict.
>
> 4. Instruction Following Failures(Minor):
>
> The model occasionally violates explicit hard constraints, particularly negative constraints provided in the system prompt.
>
> Example: In Word Ladder Task, The prompt explicitly forbids adding or deleting letters (only substitution is allowed). The model violated this constraint by transitioning from "cappy" (5 letters) to "clappy" (6 letters), disregarding the length invariance requirement.

---

> ### Author Response · Authors · 2025-11-21
> **Official Comment by Authors for Reviewer 3idk (2/2)**
>
> > ***Weakness 3 & Question 1: Training mix includes both MM-HELIX-100K and MMK12; unclear how much each contributes***
>
> Thank you for this insightful question concerning the contribution of each dataset within our mixed training strategy.
>
> We first wish to clarify the intrinsic differences between the two datasets:
> 1. MMK12 is a math-specific dataset primarily used for Reinforcement Learning. It contains only Question and Ground Truth answers, lacking the Chain-of-Thought trajectories necessary for our AHPO method's reflection mechanism. Training solely on MMK12  thus functionally aligns with the GRPO baseline.
> 2. MM-HELIX dataset(ours) is an auxiliary dataset providing high-quality, multi-modal reflective CoT trajectories across general reasoning tasks, with no overlap with the mathematical content of MMK12.
>
> To directly address the contribution of MM-HELIX dataset and MMK12, we conducted an ablation study comparing two key scenarios (as detailed in Table below):
>
> | Model                   | MM-HELIX | MathVision | MathVerse-V | LogicVista | WeMath |
> |-------------------------|----------|------------|-------------|------------|--------|
> | Qwen2.5VL-7B            | 6.3      | 25.2       | 40.5        | 45.6       | 34.5   |
> | + MMK12 Only(AHPO=GRPO) | 5.5      | 26.0       | 44.0        | 43.2       | 35.2   |
> | + MM-HELIX Only         | 24.4     | 23.2       | 41.9        | 48.3       | 36.5   |
> | + Mixed                 | 24.9     | 26.6       | 47.5        | 53.5       | 41.1   |
>
> The results clearly demonstrate that our AHPO utilizing the mixed dataset significantly surpasses the GRPO baseline trained only on MMK12. This empirical evidence strongly indicates that the MM-HELIX dataset, via its high-quality reflective CoT, contributes the essential supervisory signal for AHPO to effectively learn and generalize sophisticated reflection and reasoning capabilities beyond the scope achievable with standard RL and MMK12 alone.
>
> We will incorporate this detailed clarification regarding the function and non-overlapping nature of the datasets into the revised manuscript to enhance clarity for the reviewers.
>
> > ***Weakness 4: No analysis of diversity in generated CoT traces (might all be similar)***
>
> Thank you for your insightful comment.
>
> Our generated CoT traces are not produced by a single fixed template. Instead, it originate from rule-based CoT generators that are task-dependent. Each task in MM-HELIX (42 tasks across algorithms, graphs, puzzles, and games) has its own rule-based CoT generation logic, reflecting the nature of the task.
>
> Moreover, within the same task, the rule-based CoT generator utilizes multiple heuristic strategies, and a heuristic is randomly selected when constructing each trajectory. These heuristics include different exploration orders, alternative decomposition paths. This stochastic heuristic selection ensures that the generated CoT traces are not deterministic and exhibit meaningful intra-task diversity.
>
> In our SERG pipeline, the rule-based CoT then serves only as a scaffold: a strong LLM further rewrites and refines it into a natural, human-like reasoning process. This enhancement stage introduces additional linguistic and stylistic diversity beyond the structured rule-based steps.
>
> Therefore, CoT traces in our dataset exhibit substantial diversity:
>
> - **Inter-task diversity** from heterogeneous rule-based reasoning structures;
> - **Intra-task diversity** through randomized heuristic selection;
> - **Linguistic diversity** from LLM refinement.
>
> This combination ensures that our MM-HELIX-100K CoT data is not repetitive or template-like, but instead diverse and robust across all tasks.
>
> > ***Question 2: Can AHPO work with smaller base models (e.g., 1B-3B parameters)?***
>
> Thank you for this highly pertinent question regarding the scalability of AHPO.
>
> We conducted an experiment utilizing the Qwen2.5-VL-3B model as the base architecture, applying our AHPO methodology and training with the MM-HELIX dataset. The  performance is summarized below:
>
> | Model                | MM-HELIX | MathVision | MathVerse-V | LogicVista | WeMath |
> |----------------------|----------|------------|-------------|------------|--------|
> | Qwen2.5VL-3B         | 3.5      | 21.5       | 31.5        | 40.0       | 23.1   |
> | MM-HELIX-3B-Thinking | 13.1     | 22.3       | 35.0        | 45.2       | 26.7   |
>
> The empirical evidence demonstrates that AHPO successfully enhances the performance of the smaller 3B-parameter model across all tested reasoning benchmarks. While the improvements are statistically significant, the magnitude of the performance gain is indeed less pronounced compared to the 7B model. Smaller models possess a lower intrinsic capacity to fully internalize the complex, long-chain reflective thought processes.
>
> We will ensure these results and this analysis are included in the revised manuscript to provide a complete view of the algorithm's scalability.

---

### Author Response · Authors · 2025-11-27
**General Response**

Dear AC and Reviewers,

Thank you for taking on this paper and for being invited to participate in our discussion with the reviewers. We understand that the special circumstances this year have placed an additional burden on ACs, and we genuinely appreciate your time and effort. To help you efficiently assess our work, we provide a concise summary of our contribution, the reviewers' evaluation, and the key improvements made during rebuttal.

**(I) Main Contribution**

This paper focuses on the long-chain reflective capabilities of current Multimodal Large Language Models (MLLMs). First, we construct the **MM-HELIX benchmark**, comprising 42 tasks that necessitate iterative multimodal reasoning, and provide a comprehensive evaluation of current MLLMs. Observing the inefficiency of existing models, we propose **Step-Elicited Response Generation (SERG)** to effectively synthesize high-quality iterative CoT data, resulting in the **MM-HELIX-100k dataset**. Furthermore, recognizing the limitations of GRPO and SFT, we propose the **Adaptive Hybrid Policy Optimization (AHPO)** method. With MM-HELIX-100k and AHPO, our model not only achieves superior performance on in-domain benchmarks (+18.6) but also generalizes reflective capabilities to general domains (+5.7).

**(II) Strengths Recognized by Reviewers**

We are glad that reviewers have recognized the strengths of our work:
- Novelty in end-to-end multimodal reflection (@3idK, EG1J)
- Rigorous benchmark construction (@3idK, cRVP, onUL)
- Practical improvements via AHPO (@3idK, cRVP, onUL, EG1J)
- Efficient and high-quality data pipeline SERG (@onUL, EG1J)
- Well-structured presentation (@3idK, EG1J)

**(III) Key Improvements Made During Rebuttal**

*1) Discussion on AHPO Applicability (Reviewer 3idK)*

We have elaborated on which scenarios are best suited for AHPO and identified specific cases where it may be less effective.

*2) Failure Analysis of Evaluated Models (Reviewers 3idK, EG1J)*

We provide a comprehensive analysis of why models fail to answer specific questions within the MM-HELIX benchmark.

*3) Ablation of Training Data Components (Reviewers 3idK, cRVP)*

We conducted ablation studies on each training data component to validate the specific contributions of the MM-HELIX dataset and AHPO.

*4) Diversity of the MM-HELIX Dataset (Reviewer 3idK)*

We have added a discussion regarding the variety and diversity of samples in the MM-HELIX dataset.

*5) Scalability Validation: AHPO on a 3B Model (Reviewer 3idK)*

We conducted experiments on Qwen2.5-VL-3B, with results further verifying the effectiveness of our method.

*6) Evaluation Settings for MM-HELIX (Reviewers cRVP, EG1J)*

We clarified the evaluation settings for both text-only and multimodal tasks, detailed the definitions of difficulty levels, and explained how verifiers are constructed.

*7) Task Proportions in MM-HELIX (Reviewer cRVP)*

We provided a statistical table detailing the proportion of each task type within the dataset.

*8) Details of the SERG Pipeline (Reviewer cRVP)*

We included the specific prompts used and provided a detailed walkthrough of the SERG pipeline.

*9) Analysis of Training Outcomes (Reviewer cRVP)*

We analyzed why graph-related tasks show smaller improvements compared to other tasks and clarified the confusion surrounding Figure 6 and Figure 7.

*10) Ablation Study of Threshold Coefficients (Reviewers cRVP, onUL, EG1J)*

We conducted ablation experiments to justify the choice of threshold coefficients in AHPO.

*11) Selection of the 22k Training Dataset (Reviewers cRVP, onUL)*

We explained the criteria and process for selecting the 22k training samples.

*12) Ablation Study on Context Length (Reviewer cRVP)*

We conducted experiments comparing different context lengths (4k vs. 8k).

*13) Generalization of the MM-HELIX Dataset (Reviewer EG1J)*

We clarified the zero-shot evaluation settings and provided examples demonstrating how reflection generalizes to broader domains.

*14) Details of Training Settings (Reviewer EG1J)*

We provided comprehensive details regarding the specific training configurations for AHPO.

*15) Efficacy of Reflection Data (Reviewer EG1J)*

We cited relevant research validating the effect of reflection data and detailed our process for building CoT with reflection.

**(IV) Reviewer Endorsement**

We are pleased that although Reviewer cRVP initially raised questions regarding our benchmark, dataset and experiments, our comprehensive response addressed these concerns, **leading Reviewer cRVP to raise their score from 4 to 6**.

Our main modifications have been incorporated into the revised manuscript, particularly in Section 4 (Experiments) and the Appendix. We hope this summary facilitates your evaluation.

Best regards,

The Authors

---

### Meta-Review · Area_Chair_dzTc · 2026-01-06

**Summary:**

This submission receives four ratings of 8, 6, 6, 4. The reviewers identify the contribution of this work as: a multimodal benchmark MM-HELIX and a large-scale verified CoT dataset MM-HELIX-100K. The rating scores are generally positive, while the major concern is about the reproducibility and the limited depth of its analysis. Other issues are raised, including the benchmark data construction details, insufficient ablations, which are mostly addressed during the rebuttal phase. Overall, this work receives positive ratings on average. Though some issues remain unaddressed, they are not the decisive factor to reject this work.

**Reviewer Concerns:**

The following issues are raised but mostly addressed during rebuttal: the benchmark data construction details, insufficient ablations, data mixing strategy, hyperparameter sensitivity.

The concerns that need further clarification include: the reproducibility and the limited depth of its analysis, 100K training results.

**Reviewer Scores:**

Reviewer 3idK: 6
Reviewer onUL: 8
Reviewer EG1J: 6
Reviewer eRVP: 4 (raised to 6)

Given all ratings are towards positive, the scores are not likely to fall during rebuttal.

---

### Decision · Program_Chairs · 2026-01-26

Accept (Poster)